# Analysis of atmospheric ammonia over South and East Asia based on the MOZART-4 model and its comparison with satellite and surface observations

Pooja V. Pawar[1*], Sachin D. Ghude[1], Chinmay Jena[1], Andrea Móring[2,7], Mark A. Sutton[2], Santosh Kulkarni[3], Deen Mani Lal[1], Divya Surendran[4], Martin Van Damme[5], Lieven Clarisse[5], Pierre-François Coheur[5], Xuejun Liu[6], Gaurav Govardhan[1,8], Wen Xu[6], Jize Jiang[7], and Tapan Kumar Adhya[9]

[1]Indian Institute of Tropical Meteorology (IITM), Ministry of Earth Sciences, Pune-411008, India
[2]Centre for Ecology & Hydrology (CEH), Edinburgh, EH26 0QB, UK
[3]Centre for Development of Advanced Computing, Pune-411008, India
[4]Indian MeteorologicalDepartment (IMD), Ministry of Earth Sciences, Pune-411005, India
[5]Université libre de Bruxelles (ULB), Spectroscopy, Quantum Chemistry and AtmosphericRemoteSensing
 (SQUARES), Brussels, B-1050, Belgium
[6]College of Resources and Environmental Sciences, National Academy of Agriculture Green
 Development, China Agricultural University, Beijing 100193, China
[7]The University of Edinburgh, Scotland, EH8 9AB, UK
[8]National Centre for Medium Range Weather Forecasting, Noida, Uttar Pradesh, India
[9]Kalinga Institute of IndustrialTechnology, Bhubaneshwar, 751016, India

*Correspondence to*: Sachin D. Ghude (sachinghude@tropmet.res.in)

**Abstract.** Limited availability of atmospheric ammonia ($NH_3$) observations, limits our understanding of controls on its spatial and temporal variability and its interactions with ecosystems. Here we used the Model for Ozone and Related chemical Tracers (MOZART-4) global chemistry transport model and the Hemispheric Transport of Air Pollution version-2 (HTAP-v2) emission inventory to simulate global $NH_3$ distribution for the year 2010. We presented a first comparison of the model with monthly averaged satellite distributions and limited ground-based observations available across South Asia. The MOZART-4 simulations over South Asia and East Asia were evaluated with the $NH_3$ retrievals obtained from the Infrared Atmospheric Sounding Interferometer (IASI) satellite and 69 ground based monitoring stations for air quality across South Asia, and 32 ground based monitoring stations from the Nationwide Nitrogen Deposition Monitoring Network (NNDMN) of China. We identified the northern region of India (Indo-Gangetic Plain, IGP) as a hotspot for $NH_3$ in Asia, both using the model and satellite observations. In general, a close agreement was found between yearly-averaged $NH_3$ total columns simulated by the model and IASI satellite measurements over the IGP, South Asia (r = 0.81) and North China Plain (NCP), of East Asia (r = 0.90). However, the MOZART-4 simulated $NH_3$ column was substantially higher over South Asia than East Asia, as compared with the IASI retrievals, which show smaller differences. Model simulated surface $NH_3$ concentrations indicated smaller concentrations in all seasons than surface $NH_3$ measured by the ground based observations over South and East Asia, although uncertainties remain in the available surface $NH_3$ measurements. Overall, the comparison of East Asia and South Asia using both MOZART-4 model and satellite observations showed smaller $NH_3$ columns in East Asia compared with South Asia for comparable emissions, indicating rapid dissipation of $NH_3$ due to secondary aerosol formation, which can be explained by larger emissions of acidic precursor gases in East Asia.

## 1. Introduction

Gaseous pollution due to various forms of nitrogen emissions plays an important role in environmental processes. Specifically, ammonia ($NH_3$) emitted from various agricultural activities, such as use of synthetic fertilizers, animal farming, etc., together with nitrogen oxides (NOx) is one of the largest sources of reactive nitrogen (Nr) emission to the atmosphere. Ammonia has great environmental implications due to its substantial influence on the global nitrogen cycle and associated air pollution, ecosystem and on public health (Behera et al., 2013; Liu et al., 2017b; Zhou et al., 2016). Emission estimates provided by latest EDGAR v4.3.2 emission inventory suggests that globally about 59 Tg of $NH_3$ was emitted in the atmosphere in 2012 out of which direct soil emissions contributed about 56 %, manure management (on farm) contributed about 19 %, and agricultural burning contributed about 1.5 % while biomass burning contribution is not included in emission estimate. Furthermore, due to lack of observed emission factors and high uncertainty of agricultural statistics, the uncertainty of $NH_3$ is the largest among all other pollutants in EDGAR v4.3.2 (Crippa et al., 2018).Ammonia is a key precursor in aerosol formation, as the reactions in the atmosphere lead to an increase in different forms of sulphates and nitrates that contribute in secondary aerosol formation (Pinder et al., 2007, 2008). India and China together accounted for an estimated 64 % of the total amount of $NH_3$ emissions in Southern Asia during 2000-2014 (Xu et al., 2018).Emissions of $NO_x$ and $NH_3$ are increasing substantially over South Asia (Sutton et al., 2017b, 2017a), which contributes to increase in particulate mass loading, visibility degradation, acidification and eutrophication (Behera et al., 2013; Ghude et al., 2008, 2013, 2016). Asia is responsible for the largest share of global $NH_3$ emissions (Janssens-Maenhout et al., 2012). Further increase in $NH_3$ emission will increase its negative impacts and societal cost (Sutton et al., 2017b).

In India, around 50 % of total $NH_3$ emissions is estimated from the fertilizer application and remaining from livestock and other $NH_3$ sources (Aneja et al., 2011; Behera et al., 2013). However, there are large uncertainties in emissions of ammonia, its deposition to surface, chemistry and transport (Sutton et al., 2013; Zhu et al., 2015). Urea is mostly used as a fertilizer (Fertlizer Association of India annual report 2018-19) and alone contributes more than 90% of total fertilizer used for the agricultural activities (Sharma et al., 2008). India is currently the second largest consumer of fertilizers after China, and fertilizer usage is bound to increase with further intensification of agriculture and the fertilizer input of India is expected to be doubled by 2050 (Alexandratos and Bruinsma, 2012).

Recent study based on Infrared Atmospheric Sounding Interferometer (IASI) satellite measurements show very high concentration of $NH_3$ over Indo-Gangetic Plain (IGP) and North China Plain (NCP) which were mainly related to agricultural (Van Damme et al., 2014a, 2014b, 2015b)and industrial activity (Clarisse et al., 2019; Van Damme et al., 2018). The seasonality was shown to be more pronounced in the northern hemisphere, with peak columns in spring and summer season (Van Damme et al., 2014a). Van Damme et al., (2015a)attempted first to validate IASI-$NH_3$ measurements using existing independent ground-based and airborne data sets. This study doesn't include comparison of ground-based $NH_3$ data sets with IASI measurements particularly over South Asia (India) due to limited availability of $NH_3$ measurements. Liu et al. (2017a)estimated the ground-based $NH_3$ concentrations over East Asia, combining IASI-$NH_3$ columns and $NH_3$ profiles from MOZART-4 and validated it with forty four sites of Chinese Nationwide Nitrogen Deposition Monitoring Network (NNDMN). In one of the recent study over South Asia, interannual variability of atmospheric $NH_3$ using IASI observations revealed large seasonal variability in atmospheric $NH_3$ concentrations

which were equivalent with highest number of urea fertilizer plants. This study highlights the importance of role
of agriculture statistics and fertilizer consumption/application in determining ammonia concentration in South
Asia (Kuttippurath et al., 2020). Available global ammonia emission inventory does not include a
comprehensive bottom up $NH_3$ emissions for South Asia compared to East Asia to be suitable for input to
atmospheric models by taking into consideration actual statistical data of various $NH_3$ sources such as livestock
excreta, fertilizer application, agricultural soil, nitrogen-fixing plants, crop residue compost, biomass burning,
urine from rural populations, chemical industry, waste disposal, traffic, etc which is currently missing (Behera et
al., 2013; Huang et al., 2012; Janssens-Maenhout et al., 2015; Li et al., 2017; Zhang et al., 2010). Han et al.
(2020) suggested that updated emission inventory as per the source activity is essential for south Asia to reduce
the uncertainties simulated $NH_3$ over this region. A recent study by Wang et al. (2020) examined the $NH_3$
column observed over the IGP during summer using regional model driven with MIX emission inventory. The
study suggested that large agriculture activity and high summer temperature contributes to high $NH_3$ emission
fluxes over IGP which leads to large total columns. Summer time increase in $NH_3$concentration at surface over
certain sites in the IGP regions are also observed from the ground based monitoring network (Datta et al., 2012;
Mandal et al., 2013; Saraswati et al., 2019; Sharma et al., 2012, 2014b).
In this study, we examined the spatio-temporal variability of atmospheric $NH_3$ over Asia (South and
East Asia) and focus on two hotspots regions of ammonia, the Indo-Gangetic Plain (IGP) and the North China
Plain (NCP). The approach for this study is a combination of simulations using chemical transport modelling,
satellite observations and *in-situ* ammonia measurements over South Asia (69 stations) and East Asia (32
stations). The analysis applies the Model for Ozone and Related chemical tracers (MOZART-4) driven by priori
ammonia emissions based on Hemispheric Transport of Air Pollution version-2 (HTAP-v2) emission inventory.
It applies HTAP-v2 data for emissions to produce estimated total columns of $NH_3$ and aerosol species for the
year 2010 over Asia. Model simulations were evaluated and compared with $NH_3$ data from IASI (over South
and East Asia) and selected ground-based observations (noted above). In addition to the regional comparison,
we examine why certain emission hotspot regions in East Asia show lower $NH_3$ total columns compared with
similar hotspot regions in South Asia, when analyzed with both model and satellite observations.

**2. Data and methodology**

**2.1 MOZART-4 model**

The global chemical transport model MOZART-4 has been employed in this study to conduct a year-
long (2010) simulation of atmospheric trace gases and aerosols over Asia using the updated HTAP-v2 emission
inventory (Janssens-Maenhout et al., 2015). These simulations were earlier performed to meet the objectives of
Task Force on Hemispheric Transport of Air Pollution, phase 2, multi-model experiments (Surendran et al.,
2015; Surendran et al., 2016). The model domain covers entire globe at a horizontal grid resolution of 1.9° ×
2.5° and 56 vertical levels from the surface upto 1hectopascal (hPa). The model has approximately 10 levels in
the boundary layer (below 850 hPa). MOZART-4 takes into account surface emissions, convection, advection,
boundary layer transport, photochemistry, and wet and dry deposition. The model simulations were driven by
the input meteorological data set of 1.9° × 2.5° resolution from Modern Era Retrospective-analysis for Research
(MERRA) and Applications of the Goddard Earth Observing System Data Assimilation System (GEOS-DAS).
Model simulations were performed for the complete year of 2010 (1 January 2010 to 31 December 2010) and its
outputs were saved every 6h (4 time steps each day) with a spin up time of six months (1 July 2009 to 31
December 2009). MOZART-4 includes 157 gas-phase reactions, 85 gas-phase species, 39 photolysis and 12
bulk aerosol compounds (Emmons et al., 2010). Dry deposition of gases and aerosols were calculated online
according to the parameterization of Wesely (1989) and wet deposition of soluble gases were calculated as
described by the method of Emmons et al. (2010). Land use cover (LUC) maps used in MOZART-4 are based
on the Advanced Very High Resolution Radiometer (AVHRR) and Moderate Resolution Imaging
Spectroradiometer (MODIS) data based on NCAR Community Land Model (CLM) (Oleson et al., 2010).
MOZART-4 represents the land surface as a hierarchy of sub-grid types: glacier, lake, wetland, urban and
vegetated land. The vegetated land is further divided into a mosaic of Plant Function Type (PFTs). These same
maps are used for the dry deposition calculations (Emmons et al., 2010; Oleson et al., 2010; Lawrence and
Chase, 2007).In MOZART-4 the tropospheric aerosol component is built on the extended work of Tie et al.
(2001 and 2005). Online fast Tropospheric Ultraviolet Visible (FTUV) scheme, based on the TUV model (Tie et
al., 2003) is used for the calculation of photolysis rates in MOZART-4. For long-lived species like $CH_4$ and $H_2$,
surface boundary conditions are constrained by observations from NOAA/ESRL/GMD (Dlugokencky et al.,
2005, 2008; Novelli et al., 1999) and as per Intergovernmental Panel $N_2O$ concentrations are set to the value as
described in Intergovernmental Panel on Climate Change 2000 report (IPCC, 2000). Biogenic emissions of
isoprene and monoterpenes are calculated online using the Model of Emissions of Gases and Aerosols from
Nature (MEGAN) (Guenther et al., 2006), using the implementation described by Pfister et al. (2008). Surface
moisture flux and all relevant physical parameters are used to calculate water vapor ($H_2O$) online. Biomass
burning emissions of a wide range of gaseous components, including $NH_3$, $SO_2$ and individual volatile organic
compounds were provided from the Global Fire Emission Database (GFED-v3), determined by scaling the
GFED $CO_2$ emissions by the emission factors provided on $1.9° \times 2.5°$ grid resolution (Emmons et al., 2010).
In MOZART-4 the ammonium nitrate distribution is determined from $NH_3$ emissions and the parameterization
of gas/aerosol partitioning using equilibrium simplified aerosol model (EQSAM) by Metzger et al. (2002),
which is a set of approximations to the equilibrium constant calculation (Seinfeld et al., 1998), based on the
level of sulphate present. In Metzger et al. (2002) cations other than $NH_4^+$, e.g., sodium ($Na^+$), potassium ($K^+$),
calcium ($Ca^{2+}$), and magnesium ($Mg^{2+}$) as well as organic acids have been neglected for the gas-aerosol
partitioning calculations. Metzger et al. (2006) found that the $NH_3/NH_4^+$ (calculated by account for ammonium-
sulfate-nitrate-sodium-chloride-water system (updated-EQSAM2 parameterization considering organic acids)
was 15 % lower than that calculated from the parameterization similar to EQSAM. Ammonia has stronger
affinity towards neutralization of sulphuric acid ($H_2SO_4$) than nitric acid ($HNO_3$) whereas formation of
ammonium chloride ($NH_4Cl(s)$ or (aq)) in atmosphere is unstable and can dissociate reversibly to $NH_3$ and HCL.
These aerosols in both dry and aqueous phase evaporate faster than the corresponding ammonium nitrate
($NH_4NO_3$) aerosols (Seinfeld and Pandis, 2012). In current modelling setup $NH_3/NH_4^+$ partitioning is mainly
controlled by sulfate and subsequently by nitrate. Recent study (Acharja et al., 2020) based on analysis of water
soluble inorganic chemical ions of $PM_1$, $PM_{2.5}$ and atmospheric trace gases over IGP revealed that $NH_4^+$ was one
of the dominant ions, collectively with $Cl^-$, $NO_3^-$ and $SO_4^-$ constituted more than 95 % of the measured ionic
mass in both $PM_1$ and $PM_{2.5}$. Remaining ionic species (i.e., $Na^+$, $K^+$, $Ca^{2+}$ and $Mg^{2+}$) formed constituted only
about 3% of the total measured ions. Although major mineral cations (i.e., $Na^+$, $K^+$, $Ca^{2+}$ and $Mg^{2+}$) contribute
actively in neutralization reaction, but their concentration in IGP was found to be very low. Also over NCP,
mineral cations contributed less than 5 % in both $PM_1$ and $PM_{2.5}$ (Dao et al., 2014). Furthermore, recent study by
Xu et al. (2017) over East Asia revealed that $NH_4^+$ was the predominant neutralizing cation with the highest
neutralization factor (NF) (above 1), whereas $Na^+$, $K^+$, $Ca^{2+}$ and $Mg^{2+}$ contributed relatively low (below 0.2).
Therefore, consideration of mineral cations and organic acids on the $NH_3/NH_4^+$ partitioning might be limited
and will not have significant impact on the results of this study.
**2.2 Emission inventory (HTAP-v2)**
The HTAP-v2 bottom-up database is used in this study as an input for anthropogenic emissions of $NH_3$ for the
year 2010 (Janssens-Maenhout et al., 2015). HTAP-v2 dataset is embedded with the activity data as per
harmonized emission factors, international standards, and gridded emissions with global proxy data. It includes
important point sources providing high spatial resolution and emission grid maps with global coverage. This
dataset consists of monthly mean $NH_3$ emission maps with $0.1^{\circ} \times 0.1^{\circ}$ grid resolution for the year 2010. The
HTAP-v2 dataset is compiled using various regional gridded emission inventories by Environmental Protection
Agency (EPA) for USA and Environment Canada for Canada, European Monitoring Evaluation Programme
(EMEP) and Netherlands Organisation for Applied Scientific Research for Europe, and Model Inter comparison
Study in Asia (MICS Asia) for China, India and other Asian countries. The emissions Database for Global
Atmospheric Research (EDGARv4.3) is used for the rest of the world (mainly South-America, Africa, Russia
and Oceania). The 'MICS Asia' dataset incorporated into the HTAP-v2 dataset includes an anthropogenic
emission inventory developed in 2010 (Li et al., 2015), which incorporates several local emission inventories,
including the Multi-resolution Emission Inventory for China (MEIC), $NH_3$ emission inventory from Peking
University (Huang et al., 2012) and Regional Emission inventory in Asia version 2.1 (REAS2.1) (Kurokawa et
al., 2013) for areas where local emission data are not available. A detailed description on HTAP-v2 datasets can
be found in Janssens-Maenhout et al. (2015).
For this study, we used emissions from five important sectors, such as, agricultural, residential (heating/cooling
of buildings and equipment/lighting of buildings and waste treatment), energy (power industry), transport
(ground transport) and industries (manufacturing, mining, metal, cement, chemical, solvent industry) for the
year 2010. The aircraft and international shipping is not considered for $NH_3$ emissions in the HTAP-v2 bottom-
up database. These emissions also includes natural emissions such as soil from the Community Earth System
Model (CESM), and biomass burning from the Global Fire Emission Database (GFED-v3) (Randerson et al.,
2013).All these emissions are re-gridded to $1.9^{\circ} \times 2.5^{\circ}$ to match the model resolution.
The spatial distribution of the total $NH_3$ emissions over Asian region is shown in Fig. 1. It shows the highest
emissions over both South and East Asia, especially over the IGP and NCP region (shown with black box in
Fig. 1). Agricultural sector is the main contributor to $NH_3$ emission, including management of manure and
agricultural soils (application of nitrogen fertilizers, including animal waste). It also includes emissions from
livestock, crop cultivation excluding emissions from agricultural waste burning and savannah burning (Janssens-
Maenhout et al., 2015). Minor contributions from the residential sector are also observed for the Asian countries
due to use of biomass combustion and coal burning which is also included in the emissions. Spatial proxies such
as population density, road networks, and land use information have been used to allocate area of emission
sources. For the REAS2 emission inventory over India, the agricultural sector follows spatial proxy of total
population (Li et al., 2017). The use of this approach is expected to be the main source of spatial uncertainty in
the estimated $NH_3$ emissions to the extent that total human population is only approximately correlated with
spatial distribution of fertilizer use and livestock numbers. Seasonal variation of average $NH_3$ emission over the
IGP and NCP region for Anthropogenic (HTAP-v2), biomass burning (GFED-v3) and Soil emission (CESM) is
shown in Fig. 2. Anthropogenic $NH_3$ emissions do not show any strong seasonal variability over the IGP region
however over the NCP region, $NH_3$ emissions show strong seasonality with peak emissions between May-
September months. It can be seen that the magnitude of peak emissions is two times more over the NCP region
than IGP region. On the other hand, seasonality in biomass burning $NH_3$ emissions is strong over the IGP
region, which shows highest emissions in the spring season (MAM). Also, contribution of $NH_3$ emissions from
the IGP region is significantly higher compared to NCP region during peak burning season, but the magnitude
of biomass burning emission is six times lower compared to the magnitude of anthropogenic emissions.
**2.3 Satellite $NH_3$ observations**
The $NH_3$ total columns data used in study are derived from the IASI space-borne remote sensing instrument on
board Metop-A, which was launched in 2006 in a polar sun-synchronous orbit. The IASI operates in the thermal
infrared spectral range (645–2760 $cm^{-1}$) with mean local solar overpass time of 9:30 am and 9:30 pm (Clerbaux
et al., 2009). It covers the globe twice a day with and each observation is composed of 4 pixels with a circular
footprint of 12 Kilometer (km) diameter at nadir and elliptical at the end of the swath (20 × 39 km). IASI is a
suitable tool for evaluation of regional and global models due to its relatively high spatial and temporal
sampling and retrieval algorithms have been continuously improved (Whitburn et al., 2016). The $NH_3$ total
column retrievals show satisfactory agreement with monthly averaged integrated ground-based measurements
with FTIR column data (Van Damme et al., 2015a). IASI measurements are also found to be consistent with
other $NH_3$ satellite products (Clarisse et al., 2010; Someya et al., 2020; Viatte et al., 2020). In present study, we
have used ANNI-$NH_3$-v2.2R-I dataset for the year 2010 which relies on ERA-Interim ECMWF meteorological
input data, along with surface temperature retrieved from a dedicated network (Van Damme et al., 2017). An
improved retrieval scheme for IASI spectra relies on the calculation of a dimensionless "Hyperspectral Range
Index," which is successively converted to the total column and allow a better identification of weak point
sources of atmospheric $NH_3$(Van Damme et al., 2017; Whitburn et al., 2016). More details about IASI satellite
and $NH_3$ data product is given in Clerbaux et al. (2009), Van Damme et al. (2017) and Whitburn et al.
(2016).We have considered the daily $NH_3$ cloud-free satellite total column data and compared with the modelled
daily $NH_3$ total column averaging paired observations across the months, seasons and year. We have used only
morning overpasses at 9:30 am measurements, as the relative errors due to the lower thermal contrast are larger
for the night-time measurements (9:30 pm overpass). For consistency with satellite retrievals, first the model
output (11:30 LT) at each day close to satellite overpass time (9:30 LT) is interpolated in space to the location of
valid satellite retrievals. Since IASI retrieval algorithm only provides total columns, in second step, we made
unweighted average distribution of the daily paired data to obtain a monthly, seasonal and annual mean value of
satellite and model total $NH_3$ columns at each horizontal resolution of the model (1.9° × 2.5°).

**2.4 Ground based observations**

To evaluate model performance in South Asia, we used hourly $NH_3$ measurements from the air quality monitoring station (AQMS) network operated by Central Pollution Control Board (CPCB) across India. CPCB follows a national program for sampling of ambient air quality as well as weather parameters measurements. An automatic analyzer (continuous) method is adopted at each monitoring location. $NH_3$ is measured by the chemiluminescence method as $NO_x$ following oxidation of $NH_3$ to $NO_x$. In this approach, $NH_3$ is determined from the difference between $NO_x$ concentration with and without inclusion of $NH_3$ oxidation (CPCB, 2011). The quality assurance and control process followed for these air quality monitoring instruments is given in CPCB (2014, 2020).Surface observations of $NH_3$ are taken from 69 different stations in South Asia. Most of the $NH_3$ monitoring stations from India used in the current study are situated in the cities representing the urban environment. Sampling of ambient $NH_3$ is done through a sampling inlet of 1 meter (m) above the roof top of container AQMS having height of 2.5 m (Technical specifications, 2019). The details of these monitoring locations are given in Table S1 (in the Supplement) and the geographical locations are shown in Fig. 3. Out of these stations thirty five locations in Delhi, six in Bangalore city, four in Hyderabad, and two in Jaipur city are averaged to get single value for the same geographical location and the remaining 22 locations are considered independently representing 26 respective cities. Hourly $NH_3$ concentrations (in µg m$^{-3}$) used in the study are for the duration of 2016 to 2019. The quality control and assurance method, followed by Central Pollution Control Board (CPCB) for these air quality monitoring stations, is given in the CPCB (2011 and 2020). The calibration procedures for $NH_3$ analyzer conforms to United States Environmental Protection Agency (USEPA) methodologies and include daily calibration checks, biweekly precision checks and linearity checks every six weeks. All analyzers undergo full calibration every six weeks. For detail on calibration procedure refer to Technical Specifications for Continuous Real Time Ambient Air Quality Monitoring Analysers (2016) and CPCB (2020). Furthermore, we take the following steps to reassure the quality of $NH_3$ observations from the CPCB network stations. For data quality, we rejected all the observations values below the lowest detection limit of the instrument (1 µg m$^{-3}$) (Technical specifications for CAAQM station, 2019) because most of the sites are situated in the urban environment. For cities where more than one monitoring station is available, we rejected all the observations above 250 µg m$^{-3}$ at a given site if other sites in the network do not show values outside this range. This step aims to eliminate any short-term local influence that cannot be captured in the models and retain the regional-scale variability. Second, we removed single peaks characterized by a change of more than 100 µg m$^{-3}$ in just one hour for all the data in CPCB monitoring stations. This step filters random fluctuations in the observations. Third, we removed some very high $NH_3$ values that appeared in the time series right after the missing values. For any given day, we removed the sites from the consideration that either experience instrument malfunction, or appear to be very heavily influenced by strong local sources. In order to verify the data quality of CBCB monitoring site, we have inter compared the $NH_3$measurement at CPCB monitoring station (R.K. Puram) in Delhi with the $NH_3$ measurements at Indira Gandhi International (IGI) Airport taken during Winter Fog Experiment (WiFEX) (Ghude et al., 2017) using Measurement of Aerosols and Gases (MARGA) instrument during winter season of 2017-2018. More details on the $NH_3$ measurements using MARGA is available with Acharja et al.(2020). Both sites were situated in the same area of Delhi (less than 1km). Our inter-comparison show that $NH_3$ measured at CPCB monitoring station by chemiluminescence method are slightly (on an average 9.8 µgm$^{-3}$) on higher side than $NH_3$ measured by ion chromatography (IC)

using MARGA (Fig. S1 in the Supplement).The differences that were observed could partly be related to the different $NH_3$ measurement techniques and partly to the locations of the two monitoring sites which were not place exactly at same location. Apparently, the difference of 9.8 µg m$^{-3}$ indicates that the $NH_3$ measurements from the CPCB do not suffer from the calibration issue. However, rigorous validation is required in the future with more data sets. Given the presence of relatively high NOx concentrations, especially at urban locations, it is recognized that the measurement of $NH_3$ by difference (i.e., between $NO_x$ and $NO_x$ plus oxidized $NH_3$), is a potentially significant source of uncertainty. Future measurement inter-comparisons are planned (rescheduled from 2020 to 2021 because of COVID-19) to allow the chemiluminescence method as used in the Indian network to be compared with a range of other $NH_3$ measurement methods (Móring et al., 2021; The Global Challenges Research Fund (GCRF) South Asia Nitrogen hub).

To further evaluate model performance over East Asia, we used monthly mean $NH_3$ measurements from the 32 stations of the Nationwide Nitrogen Deposition Monitoring Network (NNDMN) of China, operated by China Agricultural University. The details of these monitoring locations are given in Table S2 (in the Supplement) and the geographical locations are shown in Fig. 3. Monthly mean $NH_3$ concentrations (in µg m$^{-3}$) used in the study are for the duration of 2010 to 2015. Ambient concentrations of gaseous $NH_3$ were measured using an active Denuder for Long-Term Atmospheric sampling (DELTA) system. More detail about the data product is given by Xu et al. (2019). To compare the model with observation, simulated $NH_3$ from the model are compared with the surface-based observations by using bi-linear interpolation of model output to the geographical location and elevation of the observational sites.

## 3. Results and Discussion

### 3.1 Annual mean $NH_3$ total columns over South Asia

Yearly-averaged 2010 distribution of $NH_3$ total columns over Asia simulated by MOZART-4 model and also retrieved with IASI instrument are shown in Fig. 4a and 4b. The total $NH_3$ columns simulated by the model show high Tropospheric Vertical Column Densities (TVCDs) of about $0.5-7\times10^{16}$ molecules cm$^{-2}$ over IGP region of India compared to any other regions of Asia. This may reflect the larger range of $NH_3$ column values for the South Asian model domain, with both more polluted and cleaner conditions. These high TVCDs values coincide with the high fertilizer-N and livestock numbers, as scaled according to human population density in Fig. 1.

Spatial differences between model simulated data and satellite data for $NH_3$ total column distribution are shown in Fig. 4c. On a quantitative level, the MOZART-4 model is found to overestimates the $NH_3$ total column compared with IASI by $1-4\times10^{16}$ molecules cm$^{-2}$ over South Asia, especially over northeast India and Bangladesh. Conversely, the MOZART-4 model underestimates $NH_3$ in comparison with IASI over the arid region of north western India (state of Rajasthan adjacent to Pakistan) and centering on Pakistan. There are several possible reasons for the spatial differences shown in Fig. 4c, including: a) uncertainties in the mapped $NH_3$ emissions data (e.g., between Afghanistan, Bangladesh, India and Pakistan, due to different relationships between human population and livestock/fertilizer activities); b) uncertainties related to turbulent mixing and dispersion (this may affect both the simulations in MOZART-4 and the assumed vertical profiles for the IASI

retrievals); and c) uncertainties related to precipitation scavenging of ammonia and ammonium, noting that the
eastern part of the IGP is substantially wetter than the western part.
According to Fig. 1, the magnitude of $NH_3$ emissions over NCP is similar to IGP. By contrast, much smaller
TVCDs of the $NH_3$ columns are estimated by MOZART-4 and IASI over NCP compared with IGP. The
MOZART-4 and IASI estimates are found to be in close agreement, with slightly smaller values estimated by
MOZART-4. The possible reasons for the difference in $NH_3$ concentrations in IGP and NCP are discussed in
Sect. 3.4. The relationship between modelled and IASI retrieved $NH_3$ total columns are further analysed in terms
of scatter plots in Fig. 5a and 5b, over IGP region of South Asia (20°N-32°N, 70°E-95°E) and NCP region of
East Asia (30°N-40°N, 110°E-120°E) (rectangular areas shown in Fig. 1). Correlation coefficients (r) between
model and satellite observed annual mean total columns over IGP and NCP are found to be 0.81 and 0.90
respectively for 2010. This indicates that spatial variability in simulated $NH_3$ by the model and satellite
observation is in closer agreement, both over IGP and NCP region. The Model simulated annual mean total $NH_3$
columns gives larger values over IGP region (Normalised Mean Bias (NMB) = 38 %) as well as over entire
South Asia (NMB = 44 %). Whereas over the NCP region (NMB = -35 %) and entire East Asia (NMB = -32 %),
the model gives values which are smaller than IASI. Other statistical indicators are summarised in Table 1.
Larger estimates of $NH_3$ columns from an atmospheric Chemistry Transport Model (CTM) compared with IASI
was also found in an earlier study for South Asia (Clarisse et al., 2009).
The overall higher value of the model simulated $NH_3$ over South Asia compared with IASI could be due to the
combination of the uncertainties in both approaches. This includes uncertainties in emissions from the HTAP-v2
datasets used for the model simulations, inaccurate modelling of the chemistry in MOZART-4, errors in dry and
wet deposition schemes used in the model, and biases inherent to infrared satellite remote sensing. For IASI,
firstly, only cloud-free satellite scenes are processed, which could result in missing partly some of the $NH_3$
values during cloudy periods and biomass burning events. Secondly, $NH_3$ vertical columns retrieved from the
IASI observations are actually sampled around 9:30 local time while the MOZART-4 simulated model output
close to overpass time (11:30 LTC) was used. Finally, the retrieval of $NH_3$ from infrared satellites is sensitive to
inaccuracies in the temperature profile, and biases in the IASI L2 temperature profiles can result in biases in the
retrieved $NH_3$ (Whitburn et al., 2016). The HTAP-v2 dataset use proxy values for agricultural activities (i.e.,
distributed by human population) instead of actual values for field fertilizer application and livestock excretion
over the South Asia. This could also result in additional uncertainty of $NH_3$ emissions from the agricultural
activities. Further work is on-going to integrate $NH_3$ emissions inventories for different countries in South Asia
based on national datasets, which should allow the emissions related uncertainties to be reduced in future.
Similarly, slight underestimation over East Asia might originates from the country specific emission inventory
used for China (Huang et al., 2012) in MOSAIC HTAP-v2 emission inventory and the limitations discussed
above. The application of any equilibrium models (EQMs) in global atmospheric studies is associated with
considerable uncertainties. In MOZART-4 chemistry, the ammonium nitrate distribution is determined from
$NH_3$ emissions and the parameterization of gas/aerosol partitioning by Metzger et al. (2002), based on the level
of sulphate present. The emission fluxes of $SO_2$ and $NO_X$ in HTAP-v2 data set also has large uncertainties over
the IGP (Jena et al., 2015b; Wang et al., 2020), which can introduce additional uncertainty in
$NH_3/NH_4^+$ gas/aerosol partitioning. In MOZART-4 chemistry, uncertainty can be also associated in dry and wet
deposition scheme which can result in overestimation (Emmons et al., 2010).

## 3.2 Seasonal variability of NH$_3$ total columns

Figure 6 shows the model (left) and IASI satellite (middle) seasonal distributions of NH$_3$ total columns over Asia. These seasons are represented as 3-month periods: Winter, December-January-February (DJF, first row), Spring, March-April-May (MAM, second row), Summer, June-July-August (JJA, third row), and Autumn, September-October-November (SON, fourth row). It can be seen in Fig. 6, that there is larger seasonal variation in IASI NH$_3$ total columns while MOZART-4 presents limited seasonality as in South Asia compare to better seasonal variation estimated in East Asia, as shown by both IASI and the MOZART-4 model. In general, during autumn, spring, summer and winter seasons MOZART-4 shows higher NH$_3$ total column compared with IASI estimates over most of South Asia. However, this difference is more pronounced during autumn (SON) and winter (DJF) seasons (Fig. 6; Right). We have seen that (Fig. 2) anthropogenic emission of NH$_3$ is nearly same in all months and biomass burning has peak during MAM over South Asia in the MOZART-4 model. Whereas, seasonality is better represented in NH$_3$ emission for East Asia.

Major drivers in anthropogenic NH$_3$ seasonal variation include differences in management and timing of fertilizer, which is not well represented in the emission over South Asia (Janssens-Maenhout et al., 2012). This can be expected to have the direct effect on NH$_3$ total column over South Asia. It is recognized that NH$_3$ emission can be strongly affected by both short term meteorological variation and longer term climatic differences (Sutton et al., 2013). This means that NH$_3$ emissions may be expected to increase in warm summer conditions than in winter (Battye and Barrows, 2004). However, magnitude of these emissions is expected to be smaller in comparison with anthropogenic emissions and may not contribute significantly to larger summertime NH$_3$ columns observed from IASI retrievals over South Asia and East Asia than MOZART-4. Additional driver in NH$_3$ seasonal variation include meteorological variation. For example, strong subsidence, lower temperature and lighter winds over South Asia in the autumn and winter months prevent venting of low altitude pollution to the higher altitudes. This means that emitted air pollutants tend to accumulate close to the source region in winter time conditions (Ghude et al., 2010, 2011). Considering the comparison of IGP with NCP, accumulation of pollutants in the boundary layer is more pronounced over IGP region due to flat land topography, and it is more during winter than the autumn months (Surendran et al., 2016). We saw that simulated mean Planetary boundary layer height (PBLH) is lower (approximately 400 m, Fig. S2 in the Supplement), and winds are lighter in winter months, compared to summer months, over South Asia, and particularly over IGP region (Surendran et al., 2016). Figure 7 (left) and 7 (right) shows the time-height distribution of NH$_3$ and mean PBLH averaged over the IGP region, respectively. It can be seen that during winter months higher atmospheric stability prevents mixing of boundary layer NH$_3$ to the free troposphere over IGP (Fig. 7 (left)), which is reflected in the higher wintertime values of MOZART-4 NH$_3$ columns. Similarly, higher NH$_3$/NH$_4$ ratio (Fig.S3 in the Supplement) and lower dry and wet deposition (Fig. S4 and S5 in the Supplement) of NH$_3$ over IGP in winter month enhances the accumulation of NH$_3$ in the boundary layer compared to summer months. On the other hand, much less NH$_3$ gets detected by the satellite at the higher altitudes where detection sensitivity of the satellite is more than that at the surface (Clarisse et al., 2010). Limited sensitivity of IASI measurements to detect boundary layer NH$_3$ (Van Damme et al., 2014a) could be one of the reasons for large differences (1-4×10$^{16}$ molecules cm$^{-2}$) between MOZART-4 and IASI in winter seasons. Also, sowing of wheat crop over IGP involves higher rate of fertilizer application during peak winter month (Sharma et al., 2014) that release significant quantity of NH$_3$ into the

atmosphere. However, this seasonality is largely missing in the emissions (Fig. 2 (top, left)) indicating that
higher MOZART-4 $NH_3$ is largely driven by the winter-time meteorology over this region.
It is interesting to note from Fig. 6 (right) that during spring the difference between modelled and observed
column $NH_3$ is smaller over the IGP region compared with the winter season. Heating of the landmass due to
large solar incidence suppresses the wintertime subsidence over the IGP and leads to deeper boundary layer
during spring and early summer. It can be seen that (Fig. 7 (right) and Fig. S2 in the Supplement) the average
PBLH is about 1100 m and 600 m deeper during spring and summer compared to winter over IGP. During this
season, significant transport of the boundary pollution in the mid and upper troposphere due to enhanced
convective activities and large scale vertical motion can be noticed in Fig. 7 (left) and is consistent with the
earlier studies over this region (Lal et al., 2014; Surendran et al., 2016). Vertical motion associated with the
convective activities is expected to redistribute the $NH_3$ concentration in the column, which leads to more $NH_3$
at the higher altitudes where detection sensitivity of the satellite is more than that at the surface (Clarisse et al.,
2010). As a result, more $NH_3$ gets detected by the satellite and we see less difference between observations and
model over the IGP. This may also partly explain the higher IASI estimates of $NH_3$ column for summertime
prior to the monsoon season. However, this hypothesis needs to be tested with higher sensitivity experiments as
a part of future work. During spring season, MOZART-4 reflects widespread $NH_3$ total column from the entire
Indian land mass and IASI observations does capture increase in $NH_3$ total column at least for seasonal mean
cycle (Fig. 8a). This seasonal maximum in $NH_3$ total column identified both in IASI and MOZART-4 over
South Asia can be explained by the two factors: Meteorology factor and biomass burning emissions.
Volatilization of $NH_3$ enhances with increase in temperature (Sutton et al., 2013), hence higher temperature
during this drier periods over IGP partly enhances $NH_3$ emission to the environment which is also evident from
the soil $NH_3$ emissions in Fig. 2 (bottom). However, magnitude of these emissions is expected to be smaller in
comparison with anthropogenic emissions. In the Indian region, emissions from the biomass burning (crop-
residue burning) peaks in March to May (Jena et al., 2015a) and emission of $NH_3$ from biomass burning is
maximum during this period (Fig. 2 (middle)). However, MOZART-4 estimates smaller $NH_3$ total columns
compared with IASI over Myanmar, Laos and Thailand during the period March-May (Fig. 6 (right)). This
period is estimated to be associated with large scale forest fires (and open crop burning) (Chan, 2017; Wu et al.,
2018; Zheng et al., 2017), the effect of which appears to be underestimated in the MOZART-4 simulations. It
suggests that the Global Fire Emissions Database (GFED-v3) used in this study is low over this region agreeing
with Zhang et al. (2020) and Huang et al. (2013). During the monsoon season (JJA) (Fig. 6 (right)) and summer,
IASI-$NH_3$ total columns are larger than the MOZART-4 estimates over north-western arid region of South Asia,
where monsoon rainfall is lowest (less than 30 cm). On the other hand, $NH_3$ columns estimated by IASI are
lower in the North-western IGP than the MOZART-4 simulations.
Figure 8 shows the comparison between IASI and modelled monthly time series of $NH_3$ total columns over IGP
(20°N-32°N, 70°E-95°E) and NCP (30°N-40°N, 110°E-120°E), respectively (rectangular areas shown on Fig.
1). We found a better consistency between modelled and measured seasonal $NH_3$ total column over NCP than
IGP. Monthly $NH_3$ columns over the IGP show bimodal distribution in the model. However, IASI does not
show such bimodal variation. Seasonal statistics show large normalised mean bias (38 %) and poor correlation (r
= 0.41) between model and IASI. The bimodal distribution in $NH_3$ total columns is partly driven by the biomass
burring emissions, which show major peak in spring and another small peak in autumn (Fig. 2 (middle)), and

partly by the meteorology as discussed in the previous section. During monsoon months (JJA), when South Asia receives significant rainfall all over, model simulations present lower $NH_3$ total column, which is not seen in the IASI observations and also in the surface observations (Fig. 8a and 9b) over IGP. The reason for this discrepancy may be related with the flat $NH_3$ emission over South Asia (Fig. 2). Usually large amount of fertilization application is expected during the warm month of June and July in the IGP which is not represented in the HTAP-v2 emissions and therefore lower values in the model during monsoon month is mostly driven by the model meteorology. Lower values observed during monsoon season in general are attributed to increase wet scavenging of $NH_3$ due to monsoon rain (Fig. S5 (left) in the Supplement) and influx of cleaner marine air from the Bay of Bengal and Arabian Sea through south-easterly and south-westerly wind (Ghude et al., 2008). On the other hand, monthly variation in IASI $NH_3$ total columns over East Asia is found to be captured well by the model (Fig. 8b) and seems to follow the variation observed in the anthropogenic $NH_3$ emission (Fig. 2), except for the month of July where IASI estimates substantially higher $NH_3$ total columns than the model. The reason for this peak in the IASI data for July may be related to urea fertilizer application in warm July conditions (see temporal course of Enhanced Vegetation Index (Li et al., 2014)), which seems to be not represented well in the HTAP-v2 emissions. The overall statistics show slight good correlation (r = 0.61) between observed and simulated $NH_3$ columns and negative normalised mean bias (NMB = -41 %).

### 3.3 Comparison between surface $NH_3$ measurements and simulated $NH_3$ concentrations in South and East Asia

To evaluate modelled surface $NH_3$ concentrations in South Asia, we have used $NH_3$ surface measurements from 69 monitoring locations over India for the years from 2016 to 2019. As 2010 data was not available, we make the hypothesis that measurement from 2016-2019 can be considered as representative from what have been measured in 2010. Out of these stations thirty five locations in Delhi, six in Bangalore city, four in Hyderabad, and two in Jaipur city are averaged to get singe value for the same geographical location and the remaining 22 locations are considered independently representing 26 respective cities. Due to the lack of ground-based measurements performed in 2010, the following comparison will mainly be qualitative, although it is estimated that the main spatial features of Indian agriculture and $NH_3$ emissions will be consistent between 2010 and 2016-2019. As per the RCP 8.5 (Kumar et al., 2018) $NH_3$ emission from South Asia is expected to increase by less than 20 % from 2010 to 2020. Assuming a linear relationship between emission and surface concentration, it is expected that $NH_3$ concentrations could be higher by about 10-15 % in 2016 to 2019.

It is interesting to note that the correlation between annual and monthly mean MOZART-4 simulated and measured $NH_3$ concentration (r = 0.82 and r = 0.62) is better than the comparison between MOZART-4 and IASI for South Asia (Fig. 9). However, the MOZART-4 has systematically smaller estimated $NH_3$ concentrations compared with the ground based measurement network (NMB = -47 %). It should be noted that most of the monitoring stations are situated in urban regions(cities) of India and therefore represents the urban environment, which may have locally higher $NH_3$ concentrations due to traffic and human activities (Sharma et al., 2014). Since the MOZART-4 model is run relatively at coarse (1.9° × 2.5°) grid resolution the emissions may not capture the true variability in emissions at city scale. These surface $NH_3$ sites are influenced by local emissions that are therefore not resolved by the MOZART-4 model. Therefore, when comparing coarse-scale models to observations, the model may have difficulties in resolving local scales effects (Surendran et al., 2015). Until the planned further evaluation of the chemiluminescence monitoring method for ammonia (measured by

difference with NOx) is evaluated (as noted in Sect. 2.4), it is not possible to be certain the extent to which
possible uncertainties in the measurement method contribute to the differences shown in Fig. 9b. While noting
these uncertainties, it is worth noting that the ground based $NH_3$ observation network confirms the occurrence of
higher ground-level $NH_3$ concentrations in autumn and winter, as simulated using MOZART-4 using the HTAP-
v2 emissions inventory (Fig. 9b).
Comparison of Fig. 8a and 9b shows that the time course of ground level $NH_3$ concentrations (as estimated by
MOZART-4) is significantly different to the time course of total $NH_3$ column (as also estimated by MOZART-
4). Whereas the total column is largest in the summer (reflective of deeper atmospheric mixing and
recirculation), and the ground level concentrations are largest during winter. Although it is not easy to use the
IASI data to infer ground level $NH_3$ concentrations, the stronger summer maximum of IASI (Fig. 8a) compared
with MOZART-4, suggests that IASI would be in less close agreement with the ground based measurement
network than MOZART-4 (Fig. 9b). While recognizing uncertainties in this interpretation, the key point is that
large $NH_3$ columns estimated by IASI for May-July are not reflected in the ground-based $NH_3$ measurements
from the Indian monitoring network.
Figure 10 shows the comparison between monthly mean (from 2010 to 2015 observations) $NH_3$ surface
measurements from 32 monitoring locations over China and modelled surface $NH_3$ concentrations from the
same location over East. Similar to South Asia the MOZART-4 has systematically smaller estimated $NH_3$
concentrations compared with the ground based measurement network (NMB = -44 %) over East Asia. Figure
10b shows maximum $NH_3$ concentration occurred in summer (JJA) denotes agreement with IASI measurements.
Other statistical indicators are summarised in Table 2. Furthermore, high $NH_3$ concentration from ground based
measurements during JJA is consistent with the higher HTAP-v2 emissions (Fig. 2) (Huang et al., 2012) and
higher $NH_4NO_3$ concentration (Fig. S6 in the Supplement). Higher concentration of $NH_4NO_3$ and can also lead
to higher $NH_3$ concentrations especially during summer due to its semi-volatile and unstable character at higher
temperatures, as it is observed in East Asia. This implies that the $NH_3$ emissions may play a vital role in
determining the seasonal pattern of the ground $NH_3$ concentrations. Summer peak may originate from fertilizer
application, livestock emissions and volatilization of $NH_3$ which is enhanced in higher temperature (Liu et al.,
2017a).
**3.4 Why were $NH_3$ total columns low over high $NH_3$ emission over East Asia compared to high $NH_3$**
**emission region of South Asia?**
Fine-scale details of the $NH_3$ emissions over Asia in Fig. 1 and 2 clearly revealed larger emission values in areas
where there is intensive agricultural management. This is the case especially in the NCP and IGP (Fig. 1, shown
with box). Earlier emission estimates suggest that fertilizer application and livestock contribute 2.6 Tg per year
($yr^{-1}$) and 1.7 Tg $yr^{-1}$ $NH_3$ emissions respectively from South Asia (Aneja et al., 2011). Over South Asia, urea
accounts for emissions of 2.5 Tg $yr^{-1}$ which contributes to 95 % of the fertilizer emission, and 58 % of total
estimated agricultural emissions (Fertlizer Association of India annual report 2018-19). For East Asia, livestock
manure management accounts for approximately 54 % (5.3 Tg $yr^{-1}$) of the total emissions and fertilizer
application accounts for 33 % (3.2 Tg $yr^{-1}$) emissions, with 13 % of emissions from other sources. Combined the
model areas for NCP and IGP (as shown in Fig. 1) accounts for ~45 % of the $NH_3$ emitted from fertilization in
East Asia and South Asia ( Huang et al., 2012).
We find that satellite observations show larger $NH_3$ columns over IGP than over similar higher emission regions
of NCP. However, in addition, we also find that the MOZART-4 model is able to capture this contrasting
columnar $NH_3$ levels between IGP and NCP. This indicates that the difference between IGP and NCP is
unrelated to differences between the mosaic of emissions over South Asia and East Asia in HTAP-v2 and
similarly not related to uncertainties in satellite retrievals. Instead, the analysis from MOZART-4 demonstrates
that the difference can be explained by differences in atmospheric chemistry between the two regions, linked to
higher $SO_2$ and $NO_x$ emissions in the NCP than in the IGP. Recent study by Wang et al. (2020), shows that
emission fluxes of $SO_2$ and $NO_x$ over IGP are only one-fourth of that over NCP.
As ammonia is a highly alkaline gas with an atmospheric lifetime usually of few hours (and rarely a few days)
(Dammers et al., 2019), it readily reacts with acid present in the atmosphere to form aerosols, which are
eventually deposited to the earth's surface by either dry or wet deposition processes (Fig. S4 and S5 in the
Supplement). In the atmosphere, ammonia therefore reacts rapidly with atmospheric sulphuric acid ($H_2SO_4$),
nitric acids ($HNO_3$) and hydrochloric acid ($HCl$) to contribute to ambient levels of fine particles, forming
ammonium sulphate, ammonium nitrate and ammonium chloride. Following reaction (R1) and (R2)
$$NH_{3(g)} + HNO_{3(g)} \leftrightarrow NH_4NO_{3(s)} \tag{R1}$$
$$2NH_{3(g)} + H_2SO_{4(g)} \leftrightarrow (NH_4)_2SO_{4(s)} \tag{R2}$$

In the atmosphere, ammonium ion ($NH_4^+$) as an aerosol is estimated to have a lifetime of about 1–15 days
(Aneja et al., 1998), though this is obviously dependent on the amount of atmospheric acids (Seinfeld and
Pandis, 2012). In addition to the large fertilizer application and livestock management activities which are
characteristic of both IGP and NCP, industrial and transportation activities are higher over the NCP (China)
which also results in higher emission of $NO_x$ and $SO_2$ over NCP compared with IGP (Zhao et al., 2013).
Ammonia has greater affinity towards oxides of sulphur, hence it first reacts to form ammonium sulphate, and
then the remaining ammonia further reacts to form ammonium nitrate (Seinfeld et al., 1998). The differences in
the secondary aerosol formation over NCP and IGP are compared by considering the MOZART-4 model
estimates of volume mixing ratio (VMR) in parts per billion ($\times 10^9$ ppb) of total sulphate, ammonium,
ammonium nitrate at surface and total column of $NO_x$ (Fig. 11). Although vertical profiles of the aerosol
components are small, there are strong vertical gradients in $NO_x$ concentrations, and for this reason we consider
the comparison with the total $NO_x$ column more reflective of overall $NO_x$ chemistry than the ground level $NO_x$
VMR.
Figure 11 shows that total sulphate VMR (Fig. 11a) and $NO_x$ total column (Fig. 11c) are significantly higher
over NCP region than IGP. Similarly, total ammonium VMR (Fig. 11b) is significantly larger over NCP than
IGP indicating how a higher fraction of the gaseous ammonia is transformed to form ammonium over NCP
region. In addition, Fig. 11d shows higher estimated levels of ammonium nitrate in MOZART-4 over NCP,
reflective of the higher NOx emissions in this region. As a consequence of the different $SO_2$ and $NO_x$ sources,
gaseous $NH_3$ is more quickly removed from atmosphere over East Asia with residence time of approximately 6
hours (Fig. S7 in the Supplement) (higher values indicates lower mean residence time), which is reflected in the
higher VMR of ammonium, sulphate and ammonium nitrate (Fig. 11a, b and d). It can be seen that $NH_3/NH_4^+$
ratio denotes lower values 0-1 (Fig. S3 in the Supplement) over East Asia than South Asia suggesting $NH_4^+$
partitioning is more over East Asia. As a result the $NH_3$ total columns over NCP are much smaller than over
IGP, even though magnitude of $NH_3$ emission fluxes is greater over NCP than IGP. This difference indicates
that the high $NH_3$ loading over the IGP is partly coming from the low gas-to-particle partitioning of $NH_3$ caused
by low $SO_2$ and $NO_x$ emission over South Asia. In contrasts high $SO_2$ and $NO_x$ emissions promote the
conversion of gaseous $NH_3$ into particulate ammonium in NCP. However, rapid decline of acidic ($SO_2$)
emissions over China after 2000, which may not be reflected correctly in HTAP_v2 (Mortier et al., 2020; Tong
et al., 2020; Zheng et al., 2018) will lead to higher $NH_3$ loading due to less partitioning of $NH_3$.

## 4. Conclusion

In this work, we have compared $NH_3$ total columns simulated by the MOZART-4 model with IASI $NH_3$ satellite
observations over South and East Asia. The annual mean distribution reveals a consistent spatial pattern
between MOZART-4 and IASI, but MOZART-4 tends to show larger $NH_3$ columns over South Asia than IASI,
particularly over the Indo-Gangetic Plain (IGP), whereas it is in close agreement over East Asia (including the
North China Plain, NCP), with the exception of a July peak seen in the IASI dataset, which may be related to
specific timing of fertilizer-related $NH_3$ emissions. Comparison for seasonally and monthly resolved IASI total
column with the MOZART-4 simulations shows inconsistencies in spatial and temporal pattern over South Asia.
This inconsistency is due to the uncertainties in emission estimate which doesn't include seasonality pattern in
HTAP-v2 over South Asia, as well as uncertainties in the processing of the IASI data. Both the MOZART-4
results and IASI estimates involve assumptions that could considerably affect the comparison between total
columns of $NH_3$.
Comparison with estimates from a ground based $NH_3$ monitoring network for both South and East Asia, our
results showed that MOZART-4 systematically gives smaller $NH_3$ concentration estimates than the monitoring
network. The $NH_3$ measurement sites used in present study mostly represent urban locations and model may not
be able to capture actual concentration at point location due to coarser grid resolution over India. In addition,
further assessment is needed to demonstrate the reliability of the $NH_3$ measurement technique used in the
monitoring network, where $NH_3$ is measured by difference with $NO_x$ concentrations, which may be uncertain in
urban areas with high $NO_x$ concentrations.
Despite the high $NH_3$ emission over both South and East Asia, a larger $NH_3$ total column is observed over South
Asia in both the IASI and MOZART-4 estimates. This difference is explained by the MOZART-4 simulation,
which treat the full atmospheric chemistry interaction with $SO_2$ and $NO_x$ emissions, leading to aerosol
formation. The MOZART-4 model showed higher sulphate volume mixing ratio and $NO_x$ total column over East
Asia, especially in the NCP, which is reflected in ammonium aerosol volume mixing ratio (VMR) over East
Asia. This suggests that the formation of ammonium aerosols (dominated by ammonium, sulphate and
ammonium nitrate) is quicker over East Asia than in South Asia, leading to lower $NH_3$ total columns in East
Asia.
To examine the present findings future studies should investigate the effect of changing emissions of $NO_x$ and
$SO_2$ on $NH_3$ columns, for example by using perturbation of these emissions through counterfactual modeling
scenarios. The comparison between model simulations using MOZART-4, satellite derived estimates from IASI
and ground-based monitoring of $NH_3$ concentrations has highlighted the known uncertainties in emissions,
satellite retrievals and measurements at point locations. In order to reduce the uncertainties in ammonia
emission, it would be a key to create an NH₃ emission inventory specifically over South Asia, which is now
currently under development as part of the GCRF South Asian Nitrogen Hub. This includes work to improve the
bottom-up NH$_3$ emission inventory, taking into account primary agricultural statistics on fertilizer use and
animal number distributions. There is also potential for top-down (inverse modelling) for NH$_3$ and NO$_x$ by
taking inference from the model, satellite and ground-based evidence. Here it is essential to recognize the need
for more ground-based observational sites to measure NH$_3$ air concentrations in rural areas where agriculture
activity is predominant. Such measurements at present are currently very few for South Asia. Coarser global
models fail to resolve the local-scale emissions, hence higher resolution regional models with advance chemistry
are also needed to resolve the sources and chemical processes on urban and rural scales.

**Data availability**

The $0.1^{\circ} \times 0.1^{\circ}$ emission grid maps can be downloaded from the EDGAR website on
https://edgar.jrc.ec.europa.eu/htap_v2/index.php?SECURE=_123 per year per sector. The model data can be
downloaded upon request from the AeroCom database (http://www.htap.org/, last accessed June 22, 2020) (TF
HTAP, 2018). The model data is available at Prithvi (IITM) super-computer and can be provided upon request
to corresponding author. The morning overpass NH$_3$ total columns measured through IASI can be accessed from
data center at http://cds-espri.ipsl.upmc.fr/etherTypo/index.php?id=1700&L=1. For India, ground based hourly
NH$_3$ measurements can be obtained from CPCB website on https://app.cpcbccr.com/ccr. For China, ground
based monthly mean NH$_3$ datasets can be downloaded from
https://figshare.com/articles/Data_Descriptor_Xu_et_al_20181211_Scientific_data_docx/7451357.

**Author contributions**

All authors contributed to the research; SDG designed the research; PVP conducted the research; PVP and SDG
wrote the paper; CJ and DS performed the MOZART model simulations; AM and MAS formulated the
research; MVD, LC and PFC performed the IASI experiments; SK, DML, GG, XL, WU, JJ and TKA
contributed to writing.

**Competing interests**

The authors declare that they have no conflict of interest.

**Acknowledgments**

We wish to thank the National Centre for Atmospheric Research (NCAR), funded by the U.S. National Science
Foundation and operated by the University Corporation for Atmospheric Research, for access to the MOZART-
4. All model runs were carried out on a Prithvi IBM High Performance Computing system at the Indian Institute
of Tropical Meteorology (IITM), Pune India. We thank the Director, IITM for providing all the essential
facilities required to complete the work. We wish to acknowledge the availability of CPCB data from CPCB
webportal (https://app.cpcbccr.com/ccr).Research at ULB has been supported by the Belgian State Federal

Office for Scientific, Technical and Cultural Affairs (Prodex arrangement IASI.FLOW). L.C. and M.V.D are respectively research associate and postdoctoral researcher with the Belgian F.R.S-FNRS. Cooperation between IITM and CEH has been facilitated through the NEWS India-UK Virtual Joint Centre, supported at CEH by the Biotechnological and Biological Sciences Research Council, and the Natural Environment Research Council of UK Research and Innovation (UKRI), and through the UKRI Global Challenges Research Fund (GCRF) South Asian Nitrogen Hub. The Nationwide Nitrogen Deposition Monitoring Network (NNDMN) of China was supported by the Chinese National Natural Science Foundation (41425007) and the Chinese National Research Program for Key Issues in Air Pollution Control (DQGG0208).We thank anonymous reviewers and the editor for their constructive comments that helped in improving quality of this manuscript.

**Financial support**

This research has been supported by "Urban modeling C-DAC" sponsored project.

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

**FIGURE CAPTIONS**

Figure 1. Spatial distribution of total $NH_3$ emissions ($\times 10^{-10}$ kg m$^{-2}$s$^{-1}$) over Asia. Data are shown at 0.1° × 0.1° grid resolution from Hemispheric Transport of Air Pollution version-2 (HTAP-v2) emission inventory. The solid rectangles indicate the Indo-Gangetic plain, IGP ( 20°N-32°N, 70°E-95°E) and the North China Plain, NCP (30°N-40°N, 110°E-120°E).

Figure 2. Monthly variation of anthropogenic (HTAP-v2) (molecules cm$^{-2}$ s$^{-1}$) (top), Biomass Burning (GEFED-v3) (molecules cm$^{-2}$ s$^{-1}$) (middle) and Soil (CESM) (molecules cm$^{-2}$ s$^{-1}$) (bottom) $NH_3$ emission averaged from Indo-Gangetic plain (20°N-32°N, 70°E-95°E) and the North China Plain (30°N-40°N, 110°E-120°E).

Figure 3. Geographical locations of surface $NH_3$ observational sites (69 locations) from the air quality automatic monitoring network operated by the Central Pollution Control Board (CPCB, 2020), India and observational sites (32 locations) from Nationwide Nitrogen Deposition Monitoring Network (NNDMN) operated by China Agricultural University, China.

Figure 4. Spatial distributions of annual mean $NH_3$ ($\times 10^{16}$ molecules cm$^{-2}$) total columns over Asia for the year 2010. (a) Simulated by MOZART-4, (b) from the IASI satellite observations and (c) spatial difference between MOZART-4 and IASI.

Figure 5. (a) Scatter plot between annual averaged IASI and MOZART-4 simulated $NH_3$ ($\times 10^{16}$ molecules cm$^{-2}$) total columns over IGP, South Asia (rectangle: 20°N-32°N, 70°E-95°E) and (b) Scatter plot between annual averaged IASI and MOZART-4 simulated $NH_3$ ($\times 10^{16}$ molecules cm$^{-2}$) total columns over NCP, East Asia (rectangle: 30°N-40°N, 110°E-120°E).

Figure 6. Seasonal $NH_3$ total columns distribution ($\times 10^{16}$ molecules cm$^{-2}$) in 2010 (left) simulated by MOZART-4, (middle) measured by IASI satellite and (right) spatial differences between MOZART-4 and IASI during (top to bottom) winter (DJF) spring (MAM) summer (JJA) and autumn (SON) seasons.

Figure 7. Daily vertical distribution of $NH_3$ (ppb) averaged over IGP South Asia (20°N-32°N, 70°E-95°E) (left) and daily mean Planetary Boundary Layer height (PBLH in meters) averaged over IGP South Asia (20°N-32°N, 70°E-95°E) (right).

Figure 8. (a) Comparison between monthly averaged IASI and MOZART-4 simulated $NH_3$ ($\times 10^{16}$ molecules cm$^{-2}$) total columns over IGP South Asia (20°N-32°N, 70°E-95°E), (b) Comparison of monthly averaged IASI and MOZART-4 simulated $NH_3$ ($\times 10^{16}$ molecules cm$^{-2}$) total columns over NCP East Asia (30°N-40°N, 110°E-120°E) (bar indicates standard error of 88 and 35 pixels in IGP and NCP respectively).

**Figure 9. (a)** Scatter plot between annual averaged surface observations from 69 monitoring sites (Fig. 2) over South Asia and MOZART-4 simulated surface $NH_3$ ($\mu g\ m^{-3}$) (992 hPa) interpolated at the locations of 69 sites **(b)** Comparison between monthly mean surface observations from 69 monitoring sites and MOZART-4 simulated monthly mean $NH_3$ ($\mu g\ m^{-3}$) concentration interpolated at the locations of 69 sites over South Asia.

**Figure 10. (a)** Scatter plot between annual averaged surface observations from 32 monitoring sites (Fig. 2) over East Asia and MOZART-4 simulated surface $NH_3$ ($\mu g\ m^{-3}$) (992 hPa) interpolated at the locations of 32 sites **(b)** Comparison between monthly mean surface observations from 32 monitoring sites and MOZART-4 simulated monthly mean $NH_3$ ($\mu g\ m^{-3}$) concentration interpolated at the locations of 32 sites over East Asia.

**Figure 11.** MOZART-4 simulated spatial distribution of annual averaged **(a)** total sulphate aerosol ($\times 10^9$ ppb), **(b)** total Ammonium aerosol ($\times 10^9$ ppb), **(c)** $NO_x$ total columns ($\times 10^{16}$ molecules $cm^{-2}$) and **(d)** total ammonium nitrate aerosol ($\times 10^9$ ppb) over Asia.

 **TABLES**

 **Table 1 Model performance statistics for NH$_3$ total columns over Asia from IASI and MOZART-4**
 **simulations for the year 2010**


| Statistics indicator | IGP, South Asia | NCP, East Asia |
|---|---|---|
| Mean (Model-IASI ) ($\times 10^{16}$ molecules cm$^{-2}$) | 0.68 | -0.24 |
| Normalized Mean Bias (NMB) | 0.38 | -0.35 |
| Variance ($\times 10^{16}$ molecules cm$^{-2}$) | 1.39 | -0.83 |
| Root Mean Square Error (RMSE) ($\times 10^{16}$ molecules cm$^{-2}$) | 0.125 | 0.05 |
| Correlation Coefficient (r) | 0.81 | 0.90 |



























**Table 2 Model performance statistics for NH$_3$ concentration over East and South Asia from MOZART-4**
**simulations and observational network for the year 2010**

| Statistics indicator | IGP, South Asia | NCP, East Asia |
|---|---|---|
| Mean (Model-Observations) ($\mu$g m$^{-3}$) | -13.47 | 3.1 |
| Normalized Mean Bias (NMB) | 0.44 | -0.46 |
| Variance ($\mu$g m$^{-3}$) | -0.629 | -0.88 |
| Root Mean Square Error (RMSE) ($\mu$g m$^{-3}$) | 1.91 | 0.728 |
| Correlation Coefficient (r) | 0.82 | 0.65 |



**Figure 1**

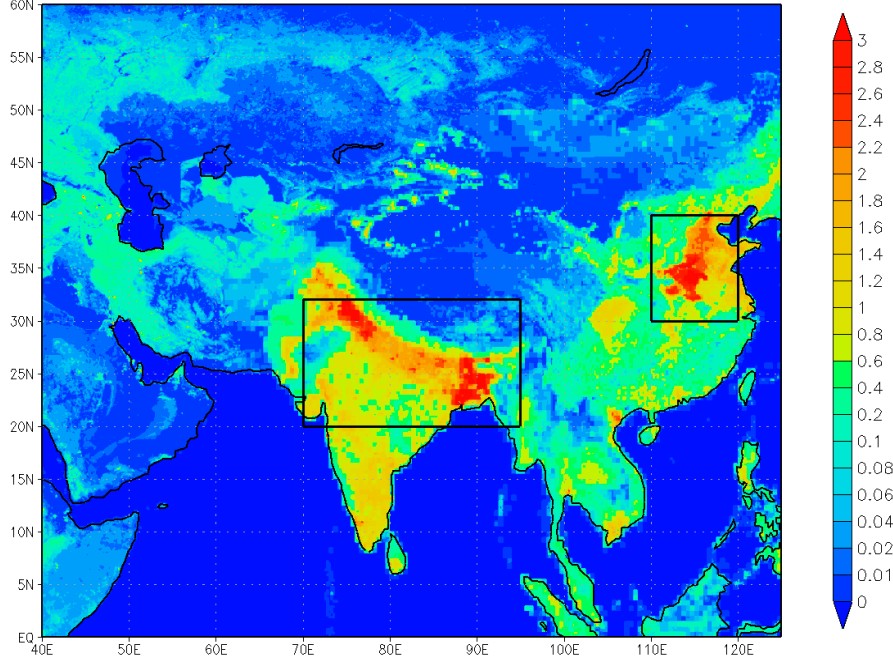





**Figure 2**

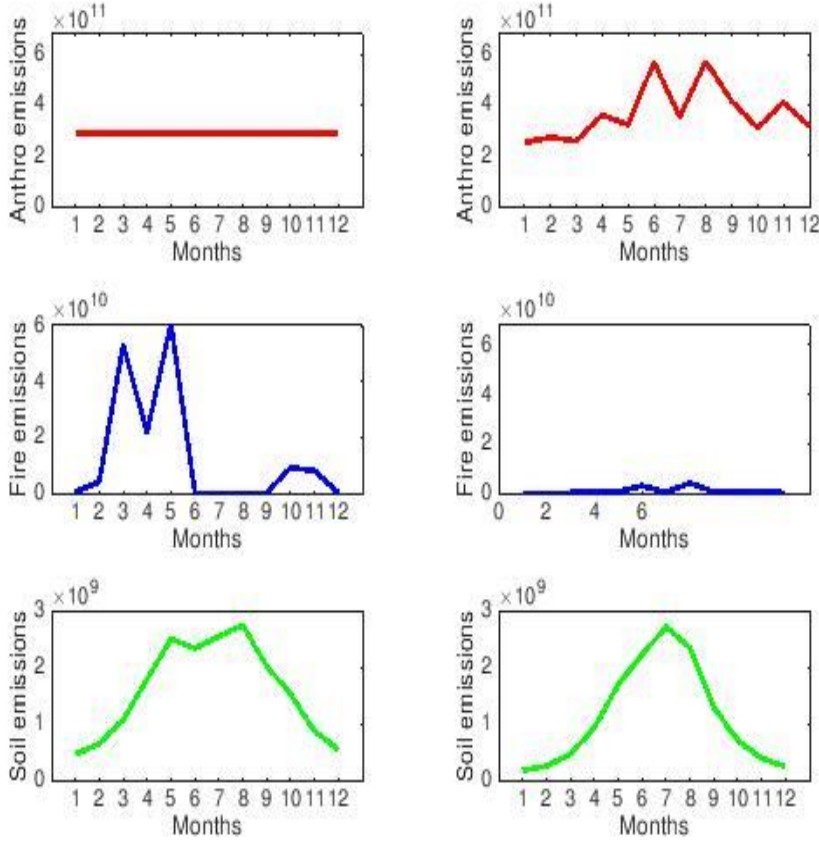


**Figure 3**

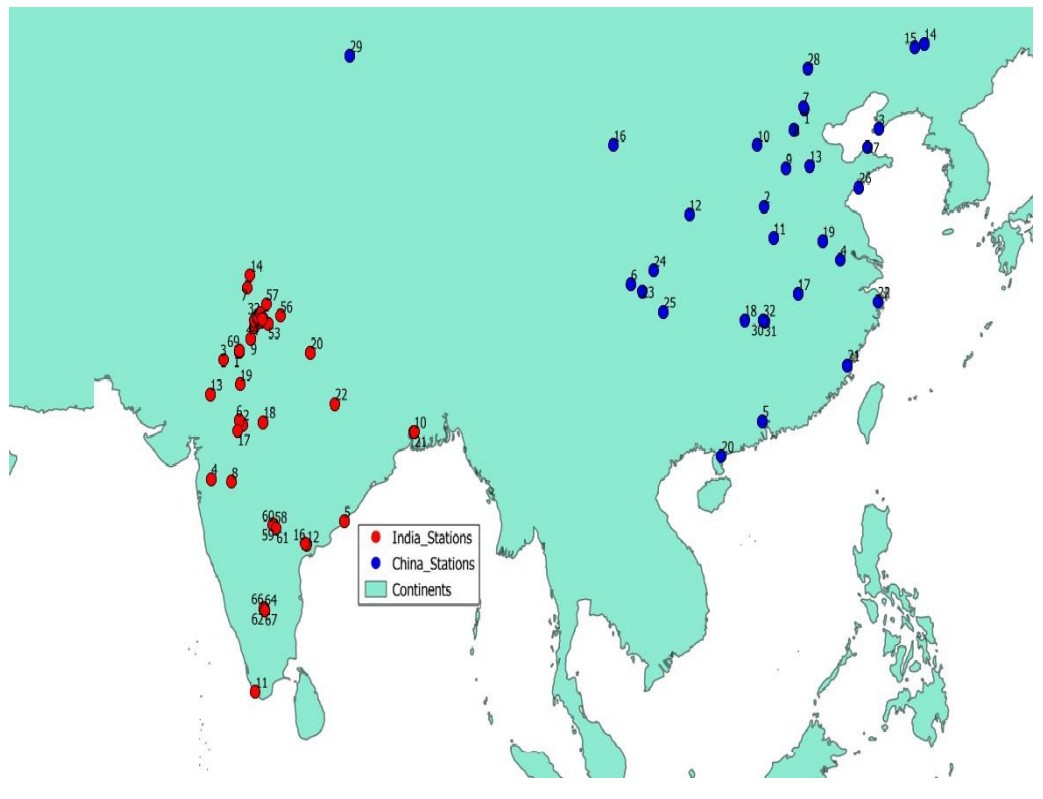


**Figure 4**

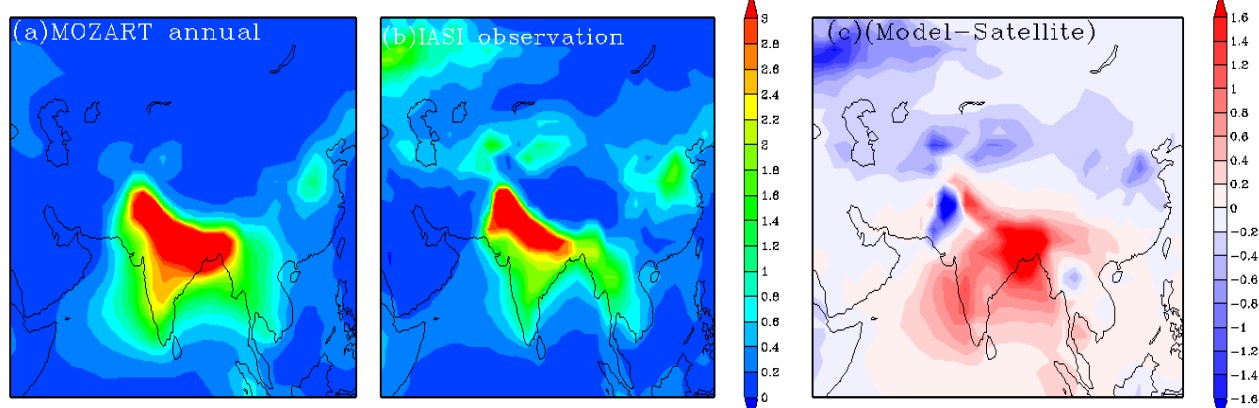






**Figure 5**

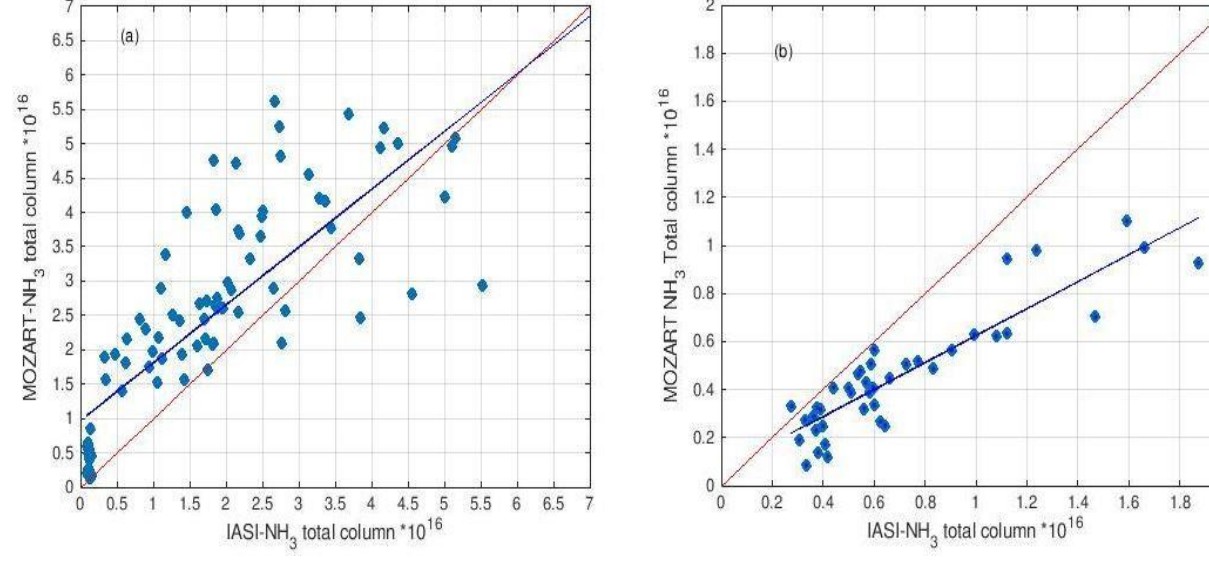



**Figure 6**

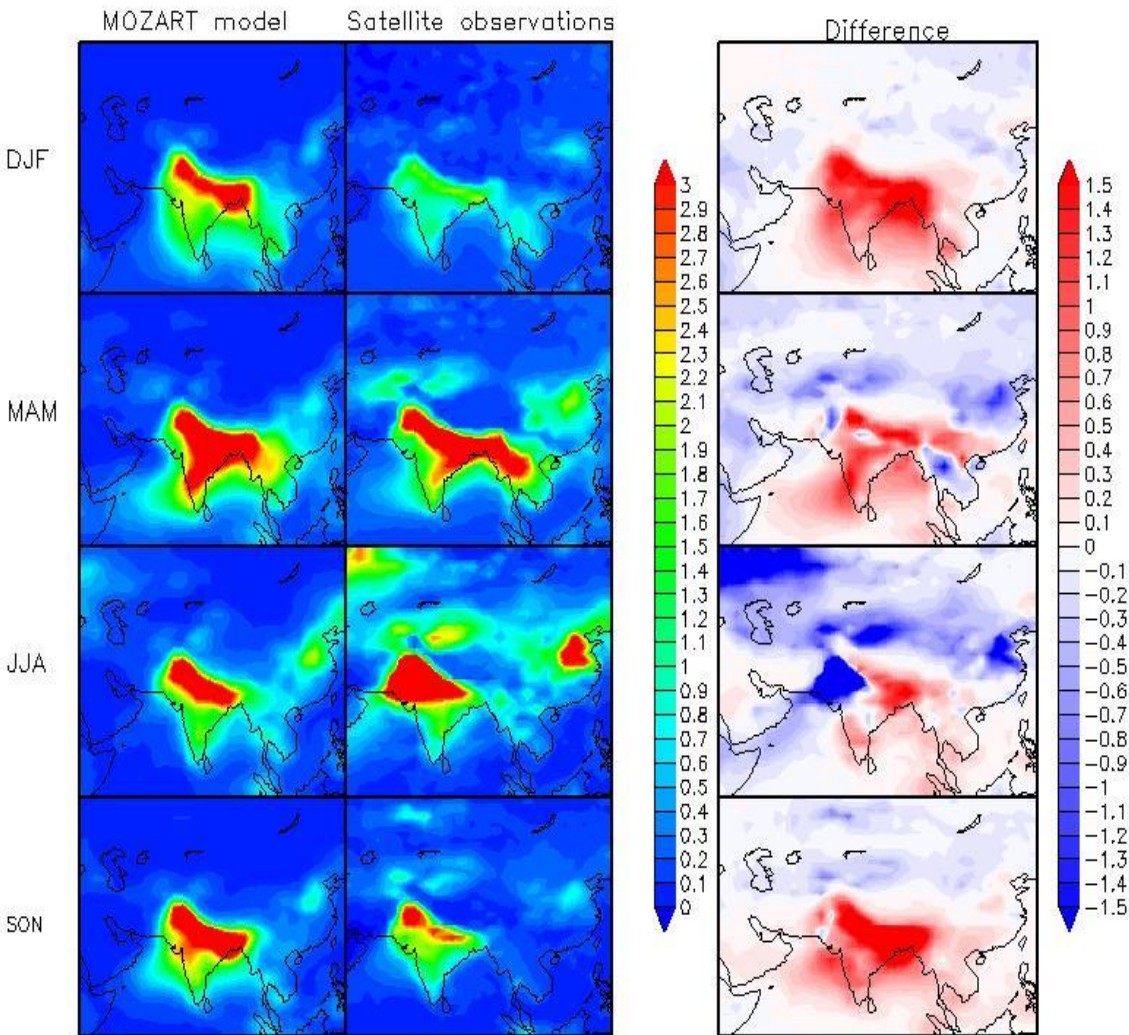

**Figure 7**

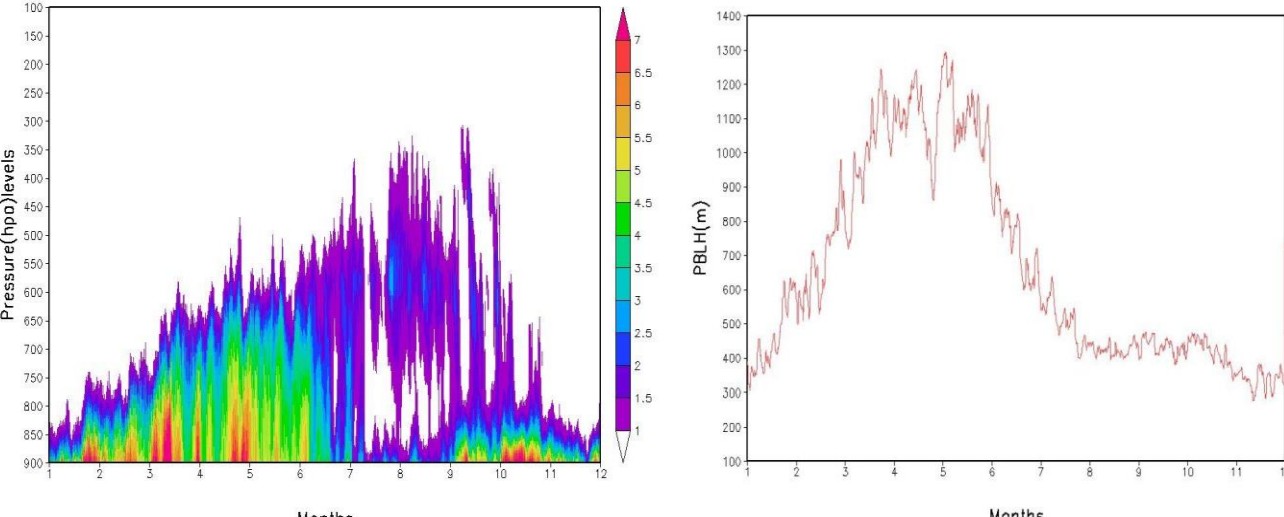

**Figure 8**

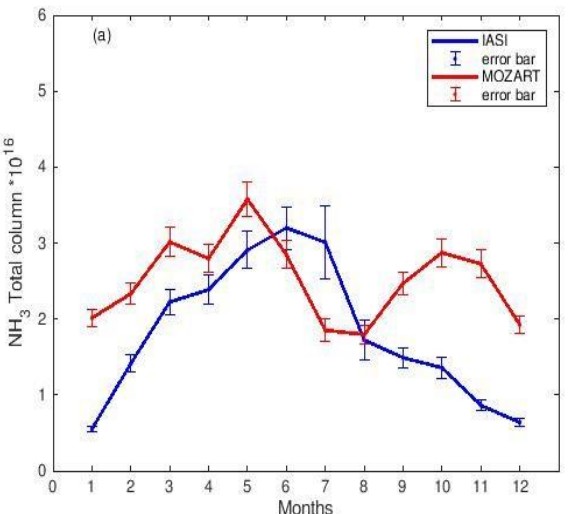 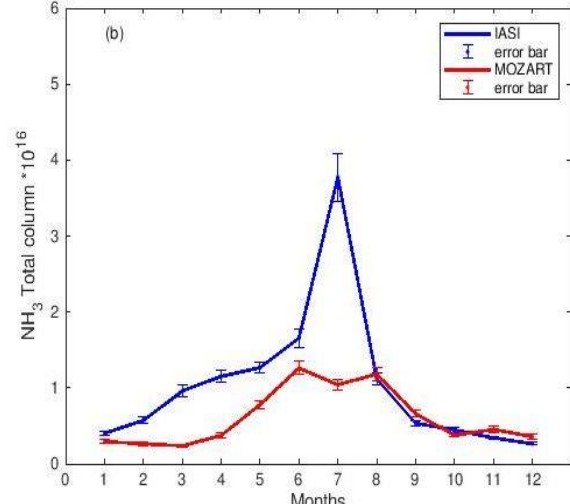

 **Figure 9**

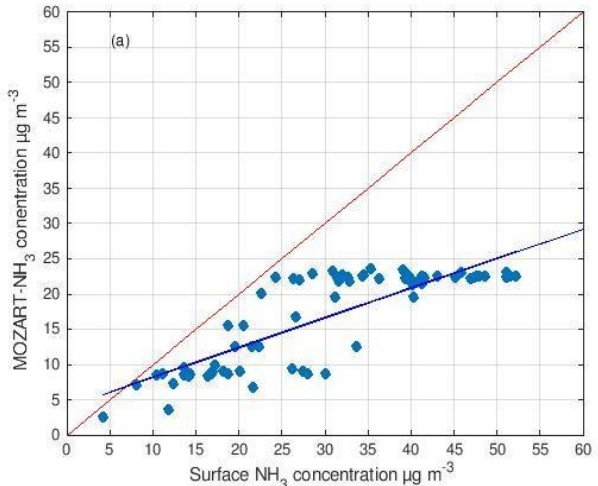
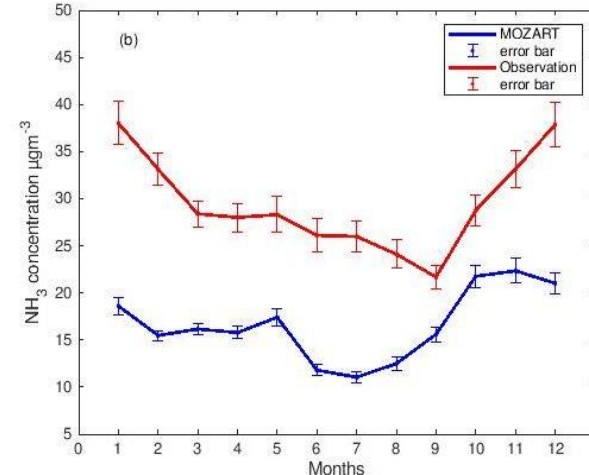

**Figure 10**

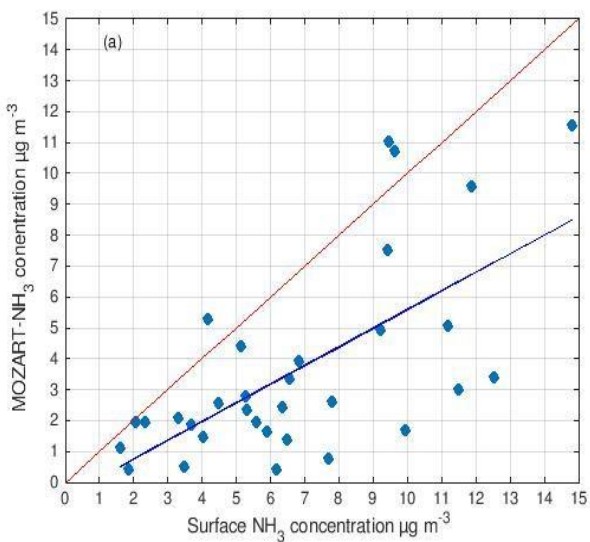 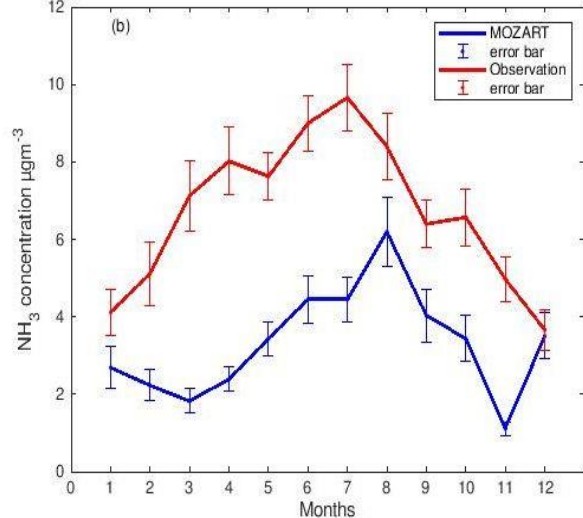

**Figure 11**

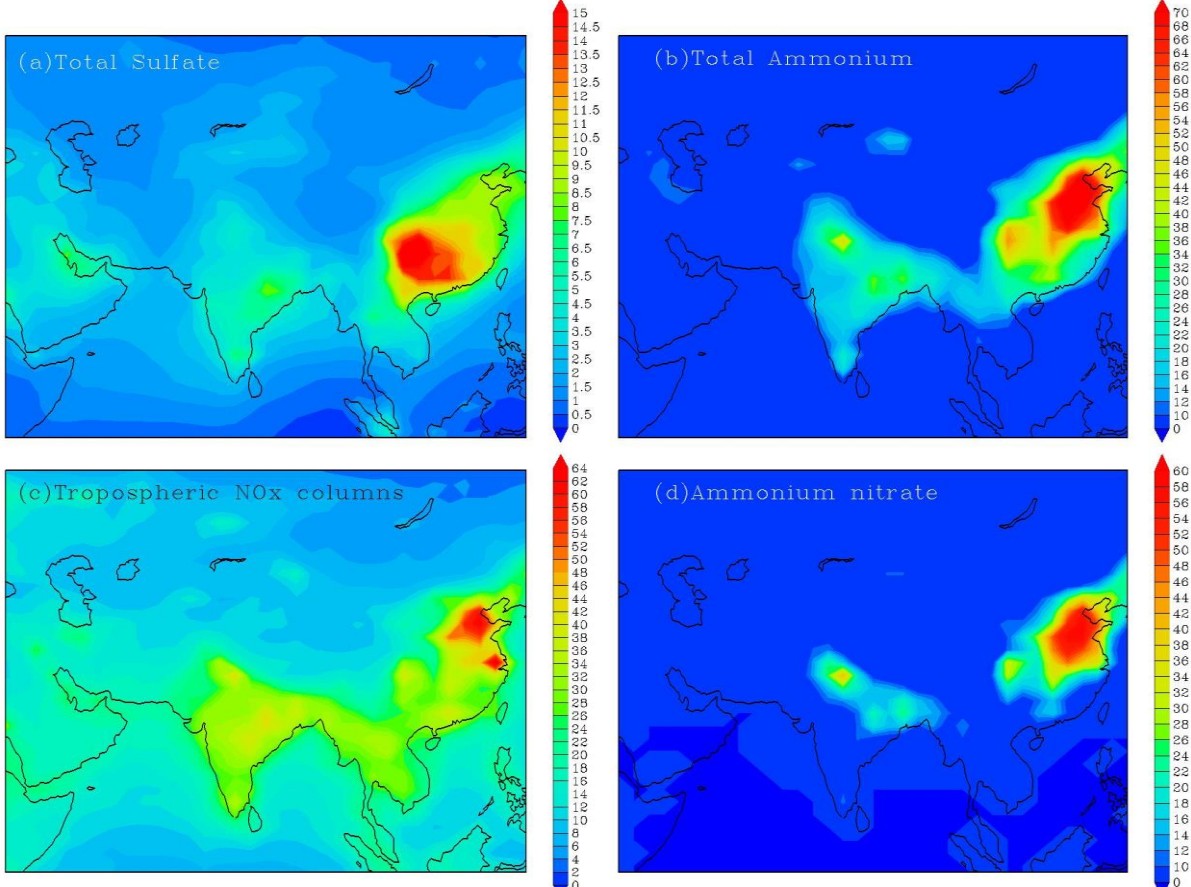
