# Peer review of "Analysis of atmospheric ammonia over South and East Asia"

_Atmospheric Chemistry and Physics, 2020_

## Short Comment (SC1) · 26 Sep 2020

Paper present the comparison of the MOZART-4 model along with monthly averaged satellite distributions of ammonia emission across South Asia. The authors are trying to identify the northern region of India i.e., Indo-Gangetic Plain, IGP as a hotspot for NH3 in Asia, both using the model and satellite observations. They highlighted a close agreement was found between yearly-averaged NH3 total columns simulated by the model and IASI satellite measurements over the IGP, South Asia (r = 0.85) and North China Plain (NCP), of East Asia (r = 0.88) with a moderate correlation coefficient. Model simulated surface NH3 concentrations and reported the under predication with the measured surface/ground based NH3 concentration of online pollution monitoring sites of India. The manuscript adds some new information on existing information over Indian sub-continent with model prediction which is compared with online NH3 monitoring sits of CPCB of India. There is lot of issues / questions about the quality of the ground based data sets which is used in the comparison of model. The present study fails to establish the NH3 emissions/scenario over Asian region due to lack of model comparison with quality controlled information of NH3.

Major issues

The NH3 and NOx datasets are used in comparison of model are taken from the online monitoring sites of Central Pollution Control Board (CPCB), India are not quality controlled. There are a lot of issues related to calibration and validation NH3. The comparison of model should also be based on available quality controlled data sets published in a peer reviewed journals.

The instruments used in NH3 and NOx at CPCB sites are molybdenum based which converts all the gaseous nitrogen at 980oC in Nitric oxide (NO) and NOx in to NO (at 350oC). The difference of these two provides the NH3. Due to available moisture in the atmosphere and conversion of all gaseous nitrogen species at very high temperature it provide/estimate the high ambient NH3 concentration. Hence, for that weekly NH3 calibration is required with certified NH3 span gases.

A comparison of surface NH3 has been performed by Saraswati et al. (2019) (published in Mapan 34 (1):56-69) based on pollution monitoring sites (4 sites in Delhi) with quality controlled measurement of NH3 and reported the 2-3 times more concentration of NH3 over Delhi in compared with quality controlled data. The similar observations are reported in this manuscript in Figure 8b. In this manuscript model under predicted surface NH3 concentration. Authors are suggested the validate the model with published quality controlled datasets.

Fig 4a and Fig. 6 shows the over prediction of NH3 emission by MOZART models
which should be validated by quality controlled datasets. The panels are showing that sand, rocks and hillocks regions are emitting the NH3. It is sowing lack of experience/knowledge of Indian co-authors (it seems that most of the co-authors have not hands on experience/expertise of NH3 measurements).

Lot of publications are available on NH3 concentration and NH3 emissions from various Indian regions. Some of them listed below which can be used for model comparison. Few papers are only sited in the manuscript.

Aneja VP, Schlesinger WH, Erisman JW, Behra SN, Sharma M (2012) Reactive nitrogen emissions from crop and livestock farming in India. Atmos Environ 47:92-103 Asman WA, Sutton MA, Schjorring JK (1998) Ammonia: emission, atmospheric transport and deposition. New Phytol 139(1): 27-48 Banerjee B, Pathak H, Aggarwal PK (2002) Effects of dicyandiamide, farmyard manure and irrigation on crop yields and ammonia volatilization from an alluvial soil under a rice (Oryza sativa L.)-wheat (Triticum aestivum L.) cropping system. Biol Fertil Soil 36: 207–214 Banerjee T, Singh S B,Srivastava RK (2011) Development and performance evaluation of statistical models correlating air pollutants and meteorological variables at Pantnagar, India. Atmos Res 99: 505-517 Behera SN, Betha R, Balasubramanian R (2013) Insights into chemical coupling among acidic gases, ammonia and secondary inorganic aerosols. Aerosol Air Qual Res 13(4): 1282-96 Behera SN, Sharma M, Aneja VP, Balasubramanian R (2013) Ammonia in the atmosphere: a review on emission sources, atmospheric chemistry and deposition on terrrestrial bodies. Environ Sci Pollut Res 20(11): 8092-8131 Biswas H, Chatterjee A, Mukhopadhya SK, De TK, Sen S, Jana TK (2005) Estimation of ammonia exchange at the land-ocean boundary condition of Sundarban mangrove north-est coast of Bay of Bengal, India. Atmos Environ 39: 4489-4499 Carmichael GR, Ferm M, Thongboonchoo N, Woo JH, Chan LY, Murano K, Viet PH, Mossberg C, Bala R, Boonjawat J, Upatum P, Mohan M, Adhikary SP, Shrestha AB, Pinaar JJ, Brunke EB, Chen T, Jie T, Guoan D, Peng LC, Dhiharto S, Harjanto H, Jose AM, Kimani W, Kirouane A, Lacaus J-P, Richard S, Barturen O, Cerda JC, Athayde A, Tavares

T, Cotrina JS, Bilici E (2003) Measurements of sulfur dioxide, ozone and ammonia concentration in Asia, Africa and South America using passive samplers. Atmos Environ 37: 1293 – 1308 Datta A, Sharma SK, Harit RC, Kumar V, Mandal TK, Pathak H (2012) Ammonia emission from subtropical crop land area of India. Asia Pacific J Atmos Sci 48 (3): 275-281 Kapoor RK, Singh G, Tiwari S (1992) Ammonia concentration vis-a-vis meteorological conditions at Delhi India. Atmos Res 28:1-9. Katyal J C, Singh B, Vlek P L G, Buresh R J (1987) Efficient nitrogen use as affected by urea application and irrigation sequence. Soil Sci Soc Am J 51: 366–370. Katyal J C, Singh B, Vlek P L G, Buresh R J (1987) Efficient nitrogen use as affected by urea application and irrigation sequence. Soil Sci Soc Am J 51: 366–370. Khemani LT, Momin GA, Naik MS, Rao PP, Safai PD, Murty ASR, (1987) Influence of alkaline particulates on pH of cloud and rain water in India. Atmos Environ 21:1137-1145 Kirchner M, Braeutigam S, Feicht E, Löflund M, (2002) Ammonia emissions from vehicles and the effects on ambient air concentrations. Fresen Environ. Bull 11:454-458 Kulshrestha UC, Sarkar AK, Srivastava SS, Parasar DC (1996) Investigation into atmospheric deposition through precipitation studies at New Delhi (India). Atmos Environ 30: 4149 – 4154 Mitra A P, Sharma C (2002) Indian aerosols: present status. Chemosphere 49(9): 1175-1190. Mosier A R, Wassmann R, Verchot L, King J. Palm C (2004) Methane and nitrogen oxide fluxes in tropical agricultural soils: Sources, sinks and mechanisms. Environ Dev Sustain 6: 11–49. Olivier JGJ, Bouwman AF, Van der Hoek KW, Berdowski JJM (1998) Global air emission inventories for anthropogenic sources of NOx, NH3 and N2O in 1990. Environ Pollut 102:135-148 Parashar DC, Granat L, Kulshreshtha UC, Pillai AG, Naik MS, Momin GA, Rao PSP, Safai PD, Khemani LT, Naqavi SWA, Narverkar PV, Thapa KB, Rodhe H (1996) Report CM-90 September 1996, Department of meteorology, Stockholm University International Meteorological Institute in Stockholm (Sweden). Parmar RS, Satsangi GS, Lakhani A, Srivastava SS, Prakash S (2001) Simultaneous measurements of ammonia and nitric acid in ambient air at Agra (27o10'N and 78o05'E) (India). Atmos Environ 35: 5979 – 5988 Patel S K, Panda D, Mohanty S K (1989) Relative ammonia loss from urea based fertilizers applied to rice under different hydrological situations. Fert Res 19: 113–119 Pathak H, Li C, Wassmann R, Ladha J K (2006) Simulation of nitrogen balance in the rice–wheat systems of the Indo-Gangetic plains. Soil Sci Soc Am J 70: 1612–1622 Paulot F, Jacob DJ, Pinder RW, Bash JO, Travis K, Henze DK (2014) Ammonia emissions in the United States, European Union, and China derived by high-resolution inversion of ammonium wet deposition data: interpretation with a new agricultural emissions inventory (MASAGE_NH3). J Geophys Res-Atmos. 119(7):4343–64 Perrino C, Catrambone M, Di Bucchianico ADM, Allegrini I, (2002) Gaseous ammonia in the urban area of Rome Italy and its relationship with traffic emissions. Atmos Environ 36:5385-5394 Santra G H, Das D K, Mandal L N (1988) Loss of nitrogen through ammonia volatilization from flooded rice fields. J Indian Soc Soil Sci 36: 652–659. Saraswati, George MP, Sharma SK, Mandal TK, Kotnala RK, (2019a) Simultaneous measurements of ambient NH3 and its relationship with other trace gases, PM2.5 and meteorological parameters over Delhi, India. Mapan-Journal of Metrology Society of India 34 (1):56-69 Saraswati, Sharma SK, Mandal TK, (2018) Five-year measurement of ambient ammonia and its interaction with other trace gases at an urban site of Delhi, India. Meteo Atmos Phys 130 (2): 241-257 Saraswati, Sharma SK, Saxena, M, Mandal TK (2019b) Characteristics of gaseous and particulate ammonia and their role in the formation of secondary inorganic particulate matter at Delhi, India. Atmos Res 218: 34-49. Sarkar M C, Banerjee N K, Rana D S, Uppal K S (1991) Field measurements of ammonia volatilization losses of nitrogen from urea applied to wheat. Fert News 25–28. Sharma C, Tiwari MK, Pathak H, (2008) Estimates of emission and depostion of reactive nitogenous species for India. Current Science 94(11): 1439-1446. Sharma M, Kishore S, Tripathi SN, Behera SN, (2007) Role of atmospheric ammonia in the formation of inorganic secondary particulate matter: a study at Kanpur, India. J Atmos Chem 58(1): 1-17 Sharma S K, Saxena M, Saud T, Korpole S, Mandal TK, (2012c) Measurement of NH3, NO, NO2 and related particulates at urban sites of Indo Gangetic Plain (IGP) of India. J Sci Indust. Res 71 (5): 360-362. Sharma SK, Kumar M, Rohtash, Gupta NC, Saraswati, Saxena M, Mandal TK (2014d) Characteristics of ambient ammonia over Delhi, India. Meteo Atmos Phy 124: 67-82

Sharma SK, Datta A, Saud T, Mandal TK, Ahammed YN, Arya BC, Tiwari MK (2010a). Study on concentration of ambient NH3 and interaction with some other ambient trace gases. Environ Monit Asses 162:225-235 Sharma SK, Datta A, Saud T, Saxena M, Mandal TK, Ahammed YN, Arya BC (2010b) Seasonal variability of ambient NH3, NO, NO2 and SO2 over Delhi. J Environ Sci 22 (7): 1023-1028 Sharma SK, Harit RC, Kumar V, Mandal TK, Pathak H, (2014a) Ammonia emission from rice-wheat cropping system in subtropical soil of India. Agril Res 3(2): 175-180. Sharma SK, Mandal TK, Rohtash, Kumar M, Gupta NC, Pathak H, Harit RC, Saxena M, (2014b) Measurement of ambient ammonia over the National Capital Region of Delhi, India. Mapan-Journal of Metrology Society of India 29 (3): 165-173 Sharma SK, Mandal TK, Sharma C, Kuniyal JC, Joshi R., Dhayani PP, Rohtash, Sen A, Ghayas H, Gupta NC, Arya BC, Kumar A, Sharma P, Saxena M, Sharma A (2014c). Measurements of particulate (PM2.5), BC and trace gases over the northwestern Himalayan region of India. Mapan-Journal of Metrology Society of India 29 (4): 243-253 Sharma SK, Rohtash, Mandal TK, Deb NC, Pal S (2016) Measurement of ambient NH3, NO and NO2 at an urban area of Kolkata, India. Mapan-Journal of Metrology Society of India 31 (1):75-80 Sharma SK, Saraswati, Mandal TK, Saxena M (2017) Inter-annual variation of ambient ammonia and related trace gases in Delhi, India. Bull Environ Contamin Toxicol 99(2):281-285 Sharma SK, Saxena M, Mandal TK, Ahammed YN, Pathak H, Datta A, Saud T, Arya BC, (2011) Variations in mixing ratios of ambient ammonia, nitric oxide and nitrogen dioxide in different environments of India. J Earth Science & Climate Change 1:102. Sharma SK, Singh AK, Saud T, Mandal TK, Saxena M, Singh S, Ghosh S, Raha S, (2012a) Study on water soluble ionic composition of PM10 and trace gases over Bay of Bengal during W_ICARB campaign. Meteo Atmos Phys 118: 37-51 Sharma SK, Singh AK, Saud T, Mandal TK, Saxena M, Singh S, Ghosh S, Raha S, (2012b) Measurement of ambient NH3, NO, NO2 and SO2 over Bay of Bengal during W_ICARB campaign. Annales Geophysicae 30: 371-377 Singh N, Murari V, Kumar M, Barman SC, Banerjee T (2017) Fine particulates over South Asia: review and meta-analysis of PM2.5 source apportionment through receptor model. Environ Pollut 223: 121-136

There are also several issues with this comparative study that needs to be taken care by Indian co-authors. They are familiar with the scenario, mainly fertilizer used and NH3 emissions from the agricultural activities.

Such type of over predication/publication of NH3 emission from Indian sub-continent may create the havoc in future. We faced the problem of CH4 emission from rice/paddy fields in India. Hence, quality controlled datasets should be used in model comparison with experimental experts.

---

## Referee Comment (RC1) · Anonymous Referee #2 · 9 Oct 2020

The authors present an analysis of atmospheric ammonia over South and East Asia based on the MOZART-4 model that is driven by the HTAP-v2 emission inventory. Model results are compared against IASI satellite observations (total column), as well as surface observations of CPCB (India) and NNDMN (China) for the year 2010. This topic is very important, since ammonia partitions into the only ubiquitous volatile cation, i.e., ammonium (NH4+). NH4+ plays a crucial role in air quality and visibility due to its volatility and ability to neutralize acidic air pollutants, which are often of anthropogenic origin. And despite the various air pollution abatement efforts, ammonia concentrations are increasing in many regions of the world and are thus still of concern, not only in Asia. Despite some fundamental weakness in the modelling approach (which is unfortunately common to most such modeling studies and therefore is not a reason for rejection), this study is overall sufficiently sound. I would therefore recommend publication, if the authors take the following comments and discussion points into account.

The study reveals that spatial differences (total column) between MOZART-4 and IASI are generally largest during local autumn / winter season, with an overestimation compared to IASI observations. This overestimation is most pronounced for IGP South Asia (20°N-32°N, 70°E-95°E), while rather an underestimation is found for NCP East Asia (30°N-40°N, 110°E-120°E), especially during the summer months. On the other hand, the comparison of surface concentrations reveals that the model underestimates the ammonia observations over South and East Asia throughout the year. This is shown by monthly mean (time series) and annual averages (scatter plot), and these results are in contrast to the total column case (model burden w.r.t. IASI observations).

Despite some potential calibration issue w.r.t. certain observations, there seems to be no obvious inconsistency with the NH3 observations used in this study. Instead, both issues (model vs surface and total column observations) rather point to an incomplete model set-up w.r.t. the gas-aerosol partitioning assumptions. Nevertheless, I also recommend that the authors make sure that the study is based on (or includes) quality controlled surface observations.

Regarding the modeling assumptions, it should be noted that the chosen set-up has its limitations w.r.t. the NH3/NH4+ partitioning. The main issue here is that in the current set-up, both (i) cations other than NH4+, e.g., sodium (Na+), potassium (K+), calcium (Ca2+), and magnesium (Mg2+), have been neglected, as well as (ii) organic acids were omitted for the gas-aerosol partitioning calculations. Both are, however, important for the NH3/NH4+ partitioning w.r.t. to real world observations. Nevertheless, since mineral cations and organic acids have been neglected in conjunction, the presented model results could be in terms of yearly averages more or less "right" for the wrong reason, as indicated by a study published some times ago in ACP 2006 (https://acp.copernicus.org/articles/6/2549/2006/). On shorter time scales, however, the incomplete model set-up could be a cause of the observed discrepancies.

The reason is that in this model set-up, the NH3/NH4+ partitioning is mainly controlled by sulfate and subsequently by nitrate, which might be in reality not the case in Asia. Consideration of at least the major mineral cations (e.g., Na+, K+, Ca2+,Mg2+) might be necessary, since all of them are ubiquitous and preferentially neutralize sulfate, which directly affects the NH3/NH4+ partitioning. In contrast to the semi-volatile compound ammonium nitrate (NH4NO3), mineral cations form more stable compounds that exhibit a distinct different temperature dependent dissociation and water uptake, but no volatilization, as it is here the case only for NH4NO3. Thus, consideration of additional (mineral) cations could lead to more free ammonia (w.r.t. sulfate neutralization), which, in addition could lead to a larger fraction of ammonia being neutralized by nitric acid (e.g. resulting from lightning and thus adding up in the vertical model column as ammonium nitrate). And, since NH4NO3 is unstable at higher temperatures and low humidities, both cases could result in higher simulated NH3 concentrations during the summer months — resulting in potentially closer NH3 total column concentrations w.r.t. IASI observations.

Also, the underestimation of the surface NH3 concentrations throughout the year over both South and East Asia could be a result of missing mineral cations in this model set-up. In reality, a larger fraction of sulfate might be neutralized by mineral cations rather than just by ammonium, which could lead to a larger fraction of free ammonia near the surface. Also, since both nitrates and sulfates preferentially react with mineral cations, nitric acid (e.g. from the traffic sector) might be neutralized by ammonia in a lower amount in reality, as it seems to be the case in this model set-up. In any case, consideration of mineral cations could also lead to a larger fraction of free ammonia near the surface, which might be even sufficient to explain discrepancies with surface observations.

Furthermore, due to the excess of ammonia in this model set-up, ammonium nitrate can be formed in both regions, although the simulated sulfate concentrations (burden) are higher in East Asia compared to South Asia. And, due to its semi-volatile character, the seasonal variability of NH4NO3 and the associated NH3 concentrations differ in both regions as observed. Since NH4NO3 is unstable at higher temperatures, more NH3 bound as NH4NO3 (compared to ammonium sulfate) can lead to higher NH3 concentrations during summer, as it is observed in East Asia. In South Asia, where both ammonia and sulfate concentrations are lower, also NH4NO3 concentrations are lower and thus the seasonality of NH3 is less pronounced, which is consistent with the surface observations.

On the other hand, the overestimation of the IASI total column NH3 concentrations over South Asia, for most of the year except the summer months, could be also a result of missing anions, e.g., of organic acids, assuming the vertical exchange processes are more or less realistically modelled. However, considering mineral cations without additional acids, could likely cause even larger differences in this case (for details see e.g., https://acp.copernicus.org/articles/6/2549/2006/).

Unfortunately, these processes (briefly touched on above) are missing in most modelling studies, and I fear their consideration is also beyond the scope (or possibilities) of this study?

---

## Referee Comment (RC2) · Anonymous Referee #3 · 19 Oct 2020

Ammonia is an important short-lived pollutant with a huge global relevance for air quality, biodiversity and climate due to the wide spread food production. Improving the nitrogen use efficiency in agriculture is of key importance, which requires an understanding of the nitrogen budgets and the ability to monitor these. The atmospheric ammonia burden is difficult to model, and hence, improving our modelling capacity is an important activity. After reading the paper in detail I recommend a major revision is required to improve the paper to a level which is beyond a simple comparison between a coarse model field and observations, which is currently basically is.

A major drawback of this study is the coarse resolution the modelling is performed on.

Not only in a spatial sense, also the output is available on 4 hours of the day, with IASI overpass (9:30) right in between the output times (06 and 12). The description of the comparison to the satellite data is very short. Giving the strong diurnal cycle of ammonia and the fact that the satellite data availability is affected by all kinds of factors I would like to see a much more detailed description on the method and the impacts of the choices made. - Were the monthly mean comparisons made by averaging paired observations across the month? How many valid pairs were required to allow for a valid number? If pairing was not done than a motivation/discussion why this is not important should be included. Normally the large degree of variability of ammonia column densities between days requires to pair. Satellite data availability and patterns in these within a large grid cell can also impact a non-paired comparison. - How was the modelled column for 09:30 estimated? Later I read that a daily mean model value is used... correct? - Which quality flags of the satellite data were used? - In our experience the diurnal emission cycle largely impacts the ammonia columns at overpass. What was assumed in this study? Given the agricultural practices in India, is it warranted to use a flat emission cycle across the year?

The paper is severely hampered by the coarse comparison and I am afraid that the comparison methodology may impact the systematic differences seen in this paper. The differences between overpass time and a daily mean for instance relate to the daylength (variability) and associated mixing, diurnal emission cycle, frequency and kind of precipitation events, etc. I would have like to see an analysis / consideration of such factors in this paper. Part of the observations might be useful for this purpose.

The discussion does not include a comparison to other modelling studies evaluating ammonia levels across Asia or studies on ammonia life time.

I could identify many grammar mistakes in the english language use. The author list includes native speakers and I would like to urge to perform a careful language check.

Minor comments:

Abstract: Please use past tense for the method description

Introduction The introduction focusses mostly on the contribution of different agricultural activities to emission estimates in south and east Asia. The challenges with respect to the emission estimation, spatial and temporal emission variability, chemistry transport modelling and model-satellite comparison are not focused on although these are relevant to the paper and partly addressed. I would like to ask the authors to address these issues in the intro.

Line 43: chemical should be syntethic Line 50: 64 % of total means total global? if yes line 53 repeats this statement Line 60-62: could you use the recent edgar numbers or thise from v4.3? Should these statements be presented with the global comparison the paragraph above? Line 63 and 67 are in direct contradiction to each other

Data and methodology Line 85: this sentence implies only trace gases were modelled, which is not the case I guess

Line 97: Does Mozart use a land use mosaic within a gridcell? Or dominant LUC? How do the wesely land use classes match those in the domain? Were the latter updated? Line 122: didn't you use emissions of all sectors? Line 133: cow dung is not fossil

Results: Line 226: the methodology describes that nitrate is present – please explain 250: the model has no maximum emissions in summer as antrop is flat and soil is a few percent of total, so this statement seems incorrect

Figure 2: the scale on the upper left figure is misleading. It seems a seasonal cycle where it is basically flat.

---

## Author Comment (AC1) · 26 Oct 2020

We thank the reviewer for his short comments and suggestions.

Below are our replies to the reviewer's short comments.

**Reviewer's short comments (*Bold Italic*) and author's response (Red font):-**

**General Comments:-**

*Paper present the comparison of the MOZART-4 model along with monthly averaged satellite distributions of ammonia emission across South Asia. The authors are trying to identify the northern region of India i.e., Indo-Gangetic Plain, IGP as a hotspot for $NH_3$ in Asia, both using the model and satellite observations. They highlighted a close agreement was found between yearly-averaged $NH_3$ total columns simulated by the model and IASI satellite measurements over the IGP, South Asia (r=0.85) and North China Plain (NCP), of East Asia (r=0.88) with a moderate correlation coefficient. Model simulated surface $NH_3$ concentrations and reported the under prediction with the measured surface/ground based $NH_3$ concentration of online pollution monitoring sites of India. The manuscript adds some new information on existing information over Indian sub-continent with model prediction which is compared with online $NH_3$ monitoring sits of CPCB of India. There is lot of issues/questions about the quality of the ground based data sets which is used in the comparison of model. The present study fails to establish the $NH_3$ emissions/scenario over Asian region due to lack of model comparison with quality controlled information of $NH_3$.*

**Reply:**

We want to bring to reviewers notice that the objective of the current study is to examine the Spatio-temporal variability of atmospheric $NH_3$ over Asia (both South and East Asia) and focus on two hotspots regions of ammonia, the Indo-Gangetic Plain (IGP) and the North China Plain (NCP) using chemical transport modeling, satellite observations and *in-situ* ammonia measurements to understand why certain emission hotspot regions in East Asia show lower $NH_3$ total columns compared with similar hotspot regions in South Asia, when analyzed with both model and satellite observations. The quality control and assurance method, followed by Central Pollution Control Board (CPCB) for these air quality monitoring stations, is given in the Central Pollution Control Board (2020). Furthermore, we take the following steps to reassure the quality of $NH_3$ observations from the CPCB network stations. For data quality, we rejected all the observations values below the lowest detection limit of the instrument (1 μg/m$^3$) (Technical specifications for CAAQM station, 2019) because most of the sites are situated in the urban environment. For cities where more than one monitoring station is available, we rejected all the observations above 250 μg/m$^3$ at a given site if other sites in the network do not show values outside this range. This step aims to eliminate any short-term local influence that cannot be captured in the models and retain the regional-scale variability. Second, we

removed single peaks characterized by a change of more than 100 µg/m3 in just one hour for all the data in CPCB monitoring stations. This step filters random fluctuations in the observations. Third, we removed some very high $NH_3$ values that appeared in the time series right after the missing values. For any given day, we removed the sites from the consideration that either experience instrument malfunction and appear to be very heavily influenced by strong local sources.

**Specific Comments:-**

1. *The NH3 and NOx datasets are used in the comparison of the model are taken from the online monitoring sites of the Central Pollution Control Board (CPCB), India are not quality controlled. There are a lot of issues related to calibration and validation NH3. The comparison of model should also be based on available quality controlled data sets published in a peer reviewed journals.*

   **Reply**
   We would like to bring to reviewers notice that we have not used nor compared the NOx data set in the present study. We have used the recent 69 stations $NH_3$ datasets of Central Pollution Control Board (CPCB)over South Asia and 32 stations of the Nationwide Nitrogen Deposition Monitoring Network (NNDMN) over East Asia. The quality assurance and control process followed for these air quality monitoring instruments is done as mentioned above and given in (Pollution and Board (2011), CPCB (2011),Central Pollution Control Board (2020) and Technical specifications for CAAQM station (2019)).

2. *The instruments used in NH3 and NOx at CPCB sites are molybdenum based which converts all the gaseous nitrogen at 980oC in Nitric oxide (NO) and NOx into NO (at $350^0$ C). The difference of these two provides the NH3. Due to available moisture in the atmosphere and conversion of all gaseous nitrogen species at very high temperature it provide/estimate the high ambient NH3 concentration. Hence, for that weekly NH3 calibration is required with certified NH3 span gases.*

   **Reply**
   Yes, we agree with the reviewers' observations that the Conventional $NH_3$ measurement technique (chemiluminescence method using *molybdenum based*) overestimates ammonia measurements. This is one of the shortcomings of the chemiluminescence based measurements, which was already discussed in this manuscript. CPCB follows strict calibration of the instrument as per Pollution and Board (2011), CPCB (2011) and Technical specifications for CAAQM station (2019). Additional quality control and assurance are followed in this study is mention above.

3. *A comparison of surface NH3 has been performed by Saraswati et al. (2019) (published in Mapan 34 (1):56-69) based on pollution monitoring sites (4 sites in Delhi) with quality controlled measurement of NH3 and reported the 2-3 times more concentration of NH3 over Delhi in compared with quality controlled data. The similar observations are reported in this manuscript in Figure 8b. In this manuscript model under predicted surface NH3 concentration. Authors are suggested to validate the model with published quality controlled datasets.*

**Reply**

Saraswati et al. (2019) carried out a comparison of $NH_3$ measured at NPL site in New Delhi with four other air quality measurements sites (Anand Vihar, Mandir Marg, Punjabi Bagh and R.K. Puram) in Delhi operated Pollution Control Committee (DPCC)) using a similar type of instrument and reported higher $NH_3$ values from other monitoring sites when compared to the reference NPL station with more than 50 % Normalised Mean Bias (NMB). However, all four sites were situated at entirely different locations from the reference site (NPL) in Saraswati et al. (2109) and may not provide the actual information on the comparison. Since $NH_3$ varies significantly from one location to another in Delhi, the reference site's difference partly could be due to the difference in the $NH_3$ measurement location. Ideally, such a comparison is made at the same location to get meaningful results.

In order to verify the data quality, we have compared the $NH_3$ measurements at R.K. Puram site operated by CPCB with our own limited measurements available at IGI airport (close to R.K. Puram site) for the winter period using MARGA instrument. Our comparison shows that mean CPCB measurements are slightly on higher side than our own measurements (see below figure). The difference of 9.8 µg m$^{-3}$ between both the measurement technique (chemiluminescence and ion chromatography (IC) based) indicates that the $NH_3$ measurements from the CPCB do not suffer from the calibration issue. However, rigorous validation is required in the future.

[Figure]

**Figure:** Comparison of $NH_3$ concentration from MARGA instrument (IGI airport) with R.K Puram (CPCB) station

4. *Fig 4a and Fig. 6 shows the over prediction of NH3 emission by MOZART models which should be validated by quality controlled datasets. The panels are showing that sand, rocks and hillocks regions are emitting the NH3. It is sowing lack of experience/ knowledge of Indian co-authors (it seems that most of the co-authors have not hands on experience/expertise of NH3 measurements).*

**Reply**

The comparison of surface concentrations reveals that the model underestimates the ammonia observations over both South and East Asia throughout the year, which is shown by monthly mean (time series) and annual averages (scatter plot). Despite some potential calibration issues w.r.t. individual observations which are mentioned in the revised manuscript, there seems to be no obvious inconsistency with the $NH_3$ observations used in this study.

*5. Lot of publications are available on NH3 concentration and NH3 emissions from various Indian regions. Some of them listed below which can be used for model comparison. Few papers are only sited in the manuscript.Aneja VP, Schlesinger WH, Erisman JW, Behra SN, Sharma M (2012) Reactive nitrogenemissions from crop and livestock farming in India. Atmos Environ 47:92-103 Asman WA, Sutton MA, Schjorring JK (1998) Ammonia: emission, atmospheric transport and deposition. New Phytol 139(1): 27-48 Banerjee B, Pathak H, Aggarwal PK (2002) Effects of dicyandiamide, farmyard manure and irrigation on crop yields and ammonia volatilization from an alluvial soil under a rice (Oryzasativa L.)-wheat (Triticumaestivum L.) cropping system.BiolFertil Soil 36: 207–214 Banerjee T, Singh S B,Srivastava RK (2011) Development and performance evaluation of statistical models correlating air pollutants and meteorological variables at Pantnagar, India. Atmos Res 99: 505-517 Behera SN, Betha R, Balasubramanian R (2013) Insights into chemicalcoupling among acidic gases, ammonia and secondary inorganic aerosols. Aerosol Air Qual Res 13(4): 1282-96 Behera SN, Sharma M, Aneja VP, Balasubramanian R (2013) Ammonia in the atmosphere: a review on emission sources, atmospheric chemistry and deposition on terrrestrial bodies. Environ SciPollut Res 20(11): 8092-8131 Biswas H, Chatterjee A, Mukhopadhya SK, De TK, Sen S, Jana TK (2005) Estimation of ammonia exchange at the land-ocean boundary condition of Sundarban mangrove north-est coast of Bay of Bengal, India. Atmos Environ 39: 4489-4499 Carmichael GR, Ferm M, Thongboonchoo N, Woo JH, Chan LY, Murano K, Viet PH, Mossberg C, Bala R, Boonjawat J, Upatum P, Mohan M, Adhikary SP, Shrestha AB, Pinaar JJ, Brunke EB, Chen T, Jie T, Guoan D, Peng LC, Dhiharto S, Harjanto H, Jose AM, Kimani W, Kirouane A, Lacaus J-P, Richard S, Barturen O, Cerda JC, Athayde A, Tavares T, Cotrina JS, Bilici E (2003) Measurements of sulfur dioxide, ozone and ammonia concentration in Asia, Africa and South America using passive samplers. Atmos Environ 37: 1293 – 1308 Datta A, Sharma SK, Harit RC, Kumar V, Mandal TK, Pathak H (2012) Ammonia emission from subtropical crop land area of India. Asia Pacific J AtmosSci 48 (3): 275-281 Kapoor RK, Singh G, Tiwari S (1992) Ammonia concentration vis-à-vis meteorological conditions at Delhi India. Atmos Res 28:1-9. Katyal J C, Singh B, Vlek P L G, Buresh R J (1987) Efficient nitrogen use as affected by urea application and irrigation sequence. Soil SciSoc Am J 51: 366–370. Katyal J C, Singh B, Vlek P L G, Buresh R J (1987) Efficient nitrogen use as affected by urea application and irrigation*

*sequence. Soil SciSoc Am J 51: 366–370. Khemani LT, Momin GA, Naik MS, Rao PP, Safai PD, Murty ASR, (1987) Influence of alkaline particulates on pH of cloud and rain water in India.Atmos Environ 21:1137-1145 Kirchner M, Braeutigam S, Feicht E, Löflund M, (2002) Ammonia emissions from vehicles and the effects on ambient air concentrations. Fresen Environ. Bull 11:454-458 Kulshrestha UC, Sarkar AK, Srivastava SS, Parasar DC (1996) Investigation into atmospheric deposition through precipitation studies at New Delhi (India). Atmos Environ 30: 4149 – 4154 Mitra A P, Sharma C (2002) Indian aerosols: present status. Chemosphere 49(9): 1175-1190. Mosier A R, Wassmann R, Verchot L, King J. Palm C (2004) Methane and nitrogen oxide fluxes in tropical agricultural soils: Sources, sinks and mechanisms. Environ Dev Sustain 6: 11–49. Olivier JGJ, Bouwman AF, Van der Hoek KW, Berdowski JJM (1998) Global air emission inventories for anthropogenic sources of NOx, NH3 and N2O in 1990. Environ Pollut 102:135-148 Parashar DC, Granat L, Kulshreshtha UC, Pillai AG, Naik MS, Momin GA, Rao PSP, Safai PD, Khemani LT, Naqavi SWA, Narverkar PV, Thapa KB, Rodhe H (1996) Report CM-90 September 1996, Department of meteorology, Stockholm University International Meteorological Institute in Stockholm (Sweden). Parmar RS, Satsangi GS, Lakhani A, Srivastava SS, Prakash S (2001) Simultaneous measurements of ammonia and nitric acid in ambient air at Agra (27o10'N and 78o05'E) (India). Atmos Environ 35: 5979 – 5988 Patel S K, Panda D, Mohanty S K (1989) Relative ammonia loss from urea based fertilizers applied to rice under different hy- drological situations. Fert Res 19: 113–119 Pathak H, Li C, Wassmann R, Ladha J K (2006) Simulation of nitrogen balance in the rice–wheat systems of the Indo-Gangetic plains. Soil SciSoc Am J 70: 1612–1622 Paulot F, Jacob DJ, Pinder RW, Bash JO, Travis K, Henze DK (2014) Ammonia emissions in the United States, European Union, and China derived by high-resolution inversion of ammonium wet deposition data: interpretation with a new agricultural emissions inventory (MASAGE_NH3). J Geophys Res-Atmos. 119(7):4343–64 Perrino C, Catrambone M, Di Bucchianico ADM, Allegrini I, (2002) Gaseous ammonia in the urban area of Rome Italy and its relationship with traffic emissions. Atmos Environ 36:5385-5394 Santra G H, Das D K, Mandal L N (1988) Loss of nitrogen through ammonia volatilization from flooded rice fields. J Indian Soc Soil Sci 36: 652–659. Saraswati, George MP, Sharma SK, Mandal TK, Kotnala RK, (2019a) Simultaneous measurements of ambient NH3 and its relationship with other trace gases, PM2.5 and meteorological parameters over Delhi, India. Mapan- Journal of Metrology Society of India 34 (1):56-69 Saraswati, Sharma SK, Mandal TK, (2018) Five-year measurement of ambient ammonia and its interaction with other trace gases at an urban site of Delhi, India. MeteoAtmosPhys 130 (2): 241-257 Saraswati,*

*Sharma SK, Saxena, M, Mandal TK (2019b) Characteristics of gaseous and particulate ammonia and their role in the formation of secondary inorganic particulate matter at Delhi, India. Atmos Res 218: 34-49. Sarkar M C, Banerjee N K, Rana D S, Uppal K S (1991) Field measurements of ammonia volatilization losses of nitrogen from urea applied to wheat. Fert News 25–28. Sharma C, Tiwari MK, Pathak H, (2008) Estimates of emission and depostion of reactive nitogenous species for India. Current Science 94(11): 1439-1446. Sharma M, Kishore S, Tripathi SN, Behera SN, (2007) Role of atmospheric ammonia in the formation of inorganic secondary particulate matter: a study at Kanpur, India. J AtmosChem 58(1): 1-17 Sharma S K, Saxena M, Saud T, Korpole S, Mandal TK, (2012c) Measurement of NH3, NO, NO2 and related particulates at urban sites of Indo Gangetic Plain (IGP) of India. J SciIndust. Res 71 (5): 360-362. Sharma SK, Kumar M, Rohtash, Gupta NC, Saraswati, Saxena M, Mandal TK (2014d) Characteristics of ambient ammonia over Delhi, India.*

*MeteoAtmosPhy 124: 67-82 Sharma SK, Datta A, Saud T, Mandal TK, Ahammed YN, Arya BC, Tiwari MK (2010a). Study on concentration of ambient NH3 and interaction with some other ambient trace gases. Environ Monit Asses 162:225-235 Sharma SK, Datta A, Saud T, Saxena M, Mandal TK, Ahammed YN, Arya BC (2010b) Seasonal variability of ambient NH3, NO, NO2 and SO2 over Delhi. J Environ Sci 22 (7): 1023-1028 Sharma SK, Harit RC, Kumar V, Mandal TK, Pathak H, (2014a) Ammonia emission from rice-wheat cropping system in subtropical soil of India. Agril Res 3(2): 175-180. Sharma SK, Mandal TK, Rohtash, Kumar M, Gupta NC, Pathak H, Harit RC, Saxena M, (2014b) Measurement of ambient ammonia over the National Capital Region of Delhi, India. Mapan-Journal of Metrology Society of India 29 (3): 165-173 Sharma SK, Mandal TK, Sharma C, Kuniyal JC, Joshi R., Dhayani PP, Rohtash, Sen A, Ghayas H, Gupta NC, Arya BC, Kumar A, Sharma P, Saxena M, Sharma A (2014c). Measurements of particulate (PM2.5), BC and trace gases over the northwestern Himalayan region of India.Mapan-Journal of Metrology Society of India 29 (4): 243-253 Sharma SK, Rohtash, Mandal TK, Deb NC, Pal S (2016) Measurement of ambient NH3, NO and NO2 at an urban area of Kolkata, India. Mapan-Journal of Metrology Society of India 31 (1):75-80 Sharma SK, Saraswati, Mandal TK, Saxena M (2017) Inter-annual variation of ambient ammonia and related trace gases in Delhi, India. Bull Environ ContaminToxicol 99(2):281-285 Sharma SK, Saxena M, Mandal TK, Ahammed YN, Pathak H, Datta A, Saud T, Arya BC, (2011) Variations in mixing ratios of ambient ammonia, nitric oxide and nitrogen dioxide in different environments of India. J Earth Science & Climate Change 1:102. Sharma SK, Singh AK, Saud T, Mandal TK, Saxena M, Singh S, Ghosh S, Raha S, (2012a) Study on water soluble ionic composition of PM10 and trace gases over Bay of Bengal during W_ICARB campaign. MeteoAtmosPhys 118: 37-51 Sharma SK, Singh AK, Saud T, Mandal TK, Saxena M, Singh S, Ghosh S, Raha S, (2012b) Measurement of ambient NH3, NO, NO2 and SO2 over Bay of Bengal during W_ICARB campaign. AnnalesGeophysicae 30: 371-377 Singh N, Murari V, Kumar M, Barman SC, Banerjee T (2017) Fine particulates over South Asia: review and meta-analysis of PM2.5 source apportionment through receptor model. Environ Pollut 223: 121-136*

**Reply**

Thank you very much for citing some of the Indian references. We have updated some of the above relevant citations in the revised manuscript. We are aware that specific region Indian $NH_3$ measurements that are published in papers are available, but they have used the same chemiluminescence technique (model may differ).

In this study, we have used the global chemical transport model- MOZART4, and the domain is set to a horizontal grid resolution of $1.9° \times 2.5°$ and driven by the same grid resolution of HTAP-V2 emission inventory. Gridded $NH_3$ emission data for India is not yet available, and if available, it is not available in the public domain. Specific region data cannot be used to run the model. Hence, in our study, we suggested developing India's specific $NH_3$ emission inventory to run the regional and global chemical transport models.

*6. There are also several issues with this comparative study that needs to be taken care by Indian co-authors. They are familiar with the scenario, mainly fertilizer used and NH3 emissions from the agricultural activities. Such type of over predication/publication of NH3 emission from Indian sub-continent may create the havoc in future. We faced the problem of*

*CH4 emission from rice/paddy fields in India. Hence, quality controlled datasets should be used in model comparison with experimental experts.*

**Reply**

We would like to bring to the reviewer notice that we are not addressing $CH_4$ emissions in this manuscript. Our intention is not to create havoc but to examine the Spatio-temporal variability of atmospheric $NH_3$ over Asia using chemical transport modeling, satellite observations, and *in-situ* ammonia measurements to understand why certain emission hotspot regions in East Asia show lower $NH_3$ total columns compared with similar hotspot regions in South Asia, when analyzed with both model and satellite observations. We have done quality control and assurance, additionally, as mention previously, before using the data. Inherent issues related to chemiluminescence are still there, which we have mentioned in the manuscript.

**References**

1. Central Pollution Control Board (2020), [online] Available from: https://cpcb.nic.in/quality-assurance-quality-control/ (Accessed 26 May 2020).

2. CPCB: Guidelines for Real Time Sampling & Analyses, 2011.

3. Pollution, C. and Board, C.: Guidelines for Manual Sampling & Analyses., 2011.

4. Saraswati, George, M. P., Sharma, S. K., Mandal, T. K. and Kotnala, R. K.: Simultaneous Measurements of Ambient $NH_3$ and Its Relationship with Other Trace Gases, $PM_{2.5}$ and Meteorological Parameters over Delhi, India, Mapan - J. Metrol. Soc. India, 34(1), 55–69, doi:10.1007/s12647-018-0286-0, 2019.

5. Technical specifications for CAAQM station: TECHNICAL SPECIFICATIONS FOR CONTINUOUS AMBIENT AIR QUALITY MONITORING ( CAAQM ) STATION ( REAL TIME ) Central Pollution Control Board East Arjun Nagar , Shahdara., 2019.

---

## Author Comment (AC2) · 14 Dec 2020

**Response to Anonymous Referee #1's Comments**

First of all we thank the reviewer for the positive evaluation of our study and sincerely appreciate the reviewer's insightful and helpful comments.

Below we explicitly respond to each of the items raised in the comments of anonymous referee #1. These comments are indicated in **bold**, whereas the author's response is presented in blue.

**R1C1:**

**The authors present an analysis of atmospheric ammonia over South and East Asia based on the MOZART-4 model that is driven by the HTAP-v2 emission inventory. Model results are compared against IASI satellite observations (total column), as well as surface observations of CPCB (India) and NNDMN (China) for the year 2010. This topic is very important, since ammonia partitions into the only ubiquitous volatile cation, i.e., ammonium ($NH_4^+$). $NH_4^+$ plays a crucial role in air quality and visibility due to its volatility and ability to neutralize acidic air pollutants, which are often of anthropogenic origin. And despite the various air pollution abatement efforts, ammonia concentrations are increasing in many regions of the world and are thus still of concern, not only in Asia. Despite some fundamental weakness in the modelling approach (which is unfortunately common to most such modeling studies and therefore is not a reason for rejection), this study is overall sufficiently sound. I would therefore recommend publication, if the authors take the following comments and discussion points into account.**

- We thank the reviewer for carefully reading the manuscript. We agree that the suggested discussion will improve the readability of the manuscript.

**R1C2:**

**The study reveals that spatial differences (total column) between MOZART-4 and IASI are generally largest during local autumn/winter season, with an overestimation compared to IASI observations. This overestimation is most pronounced for IGP South Asia (20°N-32°N, 70°E-95°E), while rather an underestimation is found for NCP East Asia (30°N-40°N, 110°E-120°E), especially during the summer months. On the other hand, the comparison of surface concentrations reveals that the model underestimates the ammonia observations over South and East Asia throughout the year. This is shown by monthly mean (time series) and annual averages (scatter plot), and these results are in contrast to the total column case (model burden w.r.t. IASI observations).**

**Despite some potential calibration issue w.r.t. certain observations, there seems to be no obvious inconsistency with the $NH_3$ observations used in this study. Instead, both issues (model vs surface and total column observations) rather point to an incomplete model**

**set-up w.r.t. the gas-aerosol partitioning assumptions. Nevertheless, I also recommend that the authors make sure that the study is based on (or includes) quality controlled surface observations.**

- The quality control and assurance method followed by CPCB for these air quality monitoring stations, is given at Central Pollution Control Board (2020). Furthermore, we take the following steps to reassure the quality of $NH_3$ observations from the CPCB network stations. For data quality, we rejected all the observations values below 1 µg/m$^3$ and above 250 µg/m$^3$ at a given site if other sites in the network do not show values outside this range. The purpose of this step is to eliminate any short-term local influence that cannot be captured in the models and to retain the regional-scale variability. Second, we removed single peaks that are characterized by a change of more than 100 µg/m$^3$ in just one hour for all the data in CPCB monitoring stations. This step filters random fluctuations in the observations. Third, we removed some very high $NH_3$ values that appeared in the time series right after the missing values. For any given day, we removed the sites from the consideration that either experience instrument malfunction and/or appear to be very heavily influenced by strong local sources. This information is updated in the revised manuscript.

**R1C3:**

**Regarding the modeling assumptions, it should be noted that the chosen set-up has its limitations w.r.t. the $NH_3/NH_4^+$ partitioning. The main issue here is that in the current set-up, both (i) cations other than $NH_4^+$, e.g., sodium ($Na^+$), potassium ($K^+$), calcium ($Ca^{2+}$), and magnesium ($Mg^{2+}$), have been neglected, as well as (ii) organic acids were omitted for the gas-aerosol partitioning calculations. Both are, however, important for the $NH_3/NH_4^+$ partitioning w.r.t. to real world observations. Nevertheless, since mineral cations and organic acids have been neglected in conjunction, the presented model results could be in terms of yearly averages more or less "right" for the wrong reason, as indicated by a study published sometimes ago in ACP 2006 (https://acp.copernicus.org/articles/6/2549/2006/). On shorter time scales, however, the incomplete model set-up could be a cause of the observed discrepancies.**

- In this present work, the ammonium nitrate distribution is determined from $NH_3$ emissions and the parameterization of gas/aerosol partitioning by Metzger et al. (2002), which is a set of approximations to the equilibrium constant calculation (Seinfeld et al., 1998), based on the level of sulphate present. We followed, equilibrium simplified aerosol model (EQSAM)-Metzger et al. (2002) gas-aerosol partitioning calculations. The assumptions used in this study are limited to the ammonium-sulfate-nitrate-water system which is valid for only inorganic salt compounds. This latter was updated-EQSAM2 which additionally accounted for organic acids (Metzger et al., 2006). The application of any equilibrium models (EQMs) in global atmospheric studies is associated with considerable uncertainties. Metzger et al. (2006) found that the total ammonium calculated by ammonium-sulfate-nitrate-sodium-chloride-water system was about 15 % lower than that calculated by Equilibrium Simplified Aerosol Model 2 (EQSAM2) which includes additional organic acids. Ammonia has stronger affinity towards

neutralization of sulphuric acid ($H_2SO_4$) than nitric acid ($HNO_3$) whereas formation of ammonium chloride ($NH_4Cl_{(s \; or \; (aq)}$) in atmosphere is unstable and can dissociate reversibly to $NH_3$ and HCL. This aerosols in both dry and aqueous phase evaporate faster than the corresponding ammonium nitrate ($NH_4NO_3$) aerosols (Seinfeld and Pandis, 2012). Thus, the influence of mineral cations and organic species (scarce study over Asia) on the $NH_3$ gas–particle partitioning might be limited and will not have a large significant impact on the results of this study if we consider 15 % uncertainty. Hence, we agree with the reviewer neglecting mineral cations and organic acids may cause observed discrepancies in a minor way. Discussion is added in the revised manuscript.

**R1C4:**

**The reason is that in this model set-up, the $NH_3/NH_4^+$ partitioning is mainly controlled by sulfate and subsequently by nitrate, which might be in reality not the case in Asia. Consideration of at least the major mineral cations (e.g., $Na^+$, $K^+$, $Ca^{2+}$, $Mg^{2+}$) might be necessary, since all of them are ubiquitous and preferentially neutralize sulfate, which directly affects the $NH_3/NH_4^+$ partitioning. In contrast to the semi-volatile compound ammonium nitrate ($NH_4NO_3$), mineral cations form more stable compounds that exhibit a distinct different temperature dependent dissociation and water uptake, but no volatilization, as it is here the case only for $NH_4NO_3$. Thus, consideration of additional (mineral) cations could lead to more free ammonia (w.r.t. sulfate neutralization), which, in addition could lead to a larger fraction of ammonia being neutralized by nitric acid (e.g. resulting from lightning and thus adding up in the vertical model column as ammonium nitrate). And, since $NH_4NO_3$ is unstable at higher temperatures and low humidities, both cases could result in higher simulated $NH_3$ concentrations during the summer months resulting in potentially closer $NH_3$ total column concentrations w.r.t. IASI observations.**

- We would like to draw attention to one of our recent study (Acharja et al., 2020) based on analysis of water soluble inorganic chemical ions of $PM_1$, $PM_{2.5}$ and atmospheric trace gases over Indo-gangetic plain (IGP), South Asia which were monitored by Monitoring AeRosol and Gases in ambient Air (MARGA). The study revealed that $NH_4^+$ was one of the dominant ions, collectively with $Cl^-$, $NO_3^-$ and $SO_4^-$ constituted more than 95% of the measured ionic mass in both $PM_1$ and $PM_{2.5}$. Remaining ionic species (i.e., $Na^+$, $K^+$, $Ca^{2+}$, $Mg^{2+}$) formed constituted only about 3% of the total measured ions. Although major mineral cations (e.g., $Na^+$, $K^+$, $Ca^{2+}$, $Mg^{2+}$) contribute actively in neutralization reaction, but their concentration in IGP was found to be very low. Whereas over NCP, $NO_3^-$ and $SO_4^-$ were found to be dominant ions followed by $NH_4^+$ and $Cl^-$ which collectively contributed more than 86-90% in both $PM_1$ and $PM_{2.5}$. Other mineral cations contributed less than 5 % in both $PM_1$ and $PM_{2.5}$ (Dao et al., 2014). Furthermore, in one of the study, over East Asia, the neutralization capacities of major cations ($K^+$, $Ca^{2+}$, $Mg^{2+}$ and $NH_4^+$) were individually estimated by estimating the Neutralization Factors (NFs) for interpretations. It was found that $NH_4^+$ was the

predominant neutralizing cation with the highest NF (above 1), whereas $K^+$, $Ca^{2+}$ and $Mg^{2+}$ contributed relatively low in neutralization of aerosol acidity with lowest NF (below 0.2) (Xu et al., 2017). Hence, consideration of mineral cations may contribute in minor in neutralization of acidic aerosol over Asian region, still rigorous study is needed in future.

**R1C5:**

**Also, the underestimation of the surface $NH_3$ concentrations throughout the year over both South and East Asia could be a result of missing mineral cations in this model set-up. In reality, a larger fraction of sulphate might be neutralized by mineral cations rather than just by ammonium, which could lead to a larger fraction of free ammonia near the surface. Also, since both nitrates and sulfates preferentially react with mineral cations, nitric acid (e.g. from the traffic sector) might be neutralized by ammonia in a lower amount in reality, as it seems to be the case in this model set-up. In any case, consideration of mineral cations could also lead to a larger fraction of free ammonia near the surface, which might be even sufficient to explain discrepancies with surface observations.**

- Metzger et al., (2006) results have shown that only if (soluble) mineral components and (lumped) organic acids are accounted for, the observed gas-aerosol partitioning of ammonia and nitric acid can be accurately reproduced for air pollution episodes. Hence, while comparing model results with surface observations, incorporation of mineral cations may lead to increase in free ammonia near surface but change will be not be significant (considering 15 % increase) (Metzger et al., 2006).
- We have added the above description in the discussion section for explaining discrepancies with surface observations in the revised manuscript.

**R1C6:**

**Furthermore, due to the excess of ammonia in this model set-up, ammonium nitrate can be formed in both regions, although the simulated sulfate concentrations (burden) are higher in East Asia compared to South Asia. And, due to its semi-volatile character, the seasonal variability of $NH_4NO_3$ and the associated $NH_3$ concentrations differ in both regions as observed. Since $NH_4NO_3$ is unstable at higher temperatures, more $NH_3$ bound as $NH_4NO_3$ (compared to ammonium sulfate) can lead to higher $NH_3$ concentrations during summer, as it is observed in East Asia. In South Asia, where both ammonia and sulfate concentrations are lower, also $NH_4NO_3$ concentrations are lower and thus the seasonality of $NH_3$ is less pronounced, which is consistent with the surface observations.**

- Yes, we agree with reviewer.
- We have added this information in revised manuscript in the discussion section of 3.3 to explain seasonal discrepancies with surface observations over both South and East Asia. We agree with the reviewer, the seasonal variability of $NH_4NO_3$ is

strong during summer over East Asia as shown in the below Fig. 1 (figure S2 in the revised supplement), which can lead to higher $NH_3$ concentrations during summer over East Asia.

[Figure]

Figure 1. MOZART-4 model estimate of $NH_3NO_3$ wet deposition flux ($\times10^{-9}$ kg m$^{-2}$ s$^{-1}$) during summer (JJA) season (left) and during winter (DJF) season (right)

**R1C7:**

**On the other hand, the overestimation of the IASI total column $NH_3$ concentrations over South Asia, for most of the year except the summer months, could be also a result of missing anions, e.g., of organic acids, assuming the vertical exchange processes are more or less realistically modelled. However, considering mineral cations without additional acids, could likely cause even larger differences in this case (for details see e.g., https://acp.copernicus.org/articles/6/2549/2006/).**

- Yes, we agree with the reviewer only if the additional anions from (lumped) organic acids are taken into account in the EQSAM2 model, ammonium realistically partitions into the aerosol phase. This resulted the predicted average ammonium partitioning is comparable to the observations within 1–2% for both the aerosol fine and coarse mode (Metzger et al., 2006). As explained previously, we agree with the reviewer for accurate comparison, absence of major cations and organic acids (currently, study of field measurements of organic acids in Asia is least done) in this current set up may explain the overestimation of modeled $NH_3$ total columns for most of the year except summer months over South Asia.
- Explanation is added in revised manuscript in section 3.3.

**R1C8:**

**Unfortunately, these processes (briefly touched on above) are missing in most modelling studies, and I fear their consideration is also beyond the scope (or possibilities) of this study?**

- We thank you reviewer for putting this additional information to improve the understanding of $NH_3/NH_4$ gas-aerosol partitioning.
- Yes, since we followed parameterization of gas/aerosol partitioning by Metzger et al. (2002), unfortunately additional mineral cations and organic acids are missing in our modeling study which are important in gas-aerosol partitioning of reactive nitrogen. For accurate reproducing modeling results and real comparison to observations, EQMs play important role in determining $NH_3/NH_4^+$ gas-aerosol partitioning.
- But as mentioned previously, over Asia, chemical characterisation of water soluble inorganic chemical ions of $PM_1$, $PM_{2.5}$ and atmospheric trace gases reveals that concentration of major mineral cations is very low both in $PM_1$ and $PM_{2.5}$. Due to lack of study on the presence of organic acids over Asia limits our understanding. Hence, due to poor understanding of impact of organic species on aerosol (Zaveri et al., 2008), organic species are not considered in the thermodynamic calculations. However, EQMs are associated with considerable uncertainties and assumptions. According to Metzger et al. (2006) total ammonium $NH_4^+$ calculated ammonium-sulfate-nitrate-sodium-chloride-water system was about 15 % lower than that calculated by EQSAM2 (Equilibrium Simplified Aerosol Model) considering organic acids and the above study was based upon Greece, which might not be the similar case for Asian region. Thus, to study the influence of mineral cations and organic species on the $NH_3$ gas–particle partitioning need rigorous study over Asian region. Currently this new setup will be out of our scope, but in future work, we will try to use EQSAM2 to study the effect of additional mineral cations and organic acids on ammonium gas-aerosol partitioning.

**References**

1. Acharja, P., Ali, K., Trivedi, D. K., Safai, P. D., Ghude, S., Prabhakaran, T. and Rajeevan, M.: Characterization of atmospheric trace gases and water soluble inorganic chemical ions of PM1 and PM2.5 at Indira Gandhi International Airport, New Delhi during 2017–18 winter, Sci. Total Environ., 729, 138800, doi:10.1016/j.scitotenv.2020.138800, 2020.
2. Central Pollution Control Board (2020), [online] Available from: https://cpcb.nic.in/quality-assurance-quality-control/ (Accessed 26 May 2020), n.d.
3. Dao, X., Wang, Z., Lv, Y., Teng, E., Zhang, L. and Wang, C.: Chemical characteristics of water-soluble ions in particulate matter in three metropolitan areas in the North China Plain, PLoS One, 9(12), 1–16, doi:10.1371/journal.pone.0113831, 2014.
4. Metzger, S., Dentener, F., Pandis, S. and Lelieveld, J.: Gas/aerosol partitioning: 1. A computationally efficient model, J. Geophys. Res. Atmos., 107(16), doi:10.1029/2001JD001102, 2002.
5. Metzger, S., Mihalopoulos, N. and Lelieveld, J.: Importance of mineral cations and organics in gas-aerosol partitioning of reactive nitrogen compounds: Case study based on MINOS results, Atmos. Chem. Phys., 6(9), 2549–2567, doi:10.5194/acp-6-2549-

2006, 2006.

6. Seinfeld, J. H. and Pandis, S. N.: Atmospheric Chemistry and Physics: From Air Pollution to Climate Change, Wiley., 2012.

7. Seinfeld, J. H., Pandis, S. N. and Noone, K.: Atmospheric Chemistry and Physics: From Air Pollution to Climate Change, Phys. Today, 51(10), 88–90, doi:10.1063/1.882420, 1998.

8. Xu, J. S., He, J., Behera, S. N., Xu, H. H., Ji, D. S., Wang, C. J., Yu, H., Xiao, H., Jiang, Y. J., Qi, B. and Du, R. G.: Temporal and spatial variation in major ion chemistry and source identification of secondary inorganic aerosols in Northern Zhejiang Province, China, Chemosphere, 179(December 2014), 316–330, doi:10.1016/j.chemosphere.2017.03.119, 2017.

9. Zaveri, R. A., Easter, R. C., Fast, J. D. and Peters, L. K.: Model for Simulating Aerosol Interactions and Chemistry (MOSAIC), J. Geophys. Res. Atmos., 113(13), 1–29, doi:10.1029/2007JD008782, 2008.

---

## Editor Comment (EC1) · Frank Dentener (Editor) · 21 Dec 2020

Dear authors,

I have taken note of the rebuttal to the reviewer's comments.

Several (main) concerns have voiced with regard to the use and quality of surface observations for comparison, the use of coarse resolution model and low temporal resolution for comparison with IASI satellite data, and issues with the gas-particle partitioning.

Although the responses were to some extent addressing the reviewers concerns, I

encourage the authors to avoid relaying issues for 'future work', and where appropriate extend the analysis with some sensitivity studies.

Further discussion is warranted wrg to quality issues of surface observations: information on calibration procedures, and in particular for what it means for this study should be described carefully.

In particular, I would like to see a somewhat more in depth discussion on the potential biases derived from mismatch of temporal matching and boundary layer dynamics in the MOZART model, in particular in winter when high atmospheric stability prevents mixing, and IASI may not observe all NH3 close to the surface. A case study with higher temporal (and spatial) resolution for a limited and more frequent output and realistic assumptions on IASI effective kernels may be helpful to illustrate the sensitivity of results.

Likewise some first order estimate of the impact of applying a temporal profile on agricultural NH3 emission would be preferable.

I encourage the author to resubmit, taken the review comments and my instructions as much as possible into account.

---

## Author Comment (AC4) · 9 Feb 2021

**Response to Anonymous Referee #1's Comments**

First of all we thank the reviewer for his valuable suggestion on our study and sincerely appreciate the reviewer's insightful and helpful comments.

Below we explicitly respond to each of the items raised in the comments by anonymous referee #1. These comments are indicated in **bold**, whereas the author's response is presented in blue and revisions in red in revised manuscript.

**R1C1:**

**The authors present an analysis of atmospheric ammonia over South and East Asia based on the MOZART-4 model that is driven by the HTAP-v2 emission inventory. Model results are compared against IASI satellite observations (total column), as well as surface observations of CPCB (India) and NNDMN (China) for the year 2010. This topic is very important, since ammonia partitions into the only ubiquitous volatile cation, i.e., ammonium ($NH_4^+$). $NH_4^+$ plays a crucial role in air quality and visibility due to its volatility and ability to neutralize acidic air pollutants, which are often of anthropogenic origin. And despite the various air pollution abatement efforts, ammonia concentrations are increasing in many regions of the world and are thus still of concern, not only in Asia. Despite some fundamental weakness in the modelling approach (which is unfortunately common to most such modeling studies and therefore is not a reason for rejection), this study is overall sufficiently sound. I would therefore recommend publication, if the authors take the following comments and discussion points into account.**

> We thank the reviewer for carefully reading the manuscript. We agree that the suggested discussion will improve the quality of the manuscript.

**R1C2:**

**The study reveals that spatial differences (total column) between MOZART-4 and IASI are generally largest during local autumn/winter season, with an overestimation compared to IASI observations. This overestimation is most pronounced for IGP South Asia (20°N-32°N, 70°E-95°E), while rather an underestimation is found for NCP East Asia (30°N-40°N, 110°E-120°E), especially during the summer months. On the other hand, the comparison of surface concentrations reveals that the model underestimates the ammonia observations over South and East Asia throughout the year. This is shown by monthly mean (time series) and annual averages (scatter plot), and these results are in contrast to the total column case (model burden w.r.t. IASI observations).**

> Yes, we agree with the reviewer's observations that the difference between is most pronounced for IGP South Asia. The following Fig. 1 below shows the time-height distribution of $NH_3$ and mean planetary boundary layer height (PBLH) averaged over the IGP region, respectively. It can be seen that during winter months, higher

atmospheric stability prevents the mixing of boundary layer $NH_3$ to the free troposphere over IGP, which is reflected in the higher winter-time values of MOZART-4 $NH_3$ columns. Similarly, a higher $NH_3/NH_4^+$ ratio (Fig. S3 in the revised Supplement) and lower dry and wet deposition (Fig. S4 and S5 in the revised Supplement) of $NH_3$ over IGP in winter months enhances the accumulation of $NH_3$ in the boundary layer compared to summer months. On the other hand, very little $NH_3$ gets detected by the satellite at the higher altitudes, where the satellite's detection sensitivity is more than that at the surface (Clarisse et al., 2010). IASI measurements' limited sensitivity to detect boundary layer $NH_3$ (Van Damme et al., 2014) could be one of the reasons for large differences between MOZART-4 and IASI in winter seasons. Also, the wheat crop sowing over IGP involves a higher fertilizer application rate during peak winter months that releases a significant quantity of $NH_3$ into the atmosphere. However, this seasonality is largely missing in the emissions (Fig. 2 (top, left in the revised manuscript)), indicating that the winter-time meteorology largely drives higher MOZART-4 $NH_3$ over this region.

Also, the emission fluxes of $SO_2$ and $NO_x$ over IGP are only one-fourth of that over NCP (Wang et al., 2020). Therefore, relatively low $SO_2$ and $NO_x$ concentration could be an important factor for Higher $NH_3$ columns over IGP during winter.

[Figure]

Figure 1: Daily vertical distribution of distribution of $NH_3$ (ppb) averaged over IGP South Asia (20°N-32°N, 70°E-95°E) (left) and daily mean Planetary Boundary Layer height (PBLH in meters) averaged over IGP South Asia (20°N-32°N, 70°E-95°E) (right)

**Despite some potential calibration issue w.r.t. certain observations, there seems to be no obvious inconsistency with the $NH_3$ observations used in this study. Instead, both issues (model vs surface and total column observations) rather point to an incomplete model set-up w.r.t. the gas-aerosol partitioning assumptions. Nevertheless, I also recommend**

**that the authors make sure that the study is based on (or includes) quality controlled surface observations.**

The quality control and assurance method followed by CPCB for these air quality monitoring stations is given at CPCB (2011 and 2020). The calibration procedures for the $NH_3$ analyzer conforms to USEPA (the United States Environmental Protection Agency) methodologies and include daily calibration checks, biweekly precision checks, and linearity checks every six weeks. All analyzers undergo full calibration every six weeks. For detail on calibration procedure, refer to CPCB (2020); Technical Specifications for Continuous Real Time Ambient Air Quality Monitoring Analysers (2016). Furthermore, we take the following steps to reassure the quality of $NH_3$ observations from the CPCB network stations. For data quality, we rejected all the observation values below 1 µg m$^{-3}$ and above 250 µg m$^{-3}$ at a given site if other sites in the network do not show values outside this range. This step aims to eliminate any short-term local influence that cannot be captured in the models and retain the regional-scale variability. Second, we removed single peaks characterized by a change of more than 100 µg m$^{-3}$ in just one hour for all the data in CPCB monitoring stations. This step filters random fluctuations in the observations. Third, we removed some very high $NH_3$ values that appeared in the time series right after the missing values. For any given day, we removed the sites from the consideration that either experience instrument malfunction and/or appear to be very heavily influenced by strong local sources. This information is updated in the revised manuscript.

In order to verify the data quality of the CBCB monitoring site, we have inter compared the $NH_3$ measurement at CPCB monitoring station (R.K. Puram) in Delhi with the $NH_3$ measurements at Indira Gandhi International (IGI) Airport taken during the Winter Fog Experiment (WiFEX) (Ghude et al., 2017) using MARGA (Measurement of Aerosols and Gases) instrument during the winter season of 2017-2018. More details on the $NH_3$ measurements using MARGA is available with Acharja et al. (2020). Both sites were situated in the same area of Delhi (less than 1km). Our inter-comparison shows that $NH_3$ measured at CPCB monitoring station by chemiluminescence method are slightly (on an average 9.8 µg m$^{-3}$) on the higher side than $NH_3$ measured by ion chromatography (IC) using MARGA (Fig. 2 below). The observed differences could partly be related to the different $NH_3$ measurement techniques and partly to the locations of the two monitoring sites that were not place exactly at the same location. Apparently, the difference of 9.8 µg m$^{-3}$ indicates that the $NH_3$ measurements from the CPCB do not suffer from the calibration issue. However, rigorous validation is required in the future with more data sets.

In the revised manuscript we have now included above discussion.

[Figure]

Figure 2. Comparison of $NH_3$ (µg m$^{-3}$) concentration from MARGA instrument with RK Puram (CPCB) station

**R1C3:**

**Regarding the modeling assumptions, it should be noted that the chosen set-up has its limitations w.r.t. the $NH_3/NH_4^+$ partitioning. The main issue here is that in the current set-up, both (i) cations other than $NH_4^+$, e.g., sodium ($Na^+$), potassium ($K^+$), calcium ($Ca^{2+}$), and magnesium ($Mg^{2+}$), have been neglected, as well as (ii) organic acids were omitted for the gas-aerosol partitioning calculations. Both are, however, important for the $NH_3/NH_4^+$ partitioning w.r.t. to real world observations. Nevertheless, since mineral cations and organic acids have been neglected in conjunction, the presented model results could be in terms of yearly averages more or less "right" for the wrong reason, as indicated by a study published sometimes ago in ACP 2006 (https://acp.copernicus.org/articles/6/2549/2006/). On shorter time scales, however, the incomplete model set-up could be a cause of the observed discrepancies.**

> Yes, we agree with the reviewer's comment that the current modeling setup has limitations w.r.t. $NH_3/NH_4^+$. In this present work, the ammonium nitrate distribution is determined from $NH_3$ emissions based on the parameterization of

gas/aerosol partitioning by Metzger et al. (2002), which is a set of approximations to the equilibrium constant calculation (Seinfeld et al., 1998), based on the level of sulfate present. The application of any equilibrium models (EQMs) in global atmospheric studies is associated with considerable uncertainties. We followed, equilibrium simplified aerosol model (EQSAM)- gas-aerosol partitioning calculations by Metzger et al. (2002) the current setup. The assumptions used in this study are limited to the ammonium-sulfate-nitrate-water system, which is valid for only inorganic salt compounds. This latter was updated (EQSAM2) to account for ammonium-sulfate-nitrate-sodium-chloride-water system, mineral citation, and organic acids (Metzger et al., 2006). Metzger et al. (2006) found that the total ammonium partitioning calculated by updated-EQSAM2 parameterization was 15 % lower than that calculated from the parameterization similar to Metzger et al., (2002). Ammonia has a stronger affinity towards the neutralization of sulphuric acid ($H_2SO_4$) than nitric acid ($HNO_3$), whereas the formation of ammonium chloride (NH4Cl(s) or (aq)) in the atmosphere is unstable and can dissociate reversibly to $NH_3$ and HCL. These aerosols in both dry and aqueous phase evaporate faster than the corresponding ammonium nitrate ($NH_4NO_3$) aerosols (Seinfeld and Pandis, 2012).

We agree with the reviewer's comment that neglecting sodium-chloride organic acids and mineral cations in the parameterization of gas/aerosol partitioning system in the present work may cause some observed discrepancies. Overall consideration of major mineral cations could lead to more free ammonia, which will potentially increase $NH_3$ total columns. This will further increase differences (total column) between MOZART-4 and IASI over IGP and a decrease in differences (total column) between MOZART-4 and IASI over NCP. However, the influence of mineral cations on the $NH_3$ gas–particle partitioning might be limited (Acharja et al., 2020; Dao et al., 2014) and requires further focused studies over south Asia. Discussion is added in the revised manuscript.

**R1C4:**

**The reason is that in this model set-up, the $NH_3/NH_4^+$ partitioning is mainly controlled by sulfate and subsequently by nitrate, which might be in reality not the case in Asia. Consideration of at least the major mineral cations (e.g., $Na^+$, $K^+$, $Ca^{2+}$, $Mg^{2+}$) might be necessary, since all of them are ubiquitous and preferentially neutralize sulfate, which directly affects the $NH_3/NH_4^+$ partitioning. In contrast to the semi-volatile compound ammonium nitrate ($NH_4NO_3$), mineral cations form more stable compounds that exhibit a distinct different temperature dependent dissociation and water uptake, but no volatilization, as it is here the case only for $NH_4NO_3$. Thus, consideration of additional (mineral) cations could lead to more free ammonia (w.r.t. sulfate neutralization), which, in addition could lead to a larger fraction of ammonia being neutralized by nitric acid (e.g. resulting from lightning and thus adding up in the vertical model column as ammonium nitrate). And, since $NH_4NO_3$ is unstable at higher temperatures and low**

**humidities, both cases could result in higher simulated NH₃ concentrations during the summer months resulting in potentially closer NH₃ total column concentrations w.r.t. IASI observations.**

Yes, we agree with the reviewer's comment that in our modeling system, the $NH_3/NH_4^+$ partitioning is mainly controlled by sulfate and subsequently by nitrate since major mineral cations (e.g., $Na^+$, $K^+$, $Ca^{2+}$, $Mg^{2+}$) are not included. We would like to draw attention to one of our recent study (Acharja et al., 2020) based on analysis of water-soluble inorganic chemical ions of $PM_1$, $PM_{2.5}$, and atmospheric trace gases over Indo-Gangetic plain (IGP), South Asia, which were monitored by Monitoring AeRosol and Gases in Ambient Air (MARGA). The study revealed that $NH_4^+$ was one of the dominant ions, collectively with $Cl^-$, $NO_3^-$ and $SO_4^-$ constituted more than 95 % of the measured ionic mass in both $PM_1$ and $PM_{2.5}$. The remaining ionic species (e.g., $Na^+$, $K^+$, $Ca^{2+}$, $Mg^{2+}$) formed constituted only about 3 % of the total measured ions. Although major mineral cations (e.g., $Na^+$, $K^+$, $Ca^{2+}$, $Mg^{2+}$) contribute actively in a neutralization reaction, but their concentration in IGP was found to be very low. Whereas over NCP, $NO_3^-$ and $SO_4^-$ were found to be dominant ions followed by $NH_4^+$ and $Cl^-$ which collectively contributed more than 86-90 % in both $PM_1$ and $PM_{2.5}$. Other mineral cations contributed less than 5 % in $PM_1$ and $PM_{2.5}$ (Dao et al., 2014). Furthermore, in one of the studies, over East Asia, the neutralization capacities of major cations (e.g., $K^+$, $Ca^{2+}$, $Mg^{2+}$ and $NH_4^+$) were individually estimated by estimating the Neutralization Factors (NFs) for interpretations. It was found that $NH_4^+$ was the predominant neutralizing cation with the highest NF (above 1), whereas $K^+$, $Ca^{2+}$ and $Mg^{2+}$ contributed relatively low in the neutralization of aerosol acidity with the lowest NF (below 0.2) (Xu et al., 2017). Hence, consideration of mineral cations may contribute in minor in the neutralization of acidic aerosol over Asian region; still, a rigorous study is needed in future. Therefore, consideration of mineral cations and organic acids on the $NH_3/NH_4^+$ partitioning might be limited and will not significantly impact the results of this study.

**R1C5:**

**Also, the underestimation of the surface NH₃ concentrations throughout the year over both South and East Asia could be a result of missing mineral cations in this model set-up. In reality, a larger fraction of sulphate might be neutralized by mineral cations rather than just by ammonium, which could lead to a larger fraction of free ammonia near the surface. Also, since both nitrates and sulfates preferentially react with mineral cations, nitric acid (e.g. from the traffic sector) might be neutralized by ammonia in a lower amount in reality, as it seems to be the case in this model set-up. In any case, consideration of mineral cations could also lead to a larger fraction of free ammonia near the surface, which might be even sufficient to explain discrepancies with surface observations.**

Metzger et al. (2006) results have shown that only if (soluble) mineral components and (lumped) organic acids are accounted for, the observed gas-aerosol partitioning of

ammonia and nitric acid can be accurately reproduced for air pollution episodes. Hence, while comparing model results with surface observations, incorporating mineral cations may lead to an increase in free ammonia near the surface, but change will not be significant (considering a 15 % increase) (Metzger et al., 2006). We have added the above description in the discussion section for explaining discrepancies with surface observations in the revised manuscript.

**R1C6:**

**Furthermore, due to the excess of ammonia in this model set-up, ammonium nitrate can be formed in both regions, although the simulated sulfate concentrations (burden) are higher in East Asia compared to South Asia. And, due to its semi-volatile character, the seasonal variability of NH4NO3 and the associated NH3 concentrations differ in both regions as observed. Since NH4NO3 is unstable at higher temperatures, more NH3 bound as NH4NO3 (compared to ammonium sulfate) can lead to higher NH3 concentrations during summer, as it is observed in East Asia. In South Asia, where both ammonia and sulfate concentrations are lower, also NH4NO3 concentrations are lower and thus the seasonality of NH3 is less pronounced, which is consistent with the surface observations.. In South Asia, where both ammonia and sulfate concentrations are lower, also $NH_4NO_3$ concentrations are lower and thus the seasonality of $NH_3$ is less pronounced, which is consistent with the surface observations.**

Yes, we agree with reviewer.
We have added this information in the revised manuscript. We agree with the reviewer, the seasonal variability of $NH_4NO_3$ is strong during summer over East Asia as shown in the below Fig. 3 (Fig. S6 in the revised Supplement), which can lead to higher $NH_3$ concentrations during summer over East Asia.

[Figure]

Figure 3. MOZART-4 model estimate of $NH_3NO_3$ wet deposition flux ($\times 10^{-9}$ kg m$^{-2}$ s$^{-1}$) during summer (JJA) season (left) and during winter (DJF) season (right)

**R1C7:**

**On the other hand, the overestimation of the IASI total column NH₃ concentrations over South Asia, for most of the year except the summer months, could be also a result of missing anions, e.g., of organic acids, assuming the vertical exchange processes are more or less realistically modelled. However, considering mineral cations without additional acids, could likely cause even larger differences in this case (for details see e.g., https://acp.copernicus.org/articles/6/2549/2006/).**

Yes, we agree with the reviewer's comment that the current modeling setup has a limitation with respect to mission anions of organic acids and could be one of the regions between observed and model discrepancies. We want to bring to reviewers' notice that model NH₃ total column concentrations are larger than IASI total column over South Asia (Fig. 4 in the revised manuscript) during most of the years, except during summer months where IASI total columns are larger than the model (Fig. 8 in the revised manuscript). It can be seen in above Fig. 1 (reply to R1C2) that during winter month's higher atmospheric stability prevents the mixing of boundary layer NH₃ to the free troposphere over IGP, which is reflected in the higher wintertime values of MOZART-4 NH₃ columns. Similarly, higher $NH_3/NH_4^+$ ratio and lower dry and wet deposition of NH₃ over IGP in winter month enhances the accumulation of NH₃ in the boundary layer compared Limited sensitivity of IASI measurements to detect boundary layer NH₃ (Van Damme et al., 2014) could be one of the reasons for large differences between MOZART-4 and IASI in winter seasons. On the other hand, heating of the landmass due to large solar incidence suppresses the wintertime subsidence over the IGP and leads to a deeper boundary layer during spring and early summer (the average PBLH is about 1100 m, 600 m deeper during spring and summer compared to winter over IGP). During this season, significant transport of the boundary pollution in the mid and upper troposphere due to enhanced convective activities and large scale vertical motion can be noticed. Vertical motion associated with the convective activities is expected to redistribute the NH₃ concentration in the column, leading to more NH₃ at the higher altitudes where the satellite's detection sensitivity is more than that of the surface (Clarisse et al., 2010). As a result, more NH₃ gets detected by the satellite, and we see less difference between observations and model over the IGP.

Therefore, the addition of missing anions will further cause a larger difference between Model and IASI total column NH₃ concentrations over South Asia during most of the months, except during summer where the difference between Model and IASI total NH₃ column will decrease.

**R1C8:**

**Unfortunately, these processes (briefly touched on above) are missing in most modelling studies, and I fear their consideration is also beyond the scope (or possibilities) of this study?**

We thank the reviewer for putting this additional information to improve the understanding of $NH_3/NH_4^+$ gas-aerosol partitioning. Yes, since we followed parameterization of gas/aerosol partitioning by Metzger et al. (2002), unfortunately, additional mineral cations and organic acids are missing in our modeling study, which is important in gas-aerosol partitioning of reactive nitrogen. For accurate reproducing modeling results and real comparison to observations, EQMs play an important role in determining $NH_3/NH_4^+$ gas-aerosol partitioning.

As mentioned previously, over Asia, chemical characterization of water-soluble inorganic chemical ions of $PM_1$, $PM_{2.5}$ and atmospheric trace gases reveals that major mineral cations' concentration is very low in $PM_1$ and $PM_{2.5}$. Lack of study on the presence of organic acids over Asia limits our understanding. Hence, due to a poor understanding of the impact of organic species on aerosol (Zaveri et al., 2008), organic species are not considered in the thermodynamic calculations. However, EQMs are associated with considerable uncertainties and assumptions. According to Metzger et al. (2006), the total $NH_3/NH_4^+$ gas-aerosol partitioning calculated ammonium-sulfate-nitrate-sodium-chloride-water system was about 15 % lower than that calculated by EQSAM2 (Equilibrium Simplified Aerosol Model) considering organic acids, and the above study was based upon Greece, which might not be the similar case for the Asian region. Thus, to study the influence of mineral cations and organic species on the $NH_3$ gas-particle partitioning need rigorous study over the Asian region. Currently, this new setup will be out of our scope. However, in future work, we will try to use EQSAM2 to study the effect of additional mineral cations and organic acids on ammonium gas-aerosol partitioning.

**References**

Acharja, P., Ali, K., Trivedi, D. K., Safai, P. D., Ghude, S., Prabhakaran, T. and Rajeevan, M.: Characterization of atmospheric trace gases and water soluble inorganic chemical ions of PM1 and PM2.5 at Indira Gandhi International Airport, New Delhi during 2017–18 winter, Sci. Total Environ., 729, 138800, doi:10.1016/j.scitotenv.2020.138800, 2020.

Clarisse, L., Shephard, M. W., Dentener, F., Hurtmans, D., Cady-Pereira, K., Karagulian, F., Van Damme, M., Clerbaux, C. and Coheur, P. F.: Satellite monitoring of ammonia: A case study of the San Joaquin Valley, J. Geophys. Res. Atmos., 115(13), 1–15, doi:10.1029/2009JD013291, 2010.

CPCB: Guidelines for Real Time Sampling & Analyses., 2011.

CPCB: Central Pollution Control Board (2020), [online] Available from: https://cpcb.nic.in/quality-assurance-quality-control/.

Van Damme, Wichink Kruit, R. J., Schaap, M., Clarisse, L., Clerbaux, C., Coheur, P. F., Dammers, E., Dolman, A. J. and Erisman, J. W.: Evaluating 4 years of atmospheric ammonia (NH3) over Europe using IASI satellite observations and LOTOS-EUROS model results, J. Geophys. Res., 119(15), 9549–9566, doi:10.1002/2014JD021911, 2014.

Dao, X., Wang, Z., Lv, Y., Teng, E., Zhang, L. and Wang, C.: Chemical characteristics of water-soluble ions in particulate matter in three metropolitan areas in the North China Plain, PLoS One, 9(12), 1–16, doi:10.1371/journal.pone.0113831, 2014.

Ghude, S. D., Bhat, G. S., Prabhakaran, T., Jenamani, R. K., Chate, D. M., Safai, P. D., Karipot, A. K., Konwar, M., Pithani, P., Sinha, V., Rao, P. S. P., Dixit, S. A., Tiwari, S., Todekar, K., Varpe, S., Srivastava, A. K., Bisht, D. S., Murugavel, P., Ali, K., Mina, U., Dharua, M., Rao, Y. J., Padmakumari, B., Hazra, A., Nigam, N., Shende, U., Lal, D. M., Chandra, B. P., Mishra, A. K., Kumar, A., Hakkim, H., Pawar, H., Acharja, P., Kulkarni, R., Subharthi, C., Balaji, B., Varghese, M., Bera, S. and Rajeevan, M.: Winter fog experiment over the Indo-Gangetic plains of India, Curr. Sci., 112(4), doi:10.18520/cs/v112/i04/767-784, 2017.

Metzger, S., Dentener, F., Pandis, S. and Lelieveld, J.: Gas/aerosol partitioning: 1. A computationally efficient model, J. Geophys. Res. Atmos., 107(16), doi:10.1029/2001JD001102, 2002.

Metzger, S., Mihalopoulos, N. and Lelieveld, J.: Importance of mineral cations and organics in gas-aerosol partitioning of reactive nitrogen compounds: Case study based on MINOS results, Atmos. Chem. Phys., 6(9), 2549–2567, doi:10.5194/acp-6-2549-2006, 2006.

Seinfeld, J. H. and Pandis, S. N.: Atmospheric Chemistry and Physics: From Air Pollution to Climate Change, Wiley., 2012.

Seinfeld, J. H., Pandis, S. N. and Noone, K.: Atmospheric Chemistry and Physics: From Air Pollution to Climate Change , Phys. Today, 51(10), 88–90, doi:10.1063/1.882420, 1998.

Technical Specifications for Continuous Real Time Ambient Air Quality Monitoring Analysers: Technical Specifications for Continuous Real Time Ambient Air Quality Monitoring Analysers / Station Volume – II., 2016.

Wang, T., Song, Y., Xu, Z., Liu, M., Xu, T., Liao, W., Yin, L., Cai, X., Kang, L., Zhang, H. and Zhu, T.: Why is the Indo-Gangetic Plain the region with the largest NH3column in the globe during pre-monsoon and monsoon seasons?, Atmos. Chem. Phys., 20(14), 8727–8736, doi:10.5194/acp-20-8727-2020, 2020.

Xu, J. S., He, J., Behera, S. N., Xu, H. H., Ji, D. S., Wang, C. J., Yu, H., Xiao, H., Jiang, Y. J., Qi, B. and Du, R. G.: Temporal and spatial variation in major ion chemistry and source identification of secondary inorganic aerosols in Northern Zhejiang Province, China, Chemosphere, 179(December 2014), 316–330, doi:10.1016/j.chemosphere.2017.03.119, 2017.

Zaveri, R. A., Easter, R. C., Fast, J. D. and Peters, L. K.: Model for Simulating Aerosol Interactions and Chemistry (MOSAIC), J. Geophys. Res. Atmos., 113(13), 1–29, doi:10.1029/2007JD008782, 2008.

---

## Author Comment (AC5) · 9 Feb 2021

**Response to Anonymous Referee #2's Comments**

First of all we thank the reviewer for the positive evaluation of our study and sincerely appreciate the reviewer's insightful and helpful comments.

Below we explicitly respond to each of the items raised in the comments of anonymous referee #2. These comments are indicated in **bold**, whereas the author's response is presented in blue and revisions in red in revised manuscript.

**R2C1:**

**Ammonia is an important short-lived pollutant with a huge global relevance for air quality, biodiversity and climate due to the wide spread food production. Improving the nitrogen use efficiency in agriculture is of key importance, which requires an understanding of the nitrogen budgets and the ability to monitor these. The atmospheric ammonia burden is difficult to model, and hence, improving our modelling capacity is an important activity. After reading the paper in detail I recommend a major revision is required to improve the paper to a level which is beyond a simple comparison between a coarse model field and observations, which is currently basically is.**

**A major drawback of this study is the coarse resolution the modelling is performed on. Not only in a spatial sense, also the output is available on 4 hours of the day, with IASI overpass (9:30) right in between the output times (06 and 12). The description of the comparison to the satellite data is very short. Giving the strong diurnal cycle of ammonia and the fact that the satellite data availability is affected by all kinds of factors I would like to see a much more detailed description on the method and the impacts of the choices made. - Were the monthly mean comparisons made by averaging paired observations across the month? How many valid pairs were required to allow for a valid number? If pairing was not done than a motivation/discussion why this is not important should be included. Normally the large degree of variability of ammonia column densities between days requires to pair. Satellite data availability and patterns in these within a large grid cell can also impact a non-paired comparison. How was the modelled column for 09:30 estimated? Later I read that a daily mean model value is used. . . correct? - Which quality flags of the satellite data were used? - In our experience the diurnal emission cycle largely impacts the ammonia columns at overpass. What was assumed in this study?**

> Yes, we agree that we compared the monthly mean $NH_3$ total column from the IASI overpass in the morning (9:30) with the monthly mean model $NH_3$ total column averaging all 4 time-steps of the day. We also agree that the diurnal emission cycle largely impacts the ammonia columns at the overpass. To check the impacts of the diurnal cycle (driven by Boundary layer dynamics), we have again compared the monthly mean $NH_3$ total column from the IASI overpass in the morning (9:30) with the monthly mean model output at 11:30 LT, which near to IASI overpass (Figure 1 below). If we compare satellite and model at the nearest time-step, the Normalised Mean Bias (NMB) over IGP is reducing by 6% (with daily mean NMB=42 % and with near-time step NMB=36 %, Fig 2 (left) below), and over NCP it is increasing by

6 % (with daily mean NMB= -20 % and with near-time step NMB= -26 %) (Figure 2 (right) below). Since our model was run with the flat diurnal emissions, we have not seen any significant change compared to 4 time-step mean columns and is one of the sources of uncertainties. In the revised version, we have now compared the monthly mean NH$_3$ total column from the IASI overpass in the morning (9:30) with the monthly mean model output at 11:30 LT near the IASI overpass.

[Figure]

Figure 1 (a) Scatter plot between annual averaged IASI and MOZART-4 (11:30 am) simulated NH$_3$ ($\times 10^{16}$ molecules cm$^{-2}$) total columns over IGP, South Asia (rectangle: 20°N-32°N, 70°E-95°E) and (b) Scatter plot between annual averaged IASI and MOZART-4 (11:30 am) simulated NH$^3$ ($\times 10^{16}$ molecules cm$^{-2}$) total columns over NCP, East Asia (rectangle: 30°N-40°N, 110°E-120°E).

[Figure]

**Figure 2 (left)** Comparison between monthly averaged IASI and MOZART-4 simulated NH$_3$ ($\times 10^{16}$ molecules cm$^{-2}$) total columns over IGP South Asia (20°N-32°N, 70°E-95°E) for daily mean (red) and near to satellite overpass (11:30, green), **(right)** Comparison between monthly averaged IASI and MOZART-4 simulated NH$_3$ ($\times 10^{16}$ molecules cm$^{-2}$) total columns over NCP East Asia (30°N-40°N, 110°E-120°E) for daily mean (red) and near to satellite overpass (11:30, green)

We agree with the reviewer's comment that a large degree of variability of ammonia column densities between days requires to pair, and Satellite data availability and patterns within a large grid cell can also impact a non-paired comparison. In the present study, we are looking at monthly, seasonal and annual data. Therefore, we considered that IASI provides representative monthly, seasonal, and annual means, despite possible biases introduced by lack of days of data due to cloud cover. We compared monthly mean column to column, as the IASI retrieval algorithm only provides total columns. We made unweighted average distributions using all the morning IASI measurements available, following the recommendation for using the dataset provided in Van Damme et al. (2017).

However, as suggested by the reviewer in the revised manuscript, we have now compared the monthly mean columns by averaging paired observations across the months. We have considered the daily $NH_3$ cloud-free satellite total column data and compared it with the modeled daily $NH_3$ total column averaging paired observations across the months, seasons and year. For consistency with satellite retrievals, first, the model output (11:30 LT) at each day close to satellite overpass time (09:30 LT) is interpolated in space to the location of valid satellite retrievals. Since the IASI retrieval algorithm only provides total columns, in the second step, we made the unweighted average distribution of the daily paired data to obtain a monthly mean value of satellite and model total $NH_3$ columns at each model grid location. The following figures show the comparison between satellite and Model $NH_3$ columns on annual (Figure 3 and 4 below) and seasonal scale (Figure 5 and 6 below) calculated by averaging paired and non-paired observations. We find that the normalized mean bias (NMB) over IGP decreased to 38 % with pair-comparison than non-paired comparison (58 %) considering the model columns close to satellite overpass time. However, normalized mean bias (NMB) increased to -41 % with pair-comparison over the NCP region than non-paired comparison (-37 %).

[Figure]

**Figure 3: (top)** Comparison between annual mean satellite and Model $NH_3$ columns calculated by averaging paired observations, **(bottom)** Comparison between annual mean satellite and Model $NH_3$ columns calculated by averaging non-paired observations.

[Figure]

**Figure 4 (top)** Scatter plot between annual averaged IASI and MOZART-4 (11:30 am) simulated $NH_3$ total columns over IGP (left) and NCP (right) calculated by averaging **paired** observations, **(bottom)** Scatter plot between annual averaged IASI and MOZART-4 (11:30 am) simulated $NH_3$ total columns over IGP (left) and NCP (right) calculated by averaging **non-paired** observations

[Figure]

**Figure 3: (left)** Comparison between annual mean satellite and Model $NH_3$ columns calculated by averaging non-paired observations considering daily mean columns, **(right)** Comparison between annual mean satellite and Model $NH_3$ columns calculated by averaging paired observations close to satellite overpass time.

[Figure]

[Figure]

**Figure 6 (left)** Comparison between monthly averaged IASI (blue, non-paired) and MOZART-4 simulated $NH_3$ total columns for daily mean (red, non-paired) and monthly averaged IASI (paired, green) and MOZART-4 simulated $NH_3$ near to satellite overpass (11:30, black, paired) over IGP South Asia (20°N-32°N, 70°E-95°E), **(right)** Comparison between monthly averaged IASI (blue, non-paired) and MOZART-4 simulated $NH_3$ total columns for daily mean (red, non-paired) and monthly averaged IASI (paired, green) and MOZART-4 simulated $NH_3$ near to satellite overpass (11:30, black, paired) over NCP East Asia (30°N-40°N, 110°E-120°E).

Based on this new analysis, we have now included all the new plots (Fig. 4, Fig. 5, Fig. 6, Fig. 8) in the revised manuscript, modified Table 1, and added a detailed description of the satellite data comparison.

Further, to see the impact of finer resolution and more frequent output (1hr), we used simulated $NH_3$ concentration for 2011 using WRF-Chem simulation for the year 2011 from Ghude et al. (2016) over south Asia at 36 km grid spacing. The model uses MOZART-4 gas-phase chemistry linked to the GOCART aerosol scheme, similar to the one used in MOZART-4 simulation in the present work. Again, we have considered the daily $NH_3$ cloud-free satellite total column data for 2011 and compared it with the modeled daily $NH_3$ total column averaging paired observations across the year. For consistency with satellite retrievals, first, the model output (9:30 LT) at each day is interpolated in space to the location of valid satellite retrievals at an overpass time of 09:30 LT. Since the IASI retrieval algorithm only provides total columns, in the second step, we made the unweighted average distribution of the daily paired data to obtain a yearly mean value of satellite and model total $NH_3$ columns at each model grid location (36 km). The following figures show the comparison between satellite and Model $NH_3$ columns on annual (Figure 7 below) scale and its scatter (Figure 8) calculated by averaging paired and non-paired observations. It can be seen that compared to coarse simulations, the bias between the model and IASI $NH_3$ total columns are even larger with finer-scale simulations. We have included this for the reviewer's reference but not included in the revised manuscript. It gives a similar difference, but the magnitude of the difference is larger with WRF-Chem simulations.

[Figure]

**Figure 7:** Comparison between annual mean IASI (left) and WRF-Chem (Middle) NH$_3$ columns and their difference (Right) calculated by averaging paired observations at 09:30 am on 36 km grid resolution.

[Figure]

**Figure 8:** Scatter plot between annual averaged IASI and WRF-Chem (09:30 am) simulated NH$_3$ total columns over IGP.

**R2C2:**
**Given the agricultural practices in India, is it warranted to use a flat emission cycle across the year?**

We agree with the reviewer's comment. A more realistic seasonal cycle of ammonia emissions is needed for the simulations involving agriculture-based countries like India. The HTAP-V2 inventory certainly lacks this information. We aim to improve the inventory by including such a seasonal cycle for ammonia emissions in our future studies.

**R2C3:**
**The paper is severely hampered by the coarse comparison and I am afraid that the comparison methodology may impact the systematic differences seen in this paper. The differences between overpass time and a daily mean for instance relate to the daylength (variability) and associated mixing, diurnal emission cycle, frequency and kind of precipitation events, etc. I would have like to see an analysis/consideration of such factors in this paper. Part of the observations might be useful for this purpose.**

We request reviewer to refer to our reply to comment R2C1.

**R2C4:**

**The discussion does not include a comparison to other modelling studies evaluating ammonia levels across Asia or studies on ammonia life time.**

Very few studies were carried out in Asia similar to Clarisse et al. (2009), which have evaluated ammonia levels and compared model simulations with satellite retrievals. In a recent study, it is shown that higher summer-time temperature along with the higher Nitrogen (N) fertilizer application rate could cause high $NH_3$ emissions resulting in the high $NH_3$ columns over Asia, particularly during June-July-August (JJA) (Wang et al., 2020). However, satellite and model evaluation is mostly missing in this study. Studies discussing ammonia lifetime are already mentioned in the discussion part of the manuscript.

**R2C5:**

**I could identify many grammar mistakes in the english language use. The author list includes native speakers and I would like to urge to perform a careful language check.**

- A careful check for grammar has been done.

**Minor comments:**

**R2C6:**

**Abstract: Please use past tense for the method description**

- A careful check for grammar has been done.

**R2C7:**

**Introduction The introduction focusses mostly on the contribution of different agricultural activities to emission estimates in south and east Asia. The challenges with respect to the emission estimation, spatial and temporal emission variability, chemistry transport modelling and model-satellite comparison are not focused on although these are relevant to the paper and partly addressed. I would like to ask the authors to address these issues in the intro.**

As suggested by the reviewer, we have now added a paragraph to address the challenges for the emission estimation, spatial and temporal emission variability, and chemistry-transport modeling and model-satellite comparison in the introduction. We hope that it addresses the reviewer's concern.

**R2C8:**

**Line 43: chemical should be synthetic**

- We have replaced "chemical" with "synthetic".

**R2C9:**

**Line 50: 64 % of total means total global? if yes line 53 repeats this statement**

No, not globally, India and China together accounted for an estimated 64 % of the total amount of $NH_3$ emissions in Southern Asia during 2000-2014 (Xu et al., 2018). We have now corrected this in the revised manuscript.

**R2C10:**
**Line 60-62: could you use the recent edgar numbers or this from v4.3? Should these statements be presented with the global comparison the paragraph above?**

As suggested by the reviewer, we have now provided the estimates from EDGAR v4.3.2 and included this statement where the global comparison was discussed in the introduction section in the revised manuscript. Emission estimates provided by the latest EDGAR v4.3.2 emission inventory suggests that globally about 59 Tg of $NH_3$ was emitted in the atmosphere in 2012, out of which agricultural soils contributed about 56 %, manure management contributed about 19 %, and agricultural burning contributed about 1.5 % (Crippa et al., 2018).

**R2C11:**
**Line 63 and 67 are in direct contradiction to each other**

Yes, we agree with the reviewer's comment that lines 63 and 67 contradict each other. In India, around 50 % of total $NH_3$ emissions is estimated from the fertilizer application and the remaining from livestock and other $NH_3$ sources. Urea is mostly used as a fertilizer and alone contributes more than 90 % of the total fertilizer used for agricultural activities (Sharma et al., 2008). We have now corrected this in the revised manuscript.

**R2C12:**
**Data and methodology Line 85: this sentence implies only trace gases were modelled, which is not the case I guess**

We have now revised the sentence in the revised manuscript.

**R2C13:**
**Line 97: Does Mozart use a land use mosaic within a gridcell? Or dominant LUC? How do the wesely land use classes match those in the domain? Were the latter updated?**

Dry deposition of gases and aerosols were calculated online according to Wesely (1989) parameterization, and wet depositions of soluble gases were calculated as described by the method of Emmons et al. (2010). Land use cover (LUC) maps used in MOZART-4 are based on the Advanced Very High-Resolution Radiometer (AVHRR) and Moderate Resolution Imaging Spectroradiometer (MODIS) data based on NCAR Community Land Model (CLM) (Oleson et al., 2010). MOZART-4 represents the land surface as a hierarchy of sub-grid types: glacier, lake, wetland, urban and vegetated land. The vegetated land is further divided into a mosaic of Plant Function Type (PFTs). These same maps are used for the dry deposition calculations (Emmons et al., 2010; Lawrence and Chase, 2007; Oleson et al., 2010). We have now included this discussion in the revised manuscript.

**R2C14:**
**Line 122: didn't you use emissions of all sectors?**

- We have used all the sectors for emissions as per the HTAP v2 emission inventory. The sectors are for all substances defined as follows:

- Air = international and domestic air,
- Shipping = international shipping,
- Energy = power industry,
- Industry = manufacturing, mining, metal, cement, chemical, solvent industry, transport = ground transport (incl. road, rail, pipeline, inland waterways),
- Residential = heating/cooling of buildings and equipment/lighting of buildings and waste treatment.
- For $NH_3$ there is in addition sector agriculture = agriculture (but not agricultural waste burning).
- However, for $NH_3$ HTAP-v2 emission inventory covers only 5 (agriculture, energy, transport, residential and industry) sectors and rest two sectors, aircraft and international shipping, is not considered for $NH_3$ emissions.

**R2C15:**
**Line 133: cow dung is not fossil**
Removed word "cow dung" and replaced with "biomass combustion".

We have now corrected it in the revised manuscript.

**R2C16:**
**Results: Line 226: the methodology describes that nitrate is present – please explain**

We have corrected it in the revised manuscript.

**R2C17:**
**250: the model has no maximum emissions in summer as antrop is flat and soil is a few percent of total, so this statement seems incorrect**

We agree with the reviewer's comment that over South Asia, anthropogenic emissions are flat. Although soil emissions show some increase during summer, the percentage contribution to total emissions is small and will not affect observed $NH_3$ seasonal variability. We have now corrected it in the revised manuscript.

**R2C18:**
**Figure 2: the scale on the upper left figure is misleading. It seems a seasonal cycle where it is basically flat.**

As per reviewer's suggestion, we have now revised the scale to make it consistent.

**References**

Clarisse, L., Clerbaux, C., Dentener, F., Hurtmans, D. and Coheur, P. F.: Global ammonia distribution derived from infrared satellite observations, Nat. Geosci., 2(7), 479–483, doi:10.1038/ngeo551, 2009.

Crippa, M., Guizzardi, D., Muntean, M., Schaaf, E., Dentener, F., Van Aardenne, J. A., Monni, S., Doering, U., Olivier, J. G. J., Pagliari, V. and Janssens-Maenhout, G.: Gridded emissions of air pollutants for the period 1970-2012 within EDGAR v4.3.2, Earth Syst. Sci. Data, 10(4), 1987–2013, doi:10.5194/essd-10-1987-2018, 2018.

Van Damme, M., Whitburn, S., Clarisse, L., Clerbaux, C., Hurtmans, D. and Coheur, P.-F.: Version 2 of the IASI NH3 neural network retrieval algorithm; near-real time and reanalysed datasets, Atmos. Meas. Tech. Discuss., 1–14, doi:10.5194/amt-2017-239, 2017.

Emmons, L. K., Walters, S., Hess, P. G., Lamarque, J. F., Pfister, G. G., Fillmore, D., Granier, C., Guenther, A., Kinnison, D., Laepple, T., Orlando, J., Tie, X., Tyndall, G., Wiedinmyer, C., Baughcum, S. L. and Kloster, S.: Description and evaluation of the Model for Ozone and Related chemical Tracers, version 4 (MOZART-4), Geosci. Model Dev., 3(1), 43–67, doi:10.5194/gmd-3-43-2010, 2010.

Ghude, S. D.: Geophysical Research Letters, Geophys. Res. Lett., 1–8, doi:10.1002/2013GL058740, 2016.

Lawrence, P. J. and Chase, T. N.: Representing a new MODIS consistent land surface in the Community Land Model (CLM 3.0), J. Geophys. Res. Biogeosciences, 112(1), doi:10.1029/2006JG000168, 2007.

Oleson, K. W., Lawrence, D. M., Bonan, G. B., Flanner, M. G., Kluzek, E., Lawrence, P. J.,Zeng, X. (2010).: Technical Description of version 4.0 of the Community Land Model (CLM)., 2010.

Sharma, C., Tiwari, M. K. and Pathak, H.: Estimates of emission and deposition of reactive nitrogenous species for India, Curr. Sci., 94(11), 1439–1446, 2008.

Wang, T., Song, Y., Xu, Z., Liu, M., Xu, T., Liao, W., Yin, L., Cai, X., Kang, L., Zhang, H. and Zhu, T.: Why is the Indo-Gangetic Plain the region with the largest NH3column in the globe during pre-monsoon and monsoon seasons?, Atmos. Chem. Phys., 20(14), 8727–8736, doi:10.5194/acp-20-8727-2020, 2020.

Wesely, M. L.: Parameterization of surface resistances to gaseous dry deposition in regional-scale numerical models, Atmos. Environ., 23(6), 1293–1304, doi:10.1016/0004-6981(89)90153-4, 1989.

Xu, R. T., Pan, S. F., Chen, J., Chen, G. S., Yang, J., Dangal, S. R. S., Shepard, J. P. and Tian, H. Q.: Half-Century Ammonia Emissions From Agricultural Systems in Southern Asia: Magnitude, Spatiotemporal Patterns, and Implications for Human Health, GeoHealth, 2(1), 40–53, doi:10.1002/2017gh000098, 2018.

---

## Author Comment (AC6) · 9 Feb 2021

**Response to Editor Comments**

I have taken note of the rebuttal to the reviewer's comments. Several (main) concerns have voiced with regard to the use and quality of surface observations for comparison, the use of coarse resolution model and low temporal resolution for comparison with IASI satellite data, and issues with the gas-particle partitioning. Although the responses were to some extent addressing the reviewers concerns, I encourage the authors to avoid relaying issues for 'future work', and where appropriate extend the analysis with some sensitivity studies. Further discussion is warranted wrg to quality issues of surface observations: information on calibration procedures, and in particular for what it means for this study should be described carefully.

> **Reply:** We thank Editor for his comment and suggestions. We agree with both the reviewers' comments and suggestions and address their concerns that have greatly improved the manuscript's quality. We have used an additional set of simulations using the WRF-Chem model over South Asia with a consistent emission inventory and chemical scheme to see the impact of finer model resolution and high temporal resolution compared to the IASI satellite data.
>
> We have also added the following discussion on the data quality and the quality control procedure adopted in this study.
>
> The quality control and assurance method, followed by Central Pollution Control Board (CPCB) for these air quality monitoring stations, is given in the CPCB (2011 and 2020). Furthermore, we take the following steps to reassure the quality of $NH_3$ observations from the CPCB network stations. For data quality, we rejected all the observations values below the lowest detection limit of the instrument (1 μg m$^{-3}$) (Technical specifications for CAAQM station, 2019) because most of the sites are situated in the urban environment. For cities where more than one monitoring station is available, we rejected all the observations above 250 μg m$^{-3}$ at a given site if other sites in the network do not show values outside this range. This step aims to eliminate any short-term local influence that cannot be captured in the models and retain the regional-scale variability. Second, we removed single peaks characterized by a change of more than 100 μg m$^{-3}$ in just one hour for all the data in CPCB monitoring stations. This step filters random fluctuations in the observations. Third, we removed some very high $NH_3$ values that appeared in the timeseries right after the missing values. For any given day, we removed the sites from the consideration that either experience instrument malfunction, or appear to be very heavily influenced by strong local sources. In order to verify the data quality of CBCB monitoring site, we have inter compared the $NH_3$ measurement at CPCB monitoring station (R.K. Puram) in Delhi with the $NH_3$ measurements at Indira Gandhi International (IGI) Airport taken during Winter Fog Experiment (WiFEX) (Ghude et al., 2017) using Measurement of Aerosols and Gases (MARGA) instrument during winter season of 2017-2018. More details on the $NH_3$ measurements using MARGA is available with Acharja et al. (2020). Both sites were situated in the same area of Delhi (less than 1km). Our inter-comparison show that $NH_3$ measured at CPCB monitoring station by chemiluminescence method are slightly (on an average 9.8 μg m$^{-3}$) on higher side than $NH_3$ measured by ion chromatography (IC) using MARGA (Fig. S1 in the revised Supplement). The differences that were observed could partly be related to the different $NH_3$ measurement techniques and partly to the locations of the two

monitoring sites which were not place exactly at same location. Apparently, the difference of 9.8 µg m$^{-3}$ indicates that the NH$_3$ measurements from the CPCB do not suffer from the calibration issue.

In particular, I would like to see a somewhat more in depth discussion on the potential biases derived from mismatch of temporal matching and boundary layer dynamics in the MOZART model, in particular in winter when high atmospheric stability prevents mixing, and IASI may not observe all NH$_3$ close to the surface. A case study with higher temporal (and spatial) resolution for a limited and more frequent output and realistic assumptions on IASI effective kernels may be helpful to illustrate the sensitivity of results.

**Reply:** We have also discussed the potential biases derived from the mismatch of temporal matching and boundary dynamics in the MOZART mode. However, as suggested by the reviewer, in the revised manuscript, we have now compared the monthly mean columns by averaging paired observations across the months. We have considered the daily NH$_3$ cloud-free satellite total column data and compared it with the modeled daily NH$_3$ total column averaging paired observations across the months, seasons, and year. For consistency with satellite retrievals, first, the model output (11:30 LT) at each day close to satellite overpass time (09:30 LT) is interpolated in space to the location of valid satellite retrievals. Since the IASI retrieval algorithm only provides total columns, in the second step, we made the unweighted average distribution of the daily paired data to obtain a monthly mean value of satellite and model total NH$_3$ columns at each model grid location. We find that the normalized mean bias (NMB) over IGP decreased to 38% with pair-comparison than non-paired comparison (58%) considering the model columns close to satellite overpass time. However, normalized mean bias (NMB) increased to -41% with paired-comparison over the NCP region than non-paired comparison (-37%).

IASI retrieval method used for NH$_3$ does not produce averaging kernels as it is not based on optimal estimation. Therefore, IASI retrievals' limitation is that it does not allow the calculation of an averaging kernel to account for the vertical sensitivity of the instrument sounding to different layers in the atmosphere. We refer to Van Damme et al. (2017); Whitburn et al. (2016) for a comprehensive discussion on the advantages and disadvantage of constrained versus unconstrained retrieval approaches for NH$_3$. In brief, the current approach's main advantage is that a priori information does not influence the retrieval. Therefore, the NH$_3$ column value is derived from the measurement only. We compared column to column, as the IASI retrieval algorithm only provides total columns. We made unweighted average distributions using all the morning IASI measurements available, following the recommendation for using the dataset provided in Van Damme et al., (2017). In this paper, we have use NH$_3$ total columns retrieved from the IASI instrument morning overpass (AM) observations (i.e., 09:30 local time).

Further, in order to see the impact of finer resolution and more frequent output (1hr), we used simulated NH$_3$ concentration for the year 2011 using WRF-Chem simulation for the year 2011 from work reported in Ghude et al. (2016) over south Asia at 36 km grid spacing. The model uses MOZART-4 gas-phase chemistry linked to the GOCART aerosol scheme, similar to the one which is used in MOZART-4 simulation in the present work. Again, we have considered the daily NH$_3$ cloud-free satellite total column data for 2011 and compared it with the modeled daily NH$_3$ total column

averaging paired observations across the year. For consistency with satellite retrievals, first, the model output (9:30 LT) at each day is interpolated in space to the location of valid satellite retrievals at an overpass time of 09:30 LT. Since the IASI retrieval algorithm only provides total columns, in the second step, we made the unweighted average distribution of the daily paired data to obtain a yearly mean value of satellite and model total $NH_3$ columns at each model grid location (36 km). We found that the bias between the model and IASI $NH_3$ total columns is even larger with finer-scale simulations compared to coarse simulations. We have included this for the reviewer's reference but not included it in the revised manuscript as it gives a similar difference, but the magnitude of the difference is larger with WRF-Chem simulations.

We requested Editor to refer to our responses and figures provided in the 'Response to Anonymous Referee #1's and Anonymous Referee #2's Comments' document enclosed with the revised manuscript.

Likewise some first order estimate of the impact of applying a temporal profile on agricultural $NH_3$ emission would be preferable.

**Reply:** Unfortunately, the application of agriculture has significant spatial and temporal variability over South Asia, which depends on the cropping season and cropping pattern, is not well documented. However, we agree that it will contribute to the mismatch observed between observed and modeled $NH_3$ columns to some extent. Under the on-going South Asia Nitrogen Hub (SANH) project (The Global Challenges Research Fund (GCRF) South Asia Nitrogen hub), it is planned to develop a high-resolution $NH_3$ emission inventory over South Asia that will account for the temporal profile of agricultural $NH_3$ emission based on agricultural statistics.

I encourage the author to resubmit, taken the review comments and my instructions as much as possible into account.

**Reply:** We requested Editor to refer to our responses in the 'Response to Anonymous Referee #1's and Anonymous Referee #2's Comments' document enclosed with the revised manuscript.

**References**

Acharja, P., Ali, K., Trivedi, D. K., Safai, P. D., Ghude, S., Prabhakaran, T. and Rajeevan, M.: Characterization of atmospheric trace gases and water soluble inorganic chemical ions of PM1 and PM2.5 at Indira Gandhi International Airport, New Delhi during 2017–18 winter, Sci. Total Environ., 729, 138800, doi:10.1016/j.scitotenv.2020.138800, 2020.

CPCB: Guidelines for Real Time Sampling & Analyses., 2011.

CPCB: Central Pollution Control Board (2020), [online] Available from: https://cpcb.nic.in/quality-assurance-quality-control/.

Van Damme, M., Whitburn, S., Clarisse, L., Clerbaux, C., Hurtmans, D. and Coheur, P.-F.: Version 2 of the IASI $NH_3$ neural network retrieval algorithm; near-real time and reanalysed datasets, Atmos. Meas. Tech. Discuss., 1–14, doi:10.5194/amt-2017-239, 2017.

Ghude, S. D., Chate, D. M., Jena, C., Beig, G., Kumar, R., Barth, M. C., Pfister, G. G., Fadnavis, S. and Pithani, P.: Premature mortality in India due to PM $_{2.5}$ and ozone exposure, Geophys. Res. Lett., 43(9), 4650–4658, doi:10.1002/2016GL068949, 2016.

Ghude, S. D, D. M. Chate, C. Jena, G. Beig, R. Kumar, M. C. Barth,G.G. Pfister, S.

Fadnavis, and P. P.: Premature mortality in India due to PM2.5 and ozone exposure, Geophys. Res. Lett., 1–8, doi:10.1002/2013GL058740.Received, 2016.

Ghude, S. D., Bhat, G. S., Prabhakaran, T., Jenamani, R. K., Chate, D. M., Safai, P. D., Karipot, A. K., Konwar, M., Pithani, P., Sinha, V., Rao, P. S. P., Dixit, S. A., Tiwari, S., Todekar, K., Varpe, S., Srivastava, A. K., Bisht, D. S., Murugavel, P., Ali, K., Mina, U., Dharua, M., Rao, Y. J., Padmakumari, B., Hazra, A., Nigam, N., Shende, U., Lal, D. M., Chandra, B. P., Mishra, A. K., Kumar, A., Hakkim, H., Pawar, H., Acharja, P., Kulkarni, R., Subharthi, C., Balaji, B., Varghese, M., Bera, S. and Rajeevan, M.: Winter fog experiment over the Indo-Gangetic plains of India, Curr. Sci., 112(4), doi:10.18520/cs/v112/i04/767-784, 2017.

Technical specifications for CAAQM station: Technical specifications for continuous ambient air quality monitoring (CAAQM) STATION (real time) Central Pollution Control Board East Arjun Nagar , Shahdara., 2019.

The Global Challenges Research Fund (GCRF) South Asia Nitrogen hub: The Global Challenges Research Fund (GCRF) project, [online] Available from: https://gtr.ukri.org/projects?ref=NE/S009019/.

Whitburn, S., Damme, M. Van, Clarisse, L., Bauduin, S., Heald, C. L., Hurtmans, D., Zondlo, M. A., Clerbaux, C. and Coheur, P.: A flexible and robust neural network IASI-$NH_3$, , 6581–6599, doi:10.1002/2016JD024828.Received, 2016.

---

## Author Comment (AC3)

**Response to Anonymous Referee #2's Comments**

First of all we thank the reviewer for the positive evaluation of our study and sincerely appreciate the reviewer's insightful and helpful comments.

Below we explicitly respond to each of the items raised in the comments of anonymous referee #2. These comments are indicated in **bold**, whereas the author's response is presented in blue and revisions in red.

**R2C1:**

**Ammonia is an important short-lived pollutant with a huge global relevance for air quality, biodiversity and climate due to the wide spread food production. Improving the nitrogen use efficiency in agriculture is of key importance, which requires an understanding of the nitrogen budgets and the ability to monitor these. The atmospheric ammonia burden is difficult to model, and hence, improving our modelling capacity is an important activity. After reading the paper in detail I recommend a major revision is required to improve the paper to a level which is beyond a simple comparison between a coarse model field and observations, which is currently basically is.**

**A major drawback of this study is the coarse resolution the modelling is performed on. Not only in a spatial sense, also the output is available on 4 hours of the day, with IASI overpass (9:30) right in between the output times (06 and 12). The description of the comparison to the satellite data is very short. Giving the strong diurnal cycle of ammonia and the fact that the satellite data availability is affected by all kinds of factors I would like to see a much more detailed description on the method and the impacts of the choices made. - Were the monthly mean comparisons made by averaging paired observations across the month? How many valid pairs were required to allow for a valid number? If pairing was not done than a motivation/discussion why this is not important should be included. Normally the large degree of variability of ammonia column densities between days requires to pair. Satellite data availability and patterns in these within a large grid cell can also impact a non-paired comparison. - How was the modelled column for 09:30 estimated? Later I read that a daily mean model value is used. . . correct? - Which quality flags of the satellite data were used? - In our experience the diurnal emission cycle largely impacts the ammonia columns at overpass. What was assumed in this study?**

- In order to add description of comparison of the model to IASI satellite, we have qualitatively compared IASI NH$_3$ columns with modeled NH$_3$ columns since the IASI-NH$_3$ retrieval does not produce an averaging kernel to properly weight the model values (Clarisse et al., 2009). We thus compared column to column, as IASI retrieval algorithm only provides total columns. We made un-weighted average distributions using all the morning IASI measurement available, following the recommendation for the use of the dataset provided in Van Damme et al. (2017). Furthermore, to check the impacts of diurnal cycle, we have compared the annual mean satellite overpass (9:30 am) with the nearest model timestep (11:30 am) and we have not seen any significant changes compared to 4 timestep mean columns, which we has shown in the manuscript. If we compare satellite and model at the nearest timestep, the Normalised Mean Bias (NMB) over India is reducing by 6% and over China it is increasing by 6 % (Fig 1 (a) and (b)). Our focus of the study is only on monthly, seasonal and annual data, hence we consider that IASI provides representative monthly, seasonal and annual means, despite possible biases introduced by lacking days of data due to cloud cover.

[Figure]

- Figure 1. (a) Scatter plot between annual averaged IASI and MOZART-4 (11:30 am) simulated NH$_3$ ($\times 10^{16}$ molecules cm$^{-2}$) total columns over IGP, South Asia (rectangle: 20°N-32°N, 70°E-95°E) and (b) Scatter plot between annual averaged IASI and MOZART-4 (11:30 am) simulated NH$_3$ ($\times 10^{16}$ molecules cm$^{-2}$) total columns over NCP, East Asia (rectangle: 30°N-40°N, 110°E-120°E).

**Revision: Description added in revised version in section 2.3 and figure in the revised supplement**

We compared column to column, as IASI retrieval algorithm only provides total columns. We made un-weighted average distributions using all the morning IASI measurement available, following the recommendation for the use of the dataset provided in Van Damme et al. (2017). Furthermore, considering satellite overpass (9:30 am) time with the nearest model timestep (11:30 am) does not show significant change in simulated annual mean $NH_3$ tropospheric column. If we compare satellite and model at the nearest timestep, the Normalised Mean Bias (NMB) over South Asia is reducing by 6 % and over East Asia it is increasing by 6 % (Fig S1 (a) and (b) in the supplement). As our aim of the study is focussed on monthly, seasonal and annual data, we consider that IASI provides representative monthly, seasonal and annual means, despite possible biases introduced by lacking days of data due to cloud cover (Van Damme et al., 2014b, 2014a, 2015). Thus, we have qualitatively compared IASI $NH_3$ columns with modeled $NH_3$ columns since the IASI-$NH_3$ retrieval does not produce an averaging kernel to properly weight the model values (Clarisse et al., 2009).

[Figure]

Figure S1. (a) Scatter plot between annual averaged IASI and MOZART-4 (11:30 am) simulated $NH_3$ ($\times 10^{16}$ molecules $cm^{-2}$) total columns over IGP, South Asia (rectangle: 20°N-32°N, 70°E-95°E) and (b) Scatter plot between annual averaged IASI and MOZART-4 (11:30 am) simulated $NH_3$ ($\times 10^{16}$ molecules $cm^{-2}$) total columns over NCP, East Asia (rectangle: 30°N-40°N, 110°E-120°E).

**R2C2:**

**Given the agricultural practices in India, is it warranted to use a flat emission cycle across the year?**

- We agree with the reviewer's comment. A more realistic seasonal cycle of emissions of ammonia is needed for the simulations involving agriculture-based country like India. The EDGAR-HTAP inventory certainly lacks this information. We aim to improve the inventory by including such a seasonal cycle for emissions of ammonia in our future studies.

**R2C3:**

**The paper is severely hampered by the coarse comparison and I am afraid that the comparison methodology may impact the systematic differences seen in this paper. The differences between overpass time and a daily mean for instance relate to the daylength (variability) and associated mixing, diurnal emission cycle, frequency and kind of precipitation events, etc. I would have like to see an analysis/consideration of such factors in this paper. Part of the observations might be useful for this purpose.**

– We understand reviewer's concern with regards to the comparison methodology. We have addressed this issue in reply to the reviewer's comment R2C2.

**R2C4:**

**The discussion does not include a comparison to other modelling studies evaluating ammonia levels across Asia or studies on ammonia life time.**

- There are very few studies carried out in Asia similar to Clarisse et al. (2009) which have evaluated ammonia levels and compared model simulations with satellite retrievals. In recent study, it is shown that higher summer-time temperature along with the higher Nitrogen (N) fertilizer application rate could cause high $NH_3$ emissions resulting in the high $NH_3$ columns over Asia particularly during June-July-August (JJA) (Wang et al., 2020). Studies discussing on ammonia lifetime are already mentioned in the discussion part of the manuscript.

**Revision:**

Higher summertime temperature along with the higher N fertilizer application rate could cause high $NH_3$ emissions, resulting in the high $NH_3$ columns over Asia (Wang et al., 2020).

**R2C5:**

**I could identify many grammar mistakes in the english language use. The author list includes native speakers and I would like to urge to perform a careful language check.**

- A careful check for grammar has been done.

**Minor comments:**

**R2C6:**

**Abstract: Please use past tense for the method description**

Accepted.

**Revision:**

Limited availability of atmospheric ammonia (NH$_3$) observations, limits our understanding of control on its spatial and temporal variability and its interactions with ecosystems. Here we used the Model for Ozone and Related chemical Tracers (MOZART-4) global chemistry transport model and the Hemispheric Transport of Air Pollution version-2 (HTAP-v2) emission inventory to simulate global NH$_3$ distribution for the year 2010. We present a first comparison of the model with monthly averaged satellite distributions and limited ground-based observations available across South Asia. The MOZART-4 simulations over South Asia and East Asia were evaluated with the NH$_3$ retrievals obtained from the Infrared Atmospheric Sounding Interferometer (IASI) satellite and 69 ground-based air-quality monitoring stations across South Asia and 32 ground based monitoring stations from the Nationwide Nitrogen Deposition Monitoring Network (NNDMN) of East Asia. On the basis of model simulations and satellite observations, we identify the northern region of South Asia (Indo-Gangetic Plain, IGP) as a hotspot for NH$_3$ in Asia. In general, a close agreement is found between yearly-averaged NH$_3$ total columns simulated by the model and IASI satellite measurements over the IGP of South Asia (r=0.85) and North China Plain (NCP) of East Asia (r=0.88). However, the MOZART-4 simulated NH$_3$ column is seen to be substantially greater over South Asia than East Asia, as compared with the IASI retrievals, which show smaller differences. The model simulated surface NH$_3$ concentrations are lesser vis-a-vis the surface NH$_3$ measured by the ground based observation stations across South and East Asia in all the seasons, although the uncertainties prevail in the available surface NH$_3$ measurements. Overall, the comparison of East Asia and South Asia using both the MOZART-4 model and the satellite observations show smaller NH$_3$ columns in East Asia compared to South Asia for comparable emissions, indicating rapid dissipation of NH$_3$ due to secondary aerosol formation, which can be explained by higher emissions of acidic precursor gases in East Asia.

**R2C7:**

**Introduction The introduction focusses mostly on the contribution of different agricultural activities to emission estimates in south and east Asia. The challenges with respect to the emission estimation, spatial and temporal emission variability, chemistry**

**transcription modelling and model-satellite comparison are not focused on although these are relevant to the paper and partly addressed. I would like to ask the authors to address these issues in the intro.**

Accepted.

**Revision: Modified in the introduction section**

Van Damme et al. (2015a) attempted first to validate Infrared Atmospheric Sounding Interferometer IASI-NH$_3$ measurements using existing independent ground-based and airborne data sets. This study doesn't include comparison of ground-based NH$_3$ data sets with IASI measurements particularly over South Asia (India) due to limited availability of NH$_3$ measurements. Liu et al. (2017a) estimated the ground-based NH$_3$ concentrations over East Asia, combining IASI-NH$_3$ columns and NH$_3$ profiles from MOZART-4 and validated it with forty four sites of Chinese Nationwide Nitrogen Deposition Monitoring Network (NNDMN). Previous studies, based on satellite observations have suggested that the high NH$_3$ loading over the IGP region during summer is caused by high NH$_3$ emissions from intensive agricultural activities (Clarisse et al., 2009; Van Damme et al., 2015b), however validation of satellite retrievals over South Asia was largely missing. In one of the recent study over South Asia, analyses of seasonal and interannual variability of atmospheric NH$_3$ using IASI observations revealed large seasonal variability in atmospheric NH$_3$ concentrations which were equivalent with highest number of urea fertilizer plants. This study highlights the importance of role of agriculture statistics and fertilizer consumption/application in determining ammonia concentration in South Asia (Kuttippurath et al., 2020). Available global ammonia emission inventory does not include a comprehensive bottom up NH$_3$ emissions for South Asia compared to East Asia to be suitable for input to atmospheric models by taking into consideration actual statistical data of various NH$_3$ sources such as livestock excreta, fertilizer application, agricultural soil, nitrogen-fixing plants, crop residue compost, biomass burning, urine from rural populations, chemical industry, waste disposal, traffic, etc which is currently missing (Behera et al., 2013; Huang et al., 2012; Janssens-Maenhout et al., 2015; Li et al., 2017; Zhang et al., 2010). A recent study by Wang et al. (2020) examined the NH$_3$ column observed over the IGP during summer using regional model driven with MIX emission inventory. The study suggested that large agriculture activity and high summer temperature contributes to high NH$_3$ emission fluxes over IGP which leads to large total columns. However, for estimating reliable influence of NH$_3$ on air

quality, updated emission inventory as per the source activity is essential following, which is lacking for South Asia (Han et al., 2020). Furthermore, some studies over South Asia (Datta et al., 2012; Mandal et al., 2013; Saraswati et al., 2019; Sharma et al., 2012, 2014) have reported site-specific analyses for $NH_3$ (ground based measurements), but are limited to a few years and scarce.

**R2C8:**

**Line 43: chemical should be synthetic**

Accepted.

Removed word "chemical" and replaced with "synthetic".

**Revision:**

Specifically, ammonia ($NH_3$) emitted from various agricultural activities, such as use of synthetic fertilizers, animal farming, etc., together with nitrogen oxides (NOx) is one of the largest sources of reactive nitrogen (Nr) emission to the atmosphere.

**R2C9:**

**Line 50: 64 % of total means total global? if yes line 53 repeats this statement**

- No, not globally, South and East Asia together accounted for an estimated 64 % of the total amount of $NH_3$ emissions in the South-east part of South Asia during 2000-2014 (Xu et al., 2018).

**Revision:**

India and China together accounted for an estimated 64 % of the total amount of $NH_3$ emissions in the South-east part of Asia during 2000-2014 (Xu et al., 2018).

**R2C10:**

**Line 60-62: could you use the recent edgar numbers or thise from v4.3? Should these statements be presented with the global comparison the paragraph above?**

- Recent emission of EDGAR v4.3.2 estimated increasing emission of 59 teragram (Tg) over the period of 1970-2012 (EDGAR, 2019). While which me mentioned in the manuscript is 49.3 Tg of $NH_3$ estimate provided by EDGAR v4.2. Global comparison

shows that EDGAR v4.3 NH$_3$ emission estimates are higher than v4.2 (EDGAR, 2019).

- To avoid confusion and to improve flow of the introduction part we have removed this statement which seems to be insignificant.

**R2C11:**

**Line 63 and 67 are in direct contradiction to each other**

- Line no. 63 says In India, around 50 % of total NH$_3$ emissions is estimated from the fertilizer application and remaining from livestock and other NH$_3$ sources. However, in case of fertilizer application, especially urea alone contributes more than 95 % to the fertilizer demand, consumption (Fertilizer Association of India annual report 2018-19), and more than 90 % to the NH$_3$ emissions (Sharma et al., 2008).

- We have removed this statement to avoid confusion.

**R2C12:**

**Data and methodology Line 85: this sentence implies only trace gases were modelled, which is not the case I guess**

- Yes only trace gases were modelled (NH$_3$, O$_3$, NO$_x$ and CO), but our focus on this study is only NH$_3$ simulations.

**R2C13:**

**Line 97: Does Mozart use a land use mosaic within a gridcell? Or dominant LUC? How do the wesely land use classes match those in the domain? Were the latter updated?**

- Land use cover (LUC) maps used in MOZART-4 are based on the Advanced Very High Resolution Radiometer (AVHRR) and Moderate Resolution Imaging Spectroradiometer (MODIS) data as used in NCAR Community Land Model (CLM). MOZART-4 represents the land surface as a hierarchy of sub-grid types: glacier, lake, wetland, urban and vegetated land. The vegetated land is further divided into a mosaic of Plant Function Type (PFTs). These same maps are used dry deposition calculations. For more detail refer to Emmons et al. (2010); Oleson et al. (2010); Lawrence and Chase (2007).

**R2C14:**

**Line 122: didn't you use emissions of all sectors?**

- HTAP-v2 covers only 5 sectors for $NH_3$ emissions from agriculture, energy, transport and industry for the year 2010, rest two sectors aircraft and international shipping is not considered for $NH_3$ emissions.

**R2C15:**
**Line 133: cow dung is not fossil**

Accepted.

Removed word "cow dung" and replaced with "biomass combustion".

**Revision:**

Minor contributions from the residential sector are also observed for the Asian countries due to use of biomass combustion and coal burning which is also included in the emissions.

**R2C16:**

**Results: Line 226: the methodology describes that nitrate is present – please explain**

Yes, accepted.

Removed Line 226: In MOZART-4 chemistry, nitrate is absent and modified the statement as follows:

**Revision:**

In MOZART-4 chemistry, equilibrium simplified aerosol model (EQSAM)-Metzger et al. (2002) followed the assumptions which are limited to the ammonium-sulfate-nitrate-water system which is valid for only inorganic salt compounds. Since mineral cations and organic acids were neglected uncertainty can be associated in dry and wet deposition scheme which can result in overestimation (Metzger et al., 2006) (Emmons et al., 2010).

**R2C17:**

**250: the model has no maximum emissions in summer as antrop is flat and soil is a few percent of total, so this statement seems incorrect**

Accepted.

**Revision:**

This means that larger NH$_3$ emissions especially from soil may be expected in warm summer conditions than in winter, which is well represented in the emission estimate (soil) over both East and South Asia (Fig. 2). Although soil is few percent of total ammonia emissions its alkaline nature may emits higher NH$_3$ in the atmosphere (W. and R., 2004).

**R2C18:**

**Figure 2: the scale on the upper left figure is misleading. It seems a seasonal cycle where it is basically flat.**

Accepted.

**Revision: Figure 2 is modified**

[revised manuscript text omitted]